# Hypoxia-driven remodeling of SELENOP⁺ macrophages shapes T cell dynamics and promotes ovarian cancer metastasis

Qing Liu [1,8], Chenzhao Feng [2,3,4,8], Tianhao Wu [1], Siyang Zhang [1], Xinyi Wang [1], Qian Zhao[5], Xueying Song [1], Shuangyan Liu [1], Linru Quan [1], Yuli Zhang [1], Shimin Zhang [1], Bin Yang [2,3], Jixin Li[1], Gang Chen [2,3], Xuanzhang Huang [6,7] ✉, Chaoyang Sun [2,3] ✉ & Xin Zhou [1] ✉

High-grade serous ovarian cancer (HGSOC) is characterized by extensive transcoelomic dissemination and the accumulation of ascites. However, how site-specific tumor microenvironment (TME) drives progression remains unknown. Here we show the co-occurrence and spatial co-localization of *SELENOP*⁺ macrophages and precursor exhausted CD8⁺ T cells and demonstrate that *SELENOP*⁺ macrophages activate T cells via selenoprotein P in vitro and in vivo. We further identify a dynamic transition in the *SELENOP*⁺/*SPP1*⁺ macrophage populations as tumor metastasis, driven by increased hypoxia malignant epithelial cells through VEGFA-EPHB2 signaling. We also reveal that anti-VEGFA intervention controls ovarian tumor growth by increasing *SELENOP*⁺ macrophages and cytotoxicity of CD8⁺ T cells in vivo. Taken together, these findings spotlight the role of tumor-induced TME remodeling in subverting immune-mediated tumor control and thus facilitating HGSOC metastasis in females. Collectively, our results provide a foundation for the development of targeted therapeutic interventions aimed at impeding HGSOC metastatic trajectory.

Ovarian cancer (OC) is the most lethal gynecological malignancy[1], typically diagnosed at an advanced stage, marked by extensive transcoelomic dissemination and the accumulation of ascites[2–4]. High-grade serous ovarian cancer (HGSOC) is the most prevalent and aggressive subtype of OC[5]. Implantation metastasis serves as the primary cause of complications and mortality in HGSOC patients[3]. Although several novel therapeutic strategies, such as anti-angiogenesis therapies, poly (ADP-ribose) polymerase inhibitors (PARPi) and immunotherapies, have been introduced[6], the disappointing response rates and the development of resistance in HGSOC patients underscore the urgent need to further investigate the spatial heterogeneity of the tumor ecosystem that contributed to treatment failures in HGSOC[7–11].

Multiple ovarian cancer studies have shown that different metastasis sites are closely related to ovarian cancer staging, treatment

[1]Department of Obstetrics and Gynecology, Shengjing Hospital of China Medical University, Shenyang, Liaoning, China. [2]Department of Obstetrics and Gynecology, National Clinical Research Center for Obstetrics and Gynecology, Tongji Hospital, Tongji Medical College, Huazhong University of Science and Technology, Wuhan, China. [3]Key Laboratory of Cancer Invasion and Metastasis (Ministry of Education), Hubei Key Laboratory of Tumor Invasion and Metastasis, Tongji Hospital, Tongji Medical College, Huazhong University of Science and Technology, Wuhan, China. [4]NHC Key Laboratory of Prevention and Treatment of Central Asia High Incidence Diseases (First Affiliated Hospital of Shihezi University), Shihezi, Xinjiang, China. [5]Institute of Health Sciences, China Medical University, Shenyang, Liaoning, China. [6]Department of Surgical Oncology and General Surgery, The First Hospital of China Medical University, Shenyang, Liaoning, China. [7]Key Laboratory of Precision Diagnosis and Treatment of Gastrointestinal Tumors, Ministry of Education, China Medical University, Shenyang, Liaoning, China. [8]These authors contributed equally: Qing Liu, Chenzhao Feng. ✉e-mail: xzhuang@cmu.edu.cn; suncydoctor@gmail.com; xzhou@cmu.edu.cn

resistance, and even prognosis[12–14]. We previously documented site-specific alterations in the tumor-infiltrating lymphocytes (TILs) status using single-cell RNA-seq (scRNA-seq) and paired T cell receptor sequencing (scTCR-seq), and identified that ovarian tumors were enriched with clonally expanded exhausted CD8[+] T cells, whereas omental lesions were predominantly infiltrated by non-tumor-specific bystander T cells[15]. Recently, it is widely recognized that anatomical sites and genetic predisposition collectively determine the evolutionary phenotypic divergence and immune resistance mechanisms in the tumor microenvironment (TME) of HGSOC[16,17]. However, the spatiotemporal heterogeneity of the tumor microenvironment associated with HGSOC metastasis, including the integrated analysis of immune cell diversity and intra-tumor heterogeneity, as well as the underlying cellular and molecular mechanisms remains to be thoroughly explored.

In this work, to provide a comprehensive analysis of the spatiotemporal heterogeneity of the tumor microenvironment during HGSOC metastasis and to explore the cellular and molecular events involved, we delineate the transcriptional architecture across multiple spatiotemporal anatomical sites pertinent to the metastasis HGSOC, utilizing single-cell resolution to illuminate the profound heterogeneity and site-specific distinctions of immune and malignant cell populations within the TME. Through an integrative, multi-scale analytical approach—encompassing scRNA-seq, scTCR-seq, spatial transcriptomics (ST), whole-exome sequencing (WES), bulk RNA sequencing (RNA-seq), organoid culture systems, and orthotopic syngeneic models—we identify the tumor microenvironmental remodeling associated with HGSOC metastatic progression. Mechanistically, our findings demonstrate that hypoxia-driven malignant epithelial cells reprogram *SELENOP*[+] macrophages through VEGFA-EPHB2 signaling, thereby impairing the cytotoxic function of precursor exhausted CD8[+] T cells within the solid tumor microenvironment, ultimately facilitating HGSOC metastasis. This research provides crucial theoretical foundations and insights for the development of more effective clinical therapeutic strategies against HGSOC.

## Results

### A single-cell RNA-seq atlas reveals site-specific characteristics and remolding of the tumor ecosystem in metastatic HGSOC

To comprehensively decipher the complex tumor ecosystem during metastasis, we conducted scRNA-seq on 34 samples from multiple spatiotemporal anatomical sites related to HGSOC metastasis, obtained from 17 patients. The eight spatiotemporal anatomical sites include: (1) the solid tumor sites (designated as solid sites) which encompass adnexal tumors from early-stage (EAT) and late-stage (LAT) HGSOC, as well as matched metastatic sites (Met), including omental (Met.Ome) and peritoneal metastases (Met.Per); (2) the malignant fluid sites (designated as fluid sites) included peritoneal lavage fluid from early-stage HGSOC (PLF.EOC) and ascites from advanced patients (Ascites); (3) the peritoneal lavage fluid from benign uterine fibroids (PLF.UF) and normal post-menopausal ovarian tissue (Nor.Ovr) (Fig. 1a and Supplementary Data 1). Meanwhile, we also integrated the spatial transcriptomic profiling of about 2 million spots across distinct anatomical sites in our previous work (Fig. 1a and Supplementary Data 2). To assess global transcriptomic variation across all samples from scRNA-seq cohort, we performed principal component analysis (PCA) of pseudo-bulk samples. The results revealed that the primary driver of heterogeneity in the HGSOC tumor microenvironment was the distinction between solid and fluid sites, as samples from these two compartments were clearly separated (Fig. 1b).

Following quality control and doublet removal, we cataloged 138,866 high quality single cells into five major cell lineages and assessed their tissue preference patterns by Ratio of Observed to Expected ($R_{o/e}$) analysis (Fig. 1c–e, Supplementary Fig. 1a, Supplementary Data 3, 4 and Supplementary Note 1). Given that T/NK cells,

myeloid cells, and epithelial cells constitute the major cellular components at both solid sites (EAT, LAT and Met) and fluid sites (PLF.EOC and Ascites) (Fig. 1f and Supplementary Fig. 1b), we next focused our analysis on the dynamic cellular states of these populations associated with metastatic HGSOC.

### Reduced *GZMH*[+] precursor exhausted CD8[+] T cells with cytotoxic effects in the solid tumor microenvironment of metastatic HGSOC

Using paired scRNA-seq and scTCR-seq analyses, our previous work demonstrated that CD8[+] T cells were tumor-specific in adnexa but functioned as bystanders in the omental metastasis of HGSOC[15]. However, the dynamic changes in functional states and differentiation of CD8[+] T cells during HGSOC metastasis remain to be fully understood. Here, we first identified CD4[+] T cells, CD8[+] T cells, NK cells, and cycling cells via unsupervised clustering (Supplementary Fig. 2a). Further reclustering led to the identification of nine CD8[+] T cell subclusters (Fig. 2a) characterized by distinct gene signatures (Fig. 2a and Supplementary Data 5).

We first uncovered unconventional and conventional CD8[+] T cells and their preferential anatomical sites (Fig. 2a, Supplementary Fig. 2b, c and Supplementary Note 2). We then aimed to seek exhausted CD8[+] T cells, which expressed multiple inhibitory receptors, including *PDCD1* (encoding PD-1), *CTLA4, LAG3, TIGIT, LAYN*, and *HAVCR2* and largely attributed to anti-tumor activity[18]. Accordingly, subcluster T06_CD8T-CXCL13, exhibited a transcriptional signature indicative of exhaustion and was thus classified as exhausted CD8[+] T cells (Tex) (Fig. 2a, b). Additionally, the T04_CD8T-GZMH subpopulation, characterized by modest exhaustion scores but distinguished by elevated levels of *GZMH, GZMA*, and *CCL4* (Fig. 2a, b), reminds us of the precursor exhausted T (Tpex) cells. Tpex cells have been reported as a specialized subset of CD8[+] T cells that develop into terminally exhausted effector cells, aiding in tumor control[19]. We further used previously reported signature genes of terminally Tex and Tpex to characterize the conventional CD8[+] T clusters in our study[20]. CD8[+]*GZMH*[+] (T04) and CD8[+]*CXCL13*[+] (T06) cells exhibited the greatest similarities with the Tpex and Tex cell populations, respectively (Fig. 2b). Moreover, we also observed the elevated expression of *TCF7* and *EOMES* (known to be highly expressed by precursor exhausted T cells[21]) in CD8[+]*GZMH*[+] (T04) compared with CD8[+]*CXCL13*[+] (T06) (Fig. 2c). Therefore, subcluster CD8[+]*GZMH*[+] (T04) was classified as Tpex.

Then we assigned cell subcluster identities from ours on an independent OC cohort with matched scRNA-seq and scTCR-seq[22] and evaluated TCR clonotypes of CD8[+] T clusters using STARTRAC indices[23]. CD8[+]*CXCL13*[+] Tex (T06) and CD8[+]*GZMH*[+] Tpex cells (T04) exhibited high clonal expansion index (Supplementary Fig. 2d), indicating that these cells could recognize more tumor antigens. Also, TCR clonotypes of T04 had the most transition index with tumor-reactive T06 cells (Fig. 2d), suggesting that CD8[+]*GZMH*[+] T cells may represent a critical reservoir for the emergence of tumor-specific CD8[+] T cells[24]. Cytotoxic CD8[+] T cells reactive against tumor-specific neoantigens eradicate cancer cells by producing cytokines, chemokines, and granzymes[25]. As expected, T04 and T06 exhibited elevated cytotoxic signature scores and tumor specific signature scores (Fig. 2e, f), suggesting that these cells might be activated under chronic antigen stimulation.

Numerous investigations have established a correlation between clinical outcomes and Tpex cells, which subsequently differentiate into a terminal Tex subset[20,26–28]. To track *GZMH*[+] Tpex dynamics in HGSOC metastasis, we performed $R_{o/e}$ analysis and found enrichment of T06, which are potentially differentiated from T04, within solid tumor sites, with higher prevalence detected in the EAT and LAT groups in comparison to the Met group (Supplementary Fig. 2c). This result prompted us to further characterize CD8[+] T cell lineages with cytotoxic

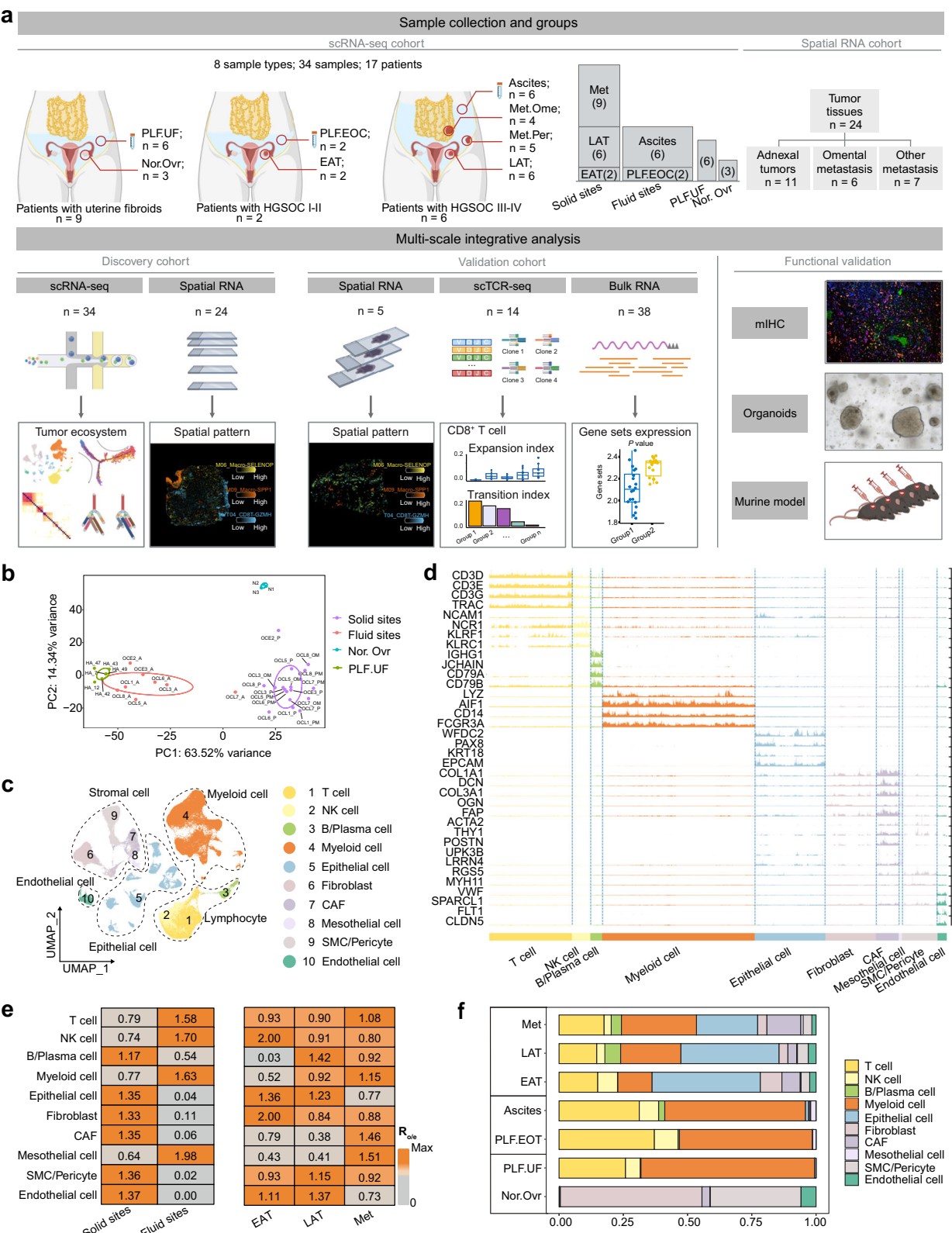

potential that respond to TCR-mediated stimulation in the solid tumor microenvironment. Thus, we performed pseudotime analysis on CD8+ T cell subsets T01, T02, T04 and T06, excluding T03 and T07 (Fig. 2g). Two distinct differentiation trajectories for CD8+ T cells were observed: one path culminated in a memory and precursor exhausted state (cell fate 1), predominantly characterized by T02 and T04 cells, while the alternative trajectory progressed toward a terminally

exhausted state (cell fate 2). The latter pathway was marked by an enrichment of T06 at its end, with T04 appearing at an intermediate position along this trajectory (Fig. 2g). Cytotoxic, tumor specific, PD-1 and immune response signatures were elevated at end of fate 2, in line with terminal dysfunction (Fig. 2h). Consistent with our previous findings, Met-derived CD8+ T cells enriched in the early pseudotime were likely to be in naïve state[15] (Fig. 2i). Moreover, EAT-derived CD8+

**Fig. 1 | A single-cell RNA-seq atlas reveals site-specific characteristics and remolding of the tumor ecosystem in metastatic HGSOC. a** Flowchart depicting the overall study design. PLF.UF peritoneal lavage fluid from benign uterine fibroids, Nor.Ovr normal post-menopausal ovarian tissue, PLF.EOC peritoneal lavage fluid from early-stage HGSOC, EAT adnexal tumors from early-stage HGSOC, Ascites ascites from advanced HGSOC patients, Met.Ome omental metastases, Met.Per peritoneal metastases, LAT adnexal tumors from late-stage HGSOC, Met metastatic sites, mIHC multiplex immunohistochemistry. Created in BioRender. Song, X. (2025) https://BioRender.com/8iuultd. **b** The principal component analysis (PCA) plot showing the PC1 vs. PC2 projection of all samples at the pseudo-bulk RNA-seq level. **c** Uniform Manifold Approximation and Projection (UMAP) plots of the clustering of main cell types from all samples of scRNA-seq cohort. Cell lineages are highlighted with black dotted lines. Individual cells (dots) are colored by clusters. NK natural killer, CAF cancer-associated fibroblasts, SMC smooth muscle cells. **d** Track plot indicating selected marker genes in each cell type. **e** Tissue preference of each major cluster across solid and fluid sites (left), EAT, LAT and Met groups (right), estimated by $R_{o/e}$ ($p = 2.2 \times 10^{-16}$). $p$ values are calculated by the two-sided chi-squared test. **f** Cell proportions of major cell types in different groups, colored by corresponding cell type colors in (**c**). For **b**–**d**, **f**, $n$ = all 34 scRNA-seq cohort samples, biological replicates. For **e**, $n$ = 25 scRNA-seq cohort samples except for Nor.Ovr and PLF.UF, biological replicates. Source data are provided as a Source Data file.

T cells showed more enrichment in the terminally functional state compared to LAT-derived cells, either in cell fate 1 and 2 (Fig. 2i and Supplementary Fig. 2e). Although T04 abundance changed little across solid sites, its cytotoxic/tumor-specific functions, including *GZMB* and *PRF1* expression, were markedly reduced in LAT and Met compared with EAT (Fig. 2j and Supplementary Fig. 2f). In contrast, T06 functions remained stable across sites (Fig. 2k). These results suggest that ovarian cancer progression involves loss of cytotoxicity and tumor reactivity in Tpex cells, alongside reduced exhausted T cell numbers during metastasis.

To delineate the heterogeneity of CD4$^+$ T cells, we performed transcriptional profiling and identified distinct subsets with characteristic gene expression signatures (Supplementary Fig. 2g–i, Supplementary Data 5 and Supplementary Note 3). Previous studies have reported that Tregs are more frequently observed in HGSOC compared to other histological subtypes and mediate poor prognosis[29–31]. We next focused on Treg dynamics across disease progression and observed a stepwise increase in the enrichment of CD4$^+$*FOXP3*$^+$ Treg from EAT to LAT and ultimately to Met (Supplementary Fig. 2j). Together, these findings highlight a potential pro-metastatic role for Tregs in HGSOC, likely mediated by their robust immunosuppressive functions within the solid tumor microenvironment.

### *SELENOP*$^+$ macrophages co-localized with CD8$^+$*GZMH*$^+$ Tpex and predict better prognosis

Tumor-associated macrophages (TAMs) demonstrate significant plasticity and heterogeneity, challenging the traditional M1/M2 polarization paradigm, with studies showing their essential role in modulating antitumor T cell responses within the tumor microenvironment, including in ovarian cancers[32–37]. However, the complexity of TAMs in HGSOC during metastasis remains largely unexplored. We analyzed a total of 46,396 myeloid cells using unsupervised clustering (Supplementary Data 3 and 4). We identified dendritic cells (DCs), monocytes, TAMs, neutrophils and mast cells according to their markers with their spatial preference (Fig. 3a, Supplementary Fig. 3a, b, Supplementary Data 5 and Supplementary Note 4). Notably, *SELENOP*$^+$ macrophages (M06) and *SPP1*$^+$ macrophages (M09) were enriched in solid sites, whereas the other macrophage clusters were predominantly observed in fluid sites (Fig. 3b, c). Further analysis using KEGG enrichment, RNA expressions and signature scoring revealed that M06 preferentially exhibited antigen processing and presentation capabilities (Fig. 3d, e and Supplementary Fig. 3c). In contrast, M09 were involved in extracellular matrix remodeling and glycolysis and gluconeogenesis pathways (Fig. 3d and Supplementary Fig. 3c). Analysis of the TCGA-OV cohort showed that M06 were linked to favorable prognosis, whereas M09 were associated with poor outcomes (Fig. 3f and Supplementary Fig. 3d), suggesting antitumor and protumor roles, respectively.

We further deconvoluted the proportions of cell subpopulations for each sample in TCGA-OV based on subcluster profiles from our scRNA-seq dataset, which showed that *SELENOP*$^+$ macrophages, but not *SPP1*$^+$ macrophages, correlated with diverse T cell subsets, including CD8$^+$*GZMH*$^+$ Tpex (T04) (Fig. 3g). We also spatially mapped cell types in high-resolution spatial transcriptomics using cell2location[38]. Compared to *SPP1*$^+$ macrophages, *SELENOP*$^+$ macrophages showed stronger co-localization with CD8$^+$*GZMH*$^+$ Tpex (T04), particularly in adnexal sites, rather than in omental metastases, as revealed by our spatial transcriptomics data using a machine-learning method mistyR (Fig. 3h, i). Similar results were also observed in the validation spatial transcriptomics cohorts[39,40] using the 10x Visium platform (Supplementary Fig. 3e, f). Furthermore, multiplex immunohistochemistry (mIHC) also confirmed the close proximity of SELENOP$^+$ macrophages and CD8$^+$GZMH$^+$ T cells or CD8$^+$PD-1$^+$ T cells (Fig. 3j, Supplementary Fig. 3g and Supplementary Data 7). These findings suggest that *SELENOP*$^+$ macrophages are spatially linked with Tpex cells, prompting further investigation into their interactions.

### *SELENOP*$^+$ macrophages increased cytotoxicity of Tpex in vitro

Previous studies have reported that a subset of TAMs possesses antigen cross-presentation capabilities and efficiently activates antigen-specific CD8$^+$ T cells[41]. Similarly, the antigen processing and presentation signature score of *SELENOP*$^+$ macrophages were positively correlated with the *GZMB* expression (Spearman $p = 0.0097$, $R = 0.62$, 95% confidence interval (CI) = [0.125, 0.896], Supplementary Fig. 3h) and the cytotoxic signature scores (Spearman $p = 0.0061$, $R = 0.65$, 95% CI = [0.129, 0.960], Supplementary Fig. 3h) of T04. Additionally, *SELENOP*, the marker of M06, was primarily expressed in *SELENOP*$^+$ macrophages rather than other macrophage subsets or cell types (Supplementary Fig. 4a). *SELENOP* encodes a secreted protein, selenoprotein P (SELENOP), which contains up to 10 selenocysteine (Sec) residues[42]. To elucidate the effect of SELENOP on CD8$^+$ T cell function, CD8$^+$ T cells derived from female OT-1 mice (OT-1 T cells) activated by ovalbumin (OVA) peptide (257–264) were cultured with or without exogenous recombinant mouse SELENOP (rmSELENOP) for 48 h (Supplementary Fig. 4b). Dose-dependent changes in both GZMB and PRF1 levels were observed (Supplementary Fig. 4c). Moreover, rmSELENOP treatment also enhanced the tumor-specific cytotoxicity of OT-1 T cells, estimated by the apoptosis rates of the co-cultured OVA-expressing ID8 ovarian cancer cells (Supplementary Fig. 4d). Furthermore, rmSELENOP also enhanced the expression of GPX1 in OT-1 T cells, which is synthesized by selenium (Supplementary Fig. 4e).

To assess the functional impact of macrophage-derived SELENOP on CD8$^+$ T cells, we established macrophage-derived conditioned medium (CM) and CD8$^+$ T cell coculture system (Fig. 3k). *Selenop* expression in bone marrow-derived macrophages (BMDMs) was modulated via lentiviral-mediated overexpression (OE) or knockdown (KD) strategies (Supplementary Fig. 5a). Then, the CM from these BMDMs were collected and co-culture with OT-1 T cells for 2 days. We observed that CM from *Selenop*-overexpressing BMDMs (BMDMs-OE-Selenop) enhanced GZMB and PRF1 expression in OT-1 T cells, along with increased tumor cell killing capacity as demonstrated by T cell cytotoxicity assay (Fig. 3l and Supplementary Fig. 4f, g). In contrast, CM from *Selenop*-knockdown BMDMs showed a weaker cytotoxic response effect, with reduced both GZMB and PRF1 expression, as well as the cytotoxic activity of CD8$^+$ T cells, which could be reversed by

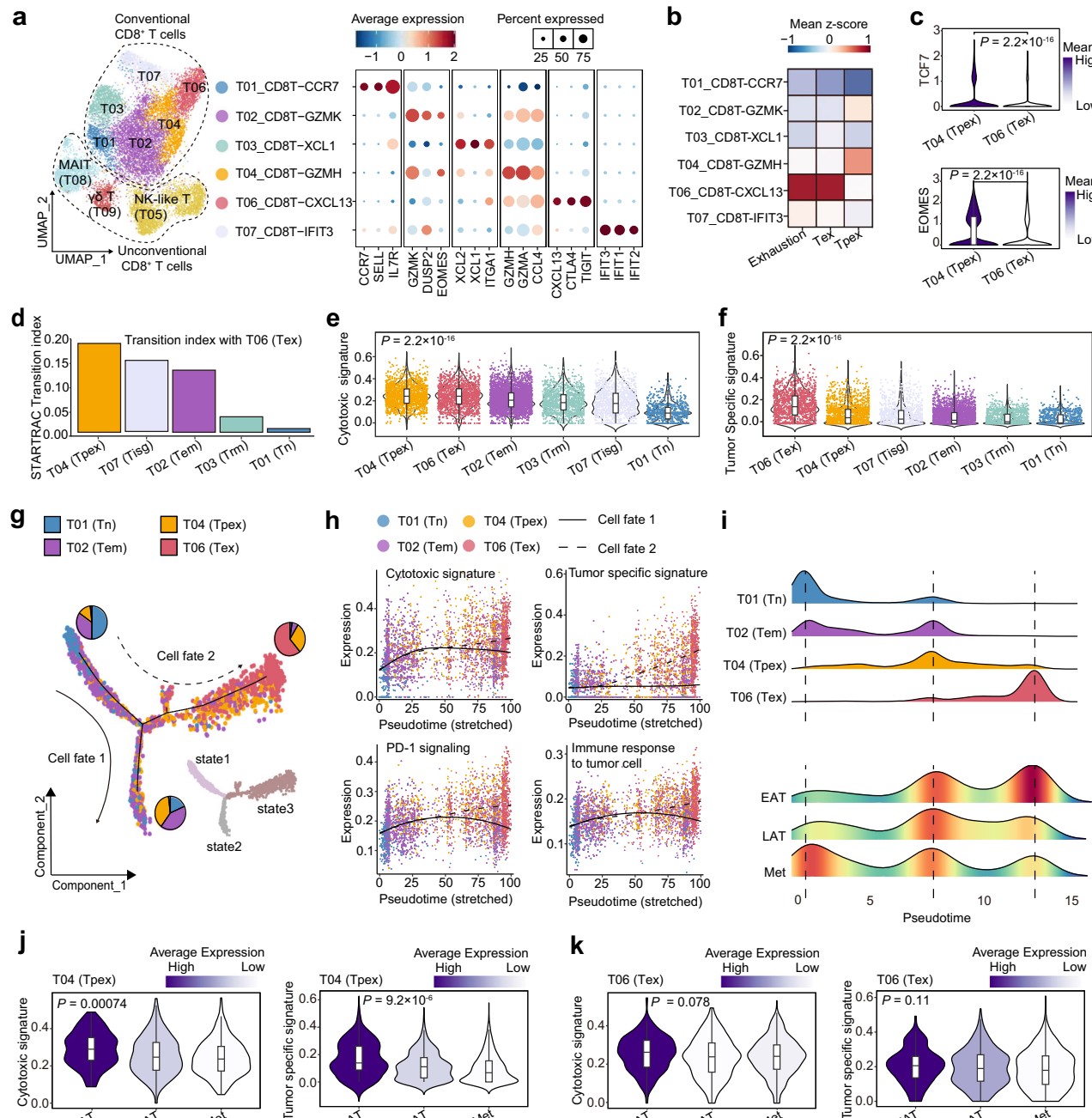

**Fig. 2 | Reduced cytotoxicity of *GZMH*+ precursor exhausted CD8+ T cells in the solid tumor microenvironment of metastatic HGSOC. a** UMAP plots depicting the clusters of CD8+ T cells, colored by cluster. Conventional and unconventional CD8+ T cells are highlighted with black dotted lines, respectively (left). Dot plot showing marker genes of conventional CD8+ T cell clusters (right). Dot size representing percent of expressing cells in each cluster and color represents *z*-score of normalized mean expression level of selected genes. **b** Heatmap indicating the expression levels of gene signatures of previously reported Tpex and Tex cell states, and exhaustion signature in conventional CD8+ T cell subtypes. Tpex precursor exhausted T cells, Tex exhausted T cells. **c** Violin plot showing *TCF7* and *EOMES* expression of T04 (Tpex) and T06 (Tex) colored by mean expression. *p* values are calculated by two-sided Wilcoxon test. **d** Bar plots showing transition index with Tex (T06) quantified by STARTRAC indices. Violin plot with cytotoxic (**e**) and tumor specific (**f**) signature scores, grouped and colored by identified conventional CD8+ T cell clusters. **g** The trajectory plot of selected CD8+ T cell clusters

of tumor sites. The pie charts showing the percentage of CD8+ T cell clusters in each state. **h** Curve plots showing dynamic expression of cytotoxic and tumor specific signature, PD-1 signaling (based on Reactome database) and immune response to tumor cell pathways (based on Gene Ontology database) along two cell fates. **i** Density plot showing the density patterns of T01 (Tn), T02 (Tem), T04 (Tpex), and T06 (Tex) (upper) or cells from different groups (lower). Violin plot showing cytotoxic and tumor specific signature scores of T04 (**j**) or T06 (**k**) across EAT, LAT and Met groups, colored by average expression. For **a**–**c**, **e**, **f**, *n* = all 34 scRNA-seq cohort samples, biological replicates. For **d**, *n* = 14 HGSOC samples (HRA002184), biological replicates. For **g**–**k**, *n* = 17 scRNA-seq cohort solid site samples, biological replicates. For **e**, **f**, **j**, **k**, box of violin plot represents median ± interquartile range, and the whiskers extend up to the minimum and maximum values. *p* values: two sided Kruskal−Wallis tests with Bonferroni post hoc test. Source data are provided as a Source Data file.

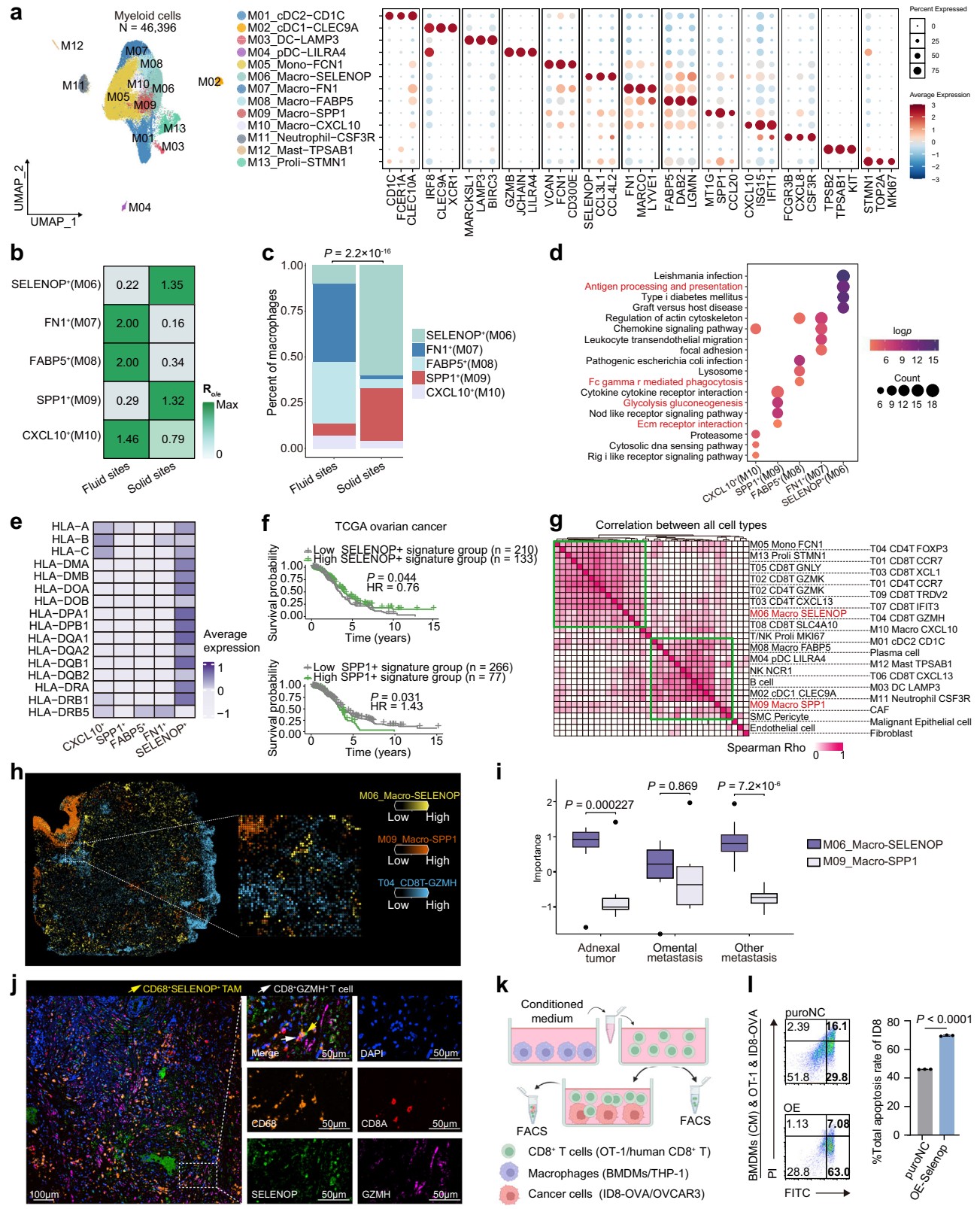

supplementation of rmSELENOP (Supplementary Fig. 4h–j). Parallel experiments using CM from THP-1 cells overexpressed *SELENOP* and PBMCs derived CD8+ T (human CD8+ T) cells of HGSOC patients yielded consistent results (Supplementary Figs. 4k–m and Supplementary Fig. 5b). To further evaluate the direct priming capacity of *SELENOP*+ macrophages on CD8+ T cells, we also performed the antigen presentation assay using OVA-loaded BMDMs, OVA-loaded BMDMs-OE-

Selenop or OVA-loaded BMDMs-KD-Selenop, cocultured directly with OT-1 T cells (Supplementary Fig. 4n). The results were also consistent with those observed in the macrophage-conditioned medium and CD8+ T cell co-culture system (Supplementary Fig. 4o–t).

Together, these results indicate that *SELENOP*+ macrophages possess immunostimulatory properties, potentially activating or maintaining the cytotoxicity of precursor exhausted CD8+ T cells

**Fig. 3 | *SELENOP*+ macrophages co-localized with CD8+*GZMH*+ Tpex and predict better prognosis. a** UMAP projection of 13 myeloid clusters (left). Dot plot showing expression patterns of selected genes (right). **b** Tissue preference of each macrophage cluster across fluid and solid sites estimated by the R$_{o/e}$ ($p = 2.2 \times 10^{-16}$). $p$ value is calculated by the two-sided chi-squared test. **c** Cell proportions of each macrophage cluster in fluid and solid sites. Each stacked bar represents each group, colored by corresponding cell type colors ($p = 2.2 \times 10^{-16}$). $p$ value is calculated by the two-sided chi-squared test. **d** Enriched pathways of each macrophage subset in KEGG databases. **e** Heatmap showing the expression levels of HLA-I and HLA-II genes in five macrophage clusters, colored by average expression. **f** The Kaplan−Meier overall survival curves of patients with HGSOC grouped based on the signature of the *SELENOP*+ macrophage signature (upper) and *SPP1*+ macrophage signature (lower). $p$ values were determined by log-rank test. HR hazard ratio. **g** Correlation among all cell types (TCGA-OV). **h** Representative spatial co-localizations of *SELENOP*+ macrophages with T04_CD8T-GZMH in tumor spots of slide from adnexal sites (left) and zoom-in image (right) in our spatial transcriptomics cohort. **i** Boxplot showing the importance (neighborhood scores) between *SELENOP*+ macrophages or *SPP1*+ macrophages and T04_CD8T-GZMH signature scores in spots of spatial transcriptomics data. $p$ values were calculated

by two-sided paired Student's $t$ test tests. Box represents median ± interquartile range, the whiskers extend up to the minimum and maximum values, statistical outliers shown as individual points. $n = 24$ spatial RNA cohort samples, biological replicates. **j** Representative immunofluorescence staining showing co-localization of CD68 (orange), SELENOP (green), CD8A (red), GZMH (magenta) and DAPI (blue) in HGSOC samples. Scale bars of each group, 100 μm (left) and 50 μm (right). The white arrow points to the CD8+GZMH+ cell, and the yellow arrow points to the CD68+SELENOP+ TAM. TAM tumor-associated macrophage. **k** Macrophage-derived conditioned medium (CM) and CD8+ T cell coculture system. FACS fluorescence-activated cell sorting, OVA ovalbumin. Created in BioRender. Song, X. (2025) https://BioRender.com/m4utusl. **l** The total apoptosis rate of ID8-OVA cells induced by corresponding CD8+ T cells. Data represent the mean ± SD, two-sided unpaired Student's $t$ test, $n = 3$ (biological replicates). BMDMs bone marrow derived macrophages, puroNC BMDMs transfected with negative control lentivirus, OE-Selenop BMDMs overexpressing *Selenop* after lentiviral transfection. For **a**, **d**, **e**, $n =$ all 34 scRNA-seq cohort samples, biological replicates. For **b**, **c** $n = 25$ scRNA-seq cohort samples except for Nor.Ovr and PLF.UF, biological replicates. Source data are provided as a Source Data file.

through SELENOP, in contrast with pro-tumorigenic effects from *SPP1*+ macrophages[43–45]. Thus, *SPP1*+ and *SELENO*P+ macrophages resembled two opposite functions in HGSOC.

## Site-specific phenotypes of macrophages and functional reprogramming of *SELENOP*+ macrophages in the solid tumor microenvironment of metastatic HGSOC

To further elucidate the dynamic alterations and differentiation of *SELENOP*+ and *SPP1*+ macrophages during the metastasis, we undertook an analysis of the tissue preference. As the disease progressed from EAT to LAT and then to Met, there was a gradual reduction in the proportion of *SELENOP*+ macrophages (M06) and a corresponding increase in *SPP1*+ macrophages (M09) (Fig. 4a and Supplementary Fig. 6a). Validation using external HGSOC scRNA-seq data[16] (syn52458609) also confirmed the similarity of M06 to the previously reported M2.SELENOP population (Supplementary Fig. 6b) and showed a modest enrichment of these M2.SELENOP macrophages in adnexal tumors compared to the metastatic lesions (Supplementary Fig. 6c). To further support this observation, we integrated our scRNA-seq dataset with the publicly available dataset GSE184880[46]. Consistently, this analysis recapitulated the observed trend, showing a progressive decrease in the inferred *SELENOP*+ macrophages and a concomitant increase in the inferred *SPP1*+ macrophages from EAT to LAT and ultimately to Met (Supplementary Fig. 6d). This trend of decreased *SELENOP*+ macrophages was further corroborated by mIHC using an independent sample set (Fig. 4b, c and Supplementary Data 7). These results elucidated that M06 and M09 not only serve opposing functional roles within the TME but also exhibited contrasting trends as the tumor progressed through metastasis.

We further characterized the dichotomy of macrophages enriched in solid tumor of HGSOC as the following attempts. First, pseudotime analysis was performed on monocyte-like cells (M05), *SELENOP*+ macrophages (M06) and *SPP1*+ macrophages (M09). This analysis revealed a distinct bifurcation: cell fate 1 dominated by *SPP1*+ macrophages (M09) and cell fate 2 dominated by *SELENOP*+ macrophages (M06) (Fig. 4d). We found that antigen processing and presentation, interferon gamma (IFNγ) response and *SELENOP* expressions were associated with fate 2 (*SELENOP*+ macrophages enriched) (Fig. 4e, f), whereas hypoxia, glycolysis, extracellular matrix degradation pathway and *SPP1* expressions were progressively upregulated along the fate 1 trajectory (*SPP1*+ macrophages enriched) (Fig. 4e, g). We also identified transcription factors driving TAM differentiation in HGSOC solid tumors. *SELENOP*+ macrophages (M06) showed activation of IRF8, CEBPA, and STAT1, whereas *SPP1*+

macrophages (M09) were enriched for HMGA2, MYB, and MX1 (Fig. 4h, i). IRF8, a transcription factor regulated by interferon[47], associated with inflammatory macrophage polarization, showed high specificity and activity in *SELENOP*+ macrophages (M06) (Fig. 4h, i), which were also positively correlated with *SELENOP* expression and the antigen processing and presentation signature scores of M06 (Spearman $p = 2.2 \times 10^{-16}$, $R = 0.3$, 95% CI = [0.272, 0.331], Supplementary Fig. 6e; $p = 2.2 \times 10^{-16}$, $R = 0.15$, 95% CI = [0.116, 0.180], Supplementary Fig. 6e). In addition, we observed that the expression of *HMGA2* and *MX1* increased along cell fate trajectory 1, while *IRF8*, *CEBPA*, and *STAT1* expression increased along cell fate trajectory 2 (Fig. 4j). Overall, these results suggested that TAM exhibited bidirectional differentiation pathways, and the phenotype of *SELENOP*+ macrophages is associated with IFNγ induction and IRF8 regulation. Functionally, IFNγ stimulation enhanced SELENOP expression in THP-1 macrophages, human monocyte-derived macrophages (MDMs) of HGSOC patients and BMDMs, confirming IFNγ regulation (Fig. 4k, l and Supplementary Fig. 6f–h). Selenium supplementation further boosted SELENOP, indicating dependence on selenium availability (Fig. 4k, l).

It is noteworthy that EAT-and LAT-derived cells preferentially localized at the terminal end of the pseudotime trajectory enriched with *SELENOP*+ macrophages, whereas Met-derived macrophages tended to reside in distinct phases enriched with *SPP1*+ macrophages (Supplementary Fig. 6i). Additionally, *SELENOP* expression (Fig. 4m), IRF8 activity and expression (Supplementary Fig. 6j) as well as antigen processing and presentation signature scores (Supplementary Fig. 6k) in *SELENOP*+ macrophages (M06) were lower in the Met group compared to the EAT and LAT groups. These findings suggest site-specific differentiation tendencies of TAMs, potentially associated with interferon and IRF8. On the contrary, the matrix remodeling score of *SPP1*+ macrophages (M09) increased in the Met group compared to the EAT and LAT groups (Supplementary Fig. 6k), further revealing that *SPP1*+ macrophages might promote HGSOC metastasis.

## Malignant epithelial cells transcriptional heterogeneity and genomic alterations

According to the seed and soil theory of metastasis, the spread of tumors results from the interplay between malignant cells and the microenvironments[48]. In addition to the immune landscape, malignant epithelial cells themselves exhibit substantial transcriptional heterogeneity and adaptive plasticity, which are critical in shaping the metastatic niche[49,50]. Intratumor heterogeneity in previous scRNA-seq studies is reflected by expression programs, and the consensus among them, termed meta-programs (MPs), represent fundamental features of tumor biology[51].

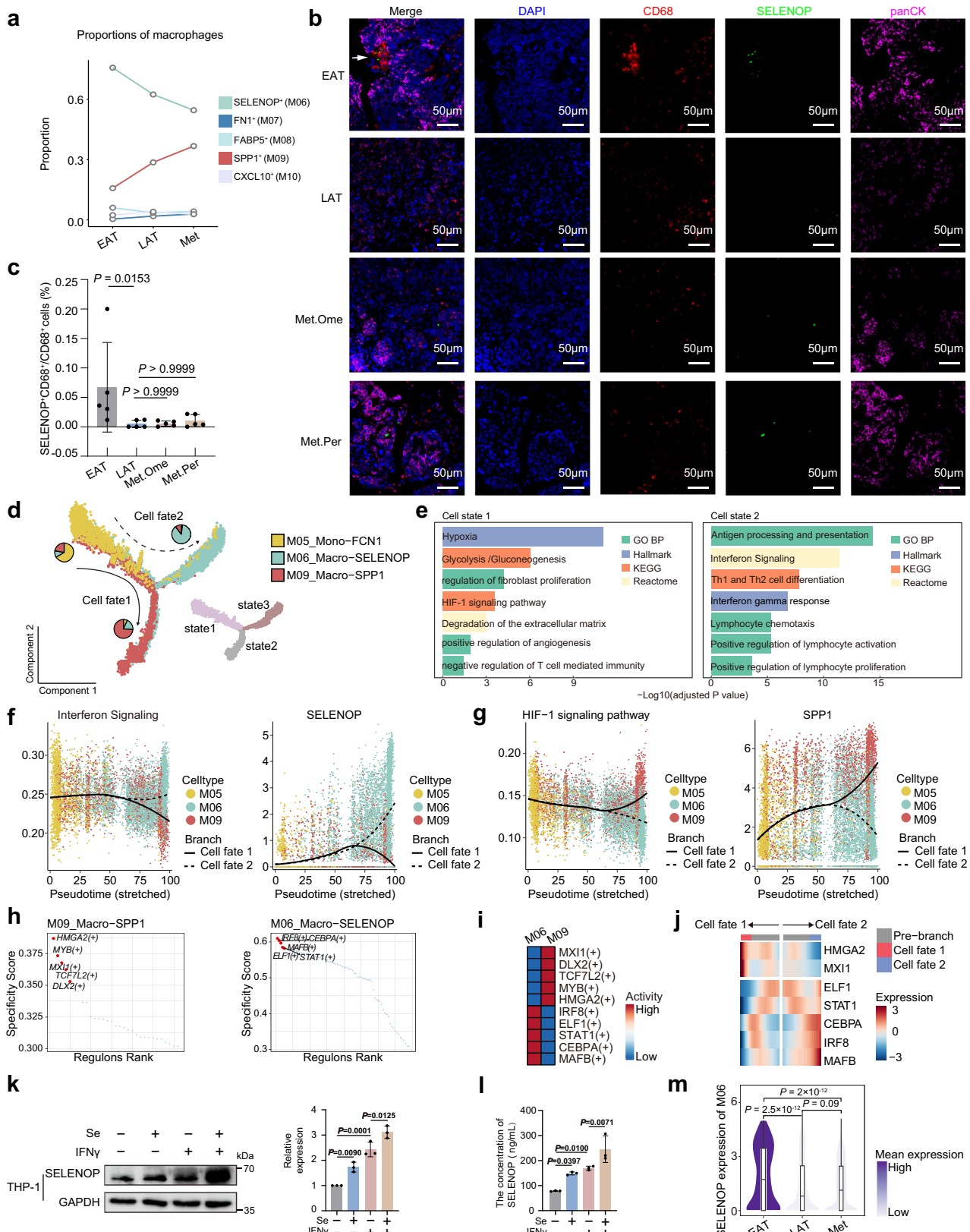

In our scRNA-seq data, inferCNV analysis identified 20,956 malignant epithelial cells (mEpi) (Supplementary Figs. 7–10 and Supplementary Data 4). To characterize the heterogeneity within these ovarian malignant epithelial cells, we applied non-negative matrix factorization (NMF) and successfully identified seven MPs along with their top-scoring genes (Fig. 5a). To further profile the MP1-7 obtained,

we first performed Gene Ontology to analyze the top 50 genes within each MP and indicate that MP1 and MP2 were related to nuclear division and DNA replication, MP4 was characterized by extracellular matrix organization, and MP6 was associated with hypoxia (Fig. 5b). Then, we analyzed their gene signature similarities with annotated MPs from previous tumor MPs, demonstrating the existence of these MPs

**Fig. 4 | Site-specific phenotypes of macrophages and functional reprogramming of *SELENOP*+ macrophages in the solid tumor microenvironment of metastatic HGSOC. a** The line plot showing the changes of relative fractions among macrophage clusters across the EAT, LAT and Met groups, $p = 2.2 \times 10^{-16}$, two-sided chi-squared test. Representative examples of ovarian tumor stained by multiplex immunohistochemistry, scale bar, 50 μm (**b**) and the quantification plots (**c**). For **c**, data are represented as means ± SD. Two-sided Kruskal−Wallis test, followed by Dunn's post hoc test with adjustment. $n = 20$ samples from 11 HGSOC patients, including $n = 5$ EAT, $n = 5$ LAT, $n = 5$ Met. Ome, and $n = 5$ Met. Per, biological replicates. **d** Developmental trajectory of M05, M06 and M09 inferred by Monocle 2 analysis, color-coded by cluster (upper) and state (lower). Each dot represents a single cell. **e** Pathway enrichment analysis of the differential genes of cells in different state. Curve plots showing expression changes of function genes related to interferon signaling (**f**, left), *SELENOP* expression (**f**, right), HIF-1 signaling pathway (**g**, left), and *SPP1* expression (**g**, right) along two cell fates. **h** Rank for regulons in *SELENOP*+ macrophages and *SPP1*+ macrophages based on regulon specificity score (RSS). **i** Heatmap showing transcription factors (TF) activity for *SELENOP*+ macrophages and *SPP1*+ macrophages. **j** Heatmap showing the RNA expression of TF along the pseudotime trajectory. **k** Western blot images (left) and quantification (right) of the levels of *SELENOP* proteins in THP-1 cells under control conditions or after treatment with 70 nM Se, 40 ng/mL IFNγ, or their combination. Se: $Na_2O_3Se$. **l** ELISA showing SELENOP level in the culture supernatants in (**k**). For **k** (right), **l**, data represent the mean ± SD. One-way ANOVA with Bonferroni post hoc test, $n = 3$, biological replicates. **m** Violin plots showing *SELENOP* expression of *SELENOP*+ macrophages across the EAT, LAT and Met groups, colored by mean expression. Box of violin plot represents median ± interquartile range, the whiskers extend up to the minimum and maximum values. $p$ values are calculated by two-sided Wilcoxon tests, adjusted by the Benjamini−Hochberg procedure. For **a**, **d**−**j**, **m**, $n = 17$ scRNA-seq cohort solid site samples, biological replicates. Source data are provided as a Source Data file.

in pan-cancer level (Supplementary Fig. 11a, b)[51,52]. Notably, MP3 and MP6 were enriched in cancer cells characterized by immune (Cancer.cell.3) and hypoxia signaling pathways (Cancer.cell.6) in a HGSOC cohort (syn52458609)[16] (Supplementary Fig. 11c). PROGENy pathway analysis also revealed consistent results, including enrichment of the JAK-STAT pathway in MP3, which is closely associated with interferon stimulation, and enrichment of the hypoxia pathway in MP6 (Supplementary Fig. 11d). Based on these findings, we designated the MPs we obtained as Cell Cycle-G2/M (MP1), Cell Cycle-G1/S (MP2), Interferon (MP3), EMT (MP4), Stress (MP5), Hypoxia (MP6), and Senescence (MP7). Among the identified metaprograms, MP2 showed a positive correlation with MP3, while MP4 was negatively correlated with MP3 and MP5 (Supplementary Fig. 11e). These results provide a molecular basis for understanding tumor heterogeneity in HGSOC.

To evaluate the relationship between these MPs and the progression of HGSOC metastasis, we first analyzed their association with the prognosis of HGSOC patients using TCGA data. Our findings confirmed that the signature score of MP6, MP4 and Senescence MP7 were associated with worse overall survival (OS) in OC (Fig. 5c), respectively. Further analysis revealed that, compared to the malignant epithelial cells from the EAT group, the signature scores of these three MPs were higher in those from the Met and LAT groups, particularly in the Met group (Fig. 5d). This finding was validated by independent bulk RNA sequencing we previously generated[15], though there was no statistically significant for MP7 (Fig. 5e). These suggested that the hypoxia-driven, EMT, and senescence state of malignant epithelial cells was associated with HGSOC metastases. We also noted that the signature score of MP6 presented strong positive correlation with the burden of copy number variation (CNV) (Fig. 5f). This finding supports the concept that hypoxia was correlated with genomic instability and tumor progression[53,54].

We constructed copy number alteration (CNA) phylogenetic trees for 8 patients with HGSOC and revealed specific clones associated with high Hypoxia (MP6) signature scores (Supplementary Fig. 11f and Supplementary Figs. 7−10). We also revealed that *FTL*, one of the MP6 signature genes, induced by hypoxia, was elevated in metastasis and predicted poor survival (Supplementary Fig. 12, Supplementary Data 8 and Supplementary Note 5).

**Hypoxia-driven malignant epithelial cells of HGSOC reprogramed *SELENOP*+ macrophages via VEGFA-EPHB2**

Tumor-host bi-directional interactions contribute to tumor adaptation and invasion within the microenvironment, leading to tumor progression, metastasis and recurrence[55–57]. Cellphone DB analysis revealed that ligand-receptor-mediated intercellular interactions between malignant epithelial cells and macrophages were more pronounced than those between malignant epithelial cells and other cell types (Supplementary Fig. 13a). In terms of macrophage subtypes,

hypoxia-driven malignant epithelial cells negatively correlated with *SELENOP*+ macrophage infiltration (Spearman $p = 0.034$, $R = -0.52$, 95% CI = [−0.823, −0.057], Fig. 6a and Supplementary Fig. 13b), while positively correlating with the proportion of *SPP1*+ macrophages (Spearman $p = 0.019$, $R = 0.57$, 95% CI = [0.051, 0.839], Fig. 6a and Supplementary Fig. 13b). Our spatial transcriptomics analysis revealed that *SPP1*+ macrophages, rather than *SELENOP*+ macrophages, were found in close proximity to malignant epithelial cells with MP6 signature (Fig. 6b, c for Stereoseq and Supplementary Fig. 13c, d for 10x Visium). mIHC further validated that *SPP1*+ macrophages rather than *SELENOP*+ macrophages contact with hypoxic tumor cells (Supplementary Fig. 13e). The above results suggest a phenotypic transition of *SELENOP*+ macrophages associated with hypoxia-driven tumor cells during HGSOC metastasis.

To probe the underlying mechanisms, we performed a comprehensive characterization of the ligand-receptor interactions between malignant epithelial cells and *SELENOP*+ macrophages. Our analysis identified the ligand-receptor pairs exhibiting stronger interaction strength between *SELENOP*+ macrophages and malignant epithelial cells with elevated MP6 signature scores, compared to those with lower scores. Subsequently, we examined the intersections between the ligands derived from malignant epithelial cells within the 7 ligand-receptor pairs and the top 50 genes in the Hypoxia (MP6) module, revealing two overlapping genes: vascular endothelial growth factor A (*VEGFA*) and macrophage migration inhibitory factor (*MIF*). In the interactions mediated by *VEGFA* or *MIF*, the VEGFA-EPHB2 interaction was notably more pronounced in peritoneal sites (Fig. 6d). Our single-cell RNA-seq data revealed a positive correlation between *VEGFA* expression and the MP6 signature score in the malignant epithelial cells (Spearman $p = 2.2 \times 10^{-16}$, $R = 0.44$, 95% CI = [0.420, 0.453], Supplementary Fig. 13f), along with a negative correlation between *VEGFA* expression in malignant epithelial cells and *SELENOP* expression in *SELENOP*+ macrophages (Spearman $p = 0.0014$, $R = -0.73$, 95% CI = [−0.921, −0.367], Supplementary Fig. 13f). Thus, MP6 malignant cells might suppress M06 macrophages through VEGFA.

Recent evidence suggests that the VEGFA-EPHB2 ligand-receptor pair mediates a synergistic interplay between VEGF and EPH signaling pathways, as shown in ovarian organoids where it facilitated follicular development[58]. EPH receptor B2 (EphB2), an important member of the Eph receptor family, have been implicated in diverse cell-cell communication events associated with cancer progression[59]. To validate the VEGFA-EPHB2 axis in modulating the crosstalk between *SELENOP*+ macrophages and hypoxia-driven malignant epithelial cells, we performed co-culture assay with CM from ovarian cancer cells (COV362, OVCAR3, CAOV3 and ID8). Hypoxia induced VEGFA upregulation in OVCAR3/CAOV3 (48 h) and COV362/ID8 (72 h) (Fig. 6e). CM from these cells suppressed SELENOP expression in THP-1 macrophages, an effect reversed by VEGFA neutralization (Fig. 6f). Parallel experiments using

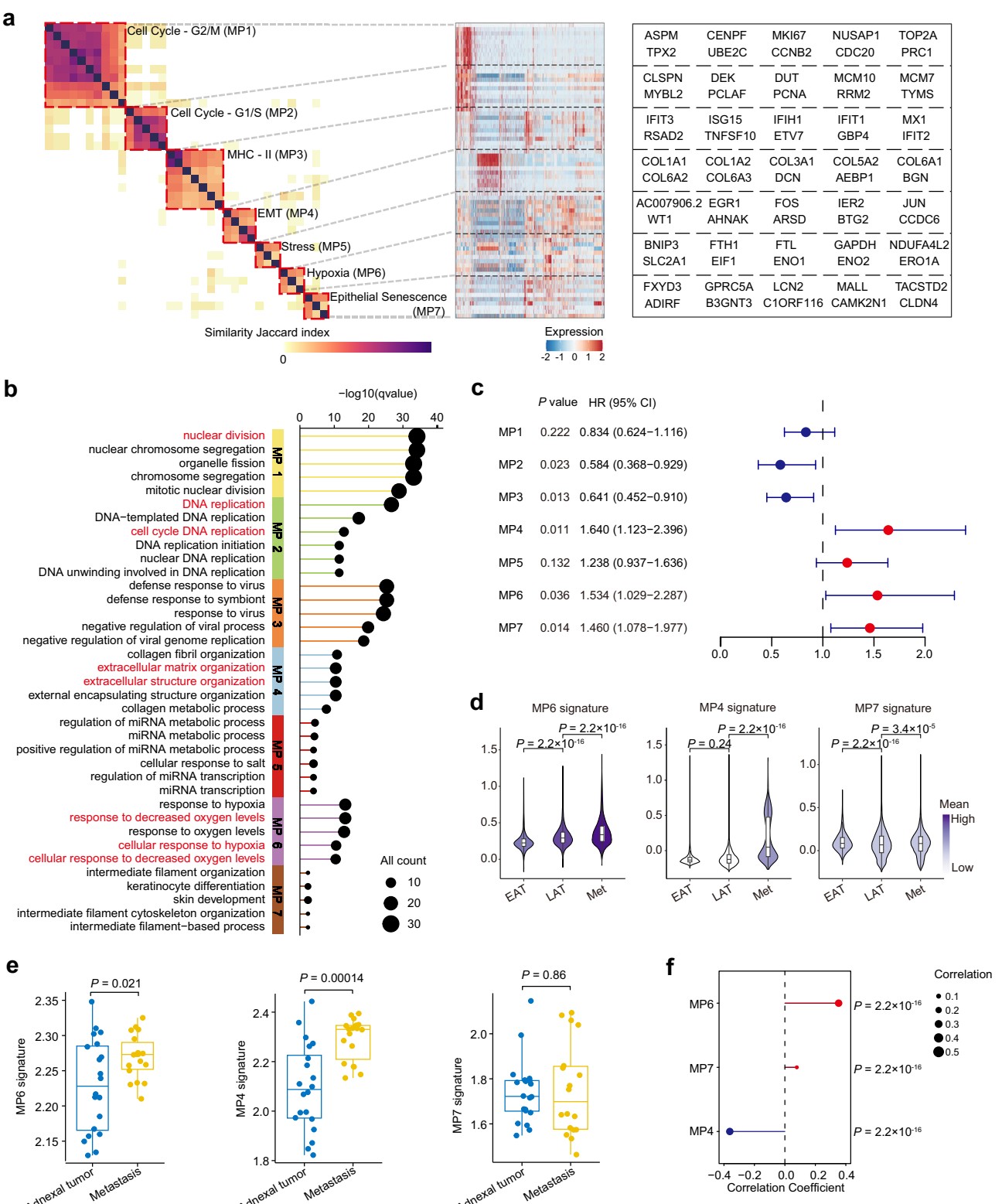

hypoxia-conditioned media from ID8 and BMDMs yielded consistent results (Fig. 6f). Either CRISPR-mediated *EPHB2* knockout in THP-1 cells or shRNA-mediated *Ephb2* knockdown in BMDMs (Supplementary Fig. 5c, d) abolished VEGFA-induced repression of SELENOP (Fig. 6g–i). Patient-derived cancer organoids (PDOs) represent effective tools for personalized medicine[60]. PDOs also confirmed the above findings: hypoxia increased VEGFA (Fig. 6j and Supplementary Data 7), and CM from hypoxic PDOs suppressed SELENOP in MDMs (Fig. 6k). Hypoxia

also reduced PDOs volume (Fig. 6l and Supplementary Data 7). Together, these data show that hypoxia-driven epithelial VEGFA represses macrophage SELENOP expression via EPHB2.

**Anti-VEGFA therapy decreased tumor burden in vivo**

To investigate the effect of VEGFA on ovarian tumor metastasis and the role of VEGFA on ovarian tumor microenvironment in vivo, we established orthotopic ovarian cancer mouse model by injecting

**Fig. 5 | Malignant epithelial cells transcriptional heterogeneity and genomic alterations. a** Heatmap showing similarities between NMF programs quantified by Jaccard index over the programs' signature genes. Tumor MPs identified from consensus NMF programs (marked by red dotted lines) are numbered and labeled (left). List of each MPs' top 10 genes (right). **b** Enriched pathways of each MPs. The signatures were obtained from the GO-BP database. **c** The forest plot for overall survival (TCGA-OC: $n = 343$) according to signature scores of distinct tumor MPs. Horizontal lines indicate 95% CI. HR hazard ratio, 95% CI 95% confidence interval. $p$ values are calculated by the two-sided Wald test in the Cox proportional hazards regression model with Benjamini–Hochberg adjustment. Red dots represent HR > 1, blue dots represent HR < 1. **d** Violin plots showing MP6 (left), MP4 (middle) and MP7 (right) signature scores in tumor cells in different tumor sites. $p$ values are calculated by the two-sided Kruskal–Wallis test with Bonferroni post hoc test. Colored by mean expression. **e** Boxplot showing MP6 (left), MP4 (middle) and MP7

(right) gene signature scores in adnexal tumors ($n = 20$) and metastasis ($n = 18$) in the bulk RNA-seq cohort from Yang et al. (ref. 15), biological replicates. For **d**, **e**, box represents median ± interquartile range, the whiskers extend up to the minimum and maximum values, for **e**, points represent individual samples. For **e**, $p$ values are calculated by the two-sided Student's $t$ test. **f** Correlation between MPs signature score and CNV frequency in malignant epithelial cells. The size of the circles represents the magnitude of the correlation coefficient. Red represents the correlation coefficient >0, blue represents the correlation coefficient <0; $p$ values are calculated by the two-sided Spearman correlation test with Benjamini–Hochberg adjustment. For **a**, **b**, **f**, data were summarized from all $n = 21$ scRNA-seq cohort samples, including $n = 17$ solid site samples of HGSOC and $n = 4$ Ascites (from OCL3, OCL5, OCL7 and OCL8), biological replicates. For **d**, all $n = 17$ scRNA-seq cohort solid site samples, biological replicates. MP metaprogram. Source data are provided as a Source Data file.

intrabursally (i.b.) luciferase-transfected ID8 cells into immunocompetent C57BL/6J (B6) mice (Fig. 7a–c). We found that VEGFA inhibition significantly reduced tumor burden at day 49, evidenced by decreased bioluminescence imaging (BLI) signal intensity (Fig. 7b, c), fewer peritoneal metastases (Fig. 7d, e), and smaller primary tumors (Fig. 7f, g). Additionally, VEGFA blockade increased the proportion of F4/80+CD11b+SELENOP+ macrophages in both primary tumors and peritoneal metastases (Fig. 7h), elevated CD3+CD8+GZMB+ T cell proportions in the primary tumors (Fig. 7i) and CD3+CD8+PD-1+ cell proportions in the primary tumors (Fig. 7j). These suggested that anti-angiogenesis therapy, such as bevacizumab, could prolonged survival not only by killing tumors[61,62], but also by increasing SELENOP+ macrophages, thus enhanced CD8+ T cell activities. mIHC further validated the spatial association between SELENOP+ macrophages and CD8+ GZMH+ T cells within the tumor microenvironment of the orthotopic ovarian cancer model (Fig. 7k). In addition, VEGFA inhibition did not significantly alter SELENOP concentrations in the serum of mice, thereby excluding a substantial systemic effect of TME-derived VEGFA on the circulating SELENOP levels (Fig. 7l).

To further evaluate the capacity of SELENOP+ macrophages to directly prime CD8+ T cells and to demonstrate the requirement of macrophage-derived SELENOP in modulating VEGFA's effect in vivo, we performed adoptive cellular transfer experiments (Supplementary Fig. 14a). Then we found that, compared with the BMDMs group, the proportion of CD45.2+CD3+CD8+GZMB+ T cells decreased, accompanied by reduced frequencies of CD45.1+F4/80+CD11b+SELENOP+ macrophages in the primary tumors of the BMDMs-KD1-Selenop group. Meanwhile, VEGFA inhibition led to a concomitant increase in CD45.2+CD3+CD8+GZMB+ T cells and CD45.1+F4/80+CD11b+SELENOP+ macrophages in the primary tumors of mice injected with BMDMs-KD1-Selenop, but had no significant effect in mice injected with BMDMs (Supplementary Fig. 14b, c). These findings support the role of SELENOP+ macrophages in priming CD8+ T cell cytotoxicity within the HGSOC tumor microenvironment in vivo, and further emphasize a VEGFA-mediated regulatory mechanism that appears to be more pronounced in macrophages with low SELENOP expression.

### Reduced interaction potential between *SELENOP*+ macrophages and *GZMH*+ precursor exhausted CD8+ T cells following neoadjuvant chemotherapy (NACT)

Chemotherapy is the classical first-line treatment regimen for HGSOC. However, the specific alterations it induces in TME and their subsequent impact on therapeutic efficacy remain largely unexplored. To investigate the impact of chemotherapy on macrophage populations in HGSOC, we analyzed an external scRNA-seq dataset (GSE266577)[63]. Chemotherapy reduced overall macrophage abundance, shifting them from anti- to pro-inflammatory states (Supplementary Fig. 15a–c). Subtype analysis showed stable proportions of *SELENOP*+ and *SPP1*+ macrophages (Supplementary Fig. 15d). However, *SELENOP*+ macrophages displayed reduced SELENOP

expression and antigen presentation, while *SPP1*+ macrophages showed diminished matrix remodeling (Supplementary Fig. 15e–g). Intercellular communication analysis revealed that *SELENOP*+ macrophages interacted strongly with T04 CD8+ T cells before NACT, but this interaction weakened post-NACT, shifting toward *SPP1*+ macrophages (Supplementary Fig. 15h). Spatial profiling confirmed that the coexistence scores of *SELENOP*+ macrophages and CD8+ T cells remained unchanged in the regions of interest (ROIs) with the presence of CD8+ T cells and IBA1+ macrophages in the GeoMX data, regardless of tumor or stromal areas of interest (AOIs) (Supplementary Fig. 15i). These results suggest a disruption of the immunoregulatory axis between *SELENOP*+ macrophages and *GZMH*+ CD8+ T cells following NACT, which may be attributable, at least in part, to a reduction in their functional interaction potential rather than changes in spatial proximity.

Clinically, *SELENOP*+ macrophages were enriched in non-relapsed patients, whereas *SPP1*+ macrophages predominated in relapsed cases in our scRNA-seq cohort (Supplementary Fig. 15j). In the pre-NACT samples from GSE266577, long-platinum-free interval (PFI) patients showed enrichment of *SELENOP*+ macrophages with higher SELENOP expression, while short-PFI patients exhibited *SPP1*+ macrophages with increased matrix remodeling (Supplementary Fig. 15k–m). The finding related to *SELENOP*+ macrophages were validated by mIHC in an independent cohort (Supplementary Fig. 15n and Supplementary Data 7).

### Distinct immune cell subpopulations enriched in solid sites across different *BRCA* statuses

*BRCA* mutation testing in first-line OC treatment has been recommended as it provides an opportunity to optimize the use of PARPi in HGSOC patients[64]. Recent studies have highlighted the association between the *BRCA* mutation and the suppressive TME[65,66]. Therefore, a deeper understanding of the TME in relation to different *BRCA* status is crucial for improving the therapeutic response of HGSOC patients to PARPi. We identified BRCA-mutated patients according to clinical assessment or WES (Supplementary Data 9) and their TME manifestations (Supplementary Fig. 16), suggesting a more active and coordinated antitumor immune response in *BRCA*-mutated HGSOC tumors, whereas *BRCA* wild-type HGSOC tumors are associated with a more suppressive TME (Supplementary Note 6). This distinction is crucial for developing effective therapeutic strategies for HGSOC.

## Discussion

Understanding the pivotal role of the TME in the initiation and progression of HGSOC is crucial for the discovery of novel therapeutics for this lethal disease[67]. However, our understanding of tumor-TME interactions and the underlying mechanisms promoting HGSOC metastasis remains limited. In this study, we systematically dissected the characterization and dynamics of key immune and malignant cells within the HGSOC tumor ecosystem, revealing that hypoxia-driven

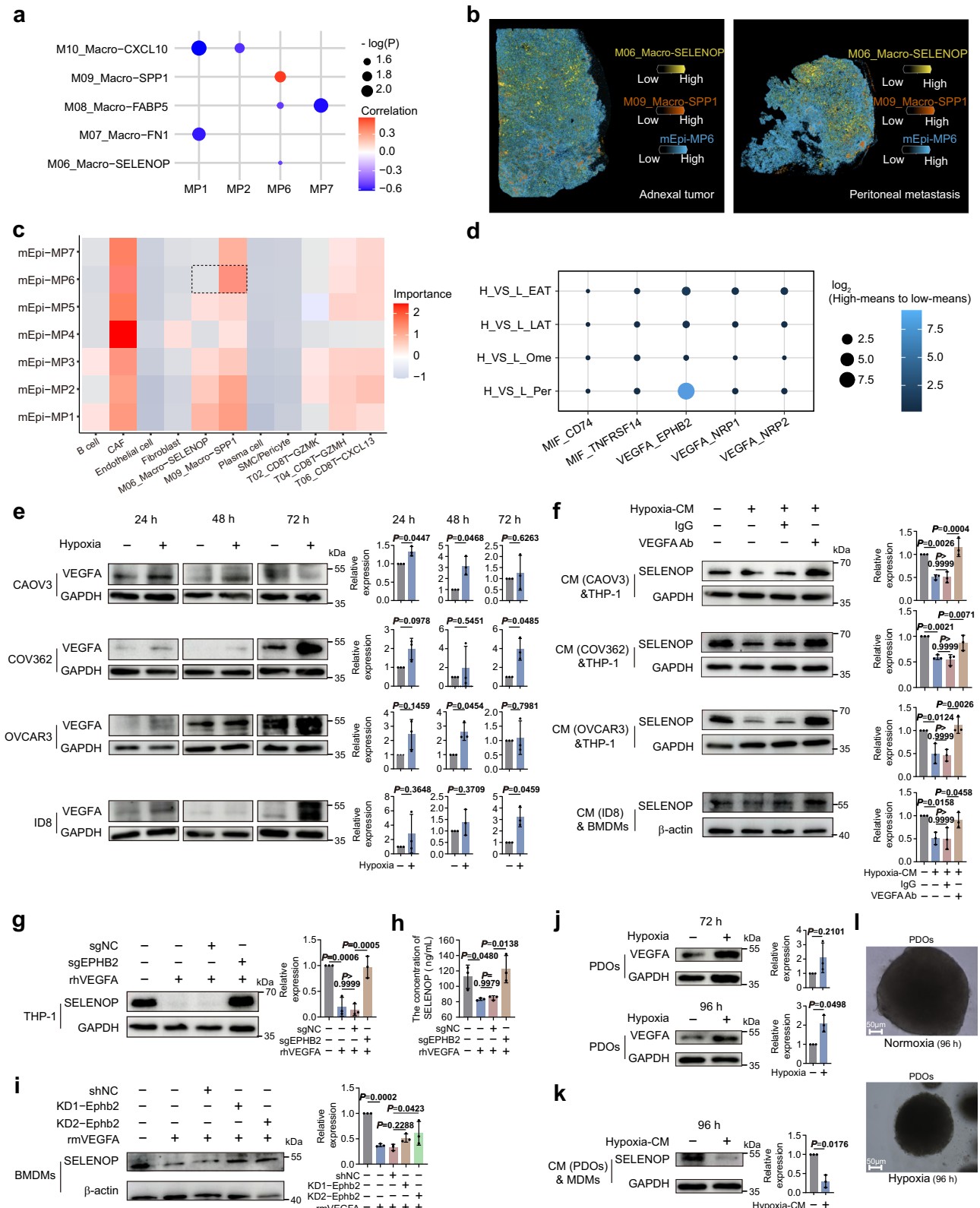

malignant epithelial cells could reprogram *SELENOP*⁺ macrophages, thereby impairing the antitumor innate-adaptive immune response, which is closely associated with HGSOC metastasis (Fig. 8). We expanded our understanding of the molecular mechanisms of anti-VEGFA therapy in ovarian cancer within the context of the TME, thereby indicating the potential for combining anti-VEGFA therapy with immunotherapy.

We identified a *SELENOP*⁺ macrophage subcluster with potential antigen processing and presentation signature that is associated with favorable HGSOC prognosis. A similar macrophage subset has been observed in lung cancer, where it plays an antitumor role[68]. However, the functional implication of *SELENOP*⁺ macrophages in HGSOC development remains underexplored. Our work provided evidence that *SELENOP*⁺ macrophages possess the capacity to activate precursor

**Fig. 6 | Hypoxia-driven malignant epithelial cells of HGSOC reprogram *SELENOP*+ macrophages via VEGFA-EPHB2. a** Dot plot of the Spearman correlation coefficients between distinct tumor MPs and the proportion of macrophage clusters (only showing *p* value < 0.05). *p* value are calculated by the two-sided Spearman correlation test with Benjamini–Hochberg adjustment. **b** Representative spatial co-localizations of *SPP1*+ macrophage with malignant epithelial cells with MP6 signature in our spatial transcriptomics cohort. **c** Heatmap showing the importances among all cell types signature scores in spots of spatial transcriptomics data. mEpi malignant epithelial cells. **d** Bubble heat map showing the mean interaction strength for selected ligand–receptor pairs between malignant epithelial cells and *SELENOP*+ macrophages (malignant epithelial cells providing ligands) across groups. H_VS_L: the discrepancy in the interaction strength between malignant cells with high or low MP6 signature scores and *SELENOP*+ macrophages. **e** Western blot images and quantifications of VEGFA in cell lines under normoxic or hypoxic conditions for 24, 48, and 72 h. **f** SELENOP in THP-1 or BMDMs after cocultured with conditioned medium (CM) from cell lines under normoxic or hypoxic conditions for 48 h, with or without VEGFA antibody (VEGFA Ab). Western blot images and quantifications of SELENOP after knocking down *EPHB2* by CRISPR-

Cas9 in THP-1 (**g**) or by shRNA in BMDMs (**i**) followed by rhVEGAF or rmVEGFA treatment. **h** ELISA showing SELENOP level in the culture supernatants in (**g**). For **g**–**i**, sgNC, empty vector for negative control of sgRNA; sgEPHB2, EPHB2 deletion mediated by CRISPR/Cas9; rh(m)VEGAF, recombinant human (mouse) VEGFA protein. shNC, transfected with negative control shRNA; KD1-Ephb2, *Ephb2* knockdown by sh*Ephb2*-1; KD2-Ephb2, *Ephb*2 knockdown by sh*Ephb*2-2. Western blot images and quantifications of VEGFA in PDOs under normoxic or hypoxic conditions for 72 h and 96 h (**j**). SELENOP in MDMs after cultured with CM from PDOs under normoxic or hypoxic conditions (96 h) (**k**). MDMs human monocyte-derived macrophages. **l** Representative images of PDOs under normoxic (upper) or hypoxic conditions (lower). For **f**, **k**, & denotes coculture. For **j**–**l**, PDOs patient-derived organoid, MDMs human monocyte-derived macrophages. For **a**, **d**, *n* = 17 scRNA-seq cohort solid site samples, biological replicates. For **b**, **c**, *n* = 24 spatial RNA cohort samples, biological replicates. For **e**–**k**, data represent the mean ± SD. For **e**, **j**, **k**, two-sided unpaired Student's *t* test, *n* = 3, biological replicates (**e**) or technical replicates (**j**, **k**). For **f**–**i**, one-way ANOVA with Bonferroni post hoc test, *n* = 3, biological replicates. Source data are provided as a Source Data file.

---

exhausted CD8+ T cells via SELENOP in the solid TME of HGSOC, thereby promoting a synergistic antitumor immune response. We further observed reduced cytotoxicity of precursor exhausted CD8+ T cells, along with decreased SELENOP expression in *SELENOP*+ macrophages within metastatic HGSOC. This immune impairment, driven by reprogrammed *SELENOP*+ macrophages, is intricately associated with HGSOC metastasis. Previous studies have established that SELENOP is a secreted glycoprotein primarily produced by hepatocytes, functioning as a key selenium transporter responsible for delivering selenium to peripheral tissues for the biosynthesis of essential selenoproteins, such as members of the GPX family, which are closely associated with redox homeostasis[69]. As redox balance is critical for cell survival and function, SELENOP-mediated selenium transport plays an important physiological role[70]. Ayako Mizuno et al. demonstrated that human T lymphoma Jurkat cells exhibit a high capacity to utilize selenium via SELENOP for GPX1 synthesis[71]. Our study expands the current understanding of SELENOP by identifying it as macrophage-derived and uncovering its potential immunoregulatory role within the tumor microenvironment. Additionally, it is worth noting that systemic selenium metabolism hierarchically regulates the expression of selenoproteins, including SELENOP[69]. Our data also demonstrate that SELENOP expression in macrophages is responsive to selenium availability, indicating that therapeutic strategies aimed at modulating macrophage-derived SELENOP to enhance CD8+ T cell-mediated antitumor immunity in HGSOC should also take selenium status into careful consideration.

Tumor cells and macrophages engage in dynamic, reciprocal interactions that reprogram macrophages to support tumor progression and immune evasion[72–74]. Our data revealed a potential close interaction between hypoxia-driven malignant epithelial cells and *SELENOP*+ macrophages. Mechanistically, our study demonstrates that hypoxia-driven malignant epithelial cells might reprogram *SELENOP*+ macrophages via VEGFA-EPHB2, subsequently diminishing the antitumor cytotoxicity of CD8+ T cells. In summary, the tumor-macrophages-CD8+ T cells interactions elucidated here reflected the impaired antitumor responses remodeled by hypoxia-driven cancer cells within the TME, which may contribute to the progression of HGSOC metastasis. Previous researches have primarily focused on identifying therapeutic targets to reverse the pro-tumor phenotype of TAMs[75–78]. In contrast, our study emphasizes the importance of blocking the reprogramming of anti-tumor TAMs to the pro-tumor phenotype, offering a different perspective for immune therapy research in HGSOC. Furthermore, while targeting VEGFA has been shown to treat HGSOC by inhibiting angiogenesis[79], our findings suggest that targeting either VEGFA or EPHB2 could remodel the immune

microenvironment and inhibit HGSOC metastasis, highlighting their potential as promising candidates for combination immune therapies. While earlier work has demonstrated that hypoxia directly represses selenoprotein biosynthesis in hepatocytes[80], our findings extend this concept by uncovering a tumor cell-dependent, paracrine mechanism.

Taken together, although previous related studies have also identified alterations in specific immune cell subsets related to ovarian cancer progression[46,81], our work goes a step further by simultaneously examining both tumor and immune cell compartments within the ovarian cancer microenvironment. Notably, we uncovered a critical cross-talk among malignant cells, macrophages, and T cells that is associated with the progression of HGSOC. Such mechanistic insights have not been systematically explored in prior studies.

From a clinical perspective, an important first consideration is the relationship between the TME and chemotherapy response or resistance, which is critical for developing strategies to overcome drug resistance and improve outcomes in ovarian cancer. Previous evidence suggests that chemotherapy may induce M1-like inflammatory features in macrophages, which can promote CD8+ T cell exhaustion and contribute to reduced PFI[63]. Notably, recent findings from the TRUST trial demonstrated that stage III non-frail ovarian cancer patients undergoing primary cytoreductive surgery experienced significantly improved progression-free survival (PFS) compared to those receiving neoadjuvant chemotherapy followed by interval cytoreductive surgery. In our study, neoadjuvant chemotherapy reduced the interaction potential between *SELENOP*+ macrophages and *GZMH*+ precursor exhausted CD8+ T cells, which could partially explain the TRUST trial result. Notably, patients with longer PFS exhibited higher pre-treatment abundance of *SELENOP*+ macrophages. Thus, baseline levels of *SELENOP*+ macrophages may serve as a predictive biomarker for chemotherapy treatment response.

Despite strength of our multi-region and multi-temporal sample collections and multi-omics dissections, the findings of this study may be limited by the relatively small sample size. Incorporating a broader cohort of HGSOC tumor samples may yield the identification of previously unrecognized malignant cell states and immune cell subsets, thereby providing a more nuanced and comprehensive understanding of the complex interplay between ovarian cancer cells and the host immune response. Moreover, the hypothesized cancer cell–macrophage–CD8+ T cell interaction model necessitates rigorous validation through both preclinical and clinical investigations. Additionally, further in-depth studies are essential to delineate the specific molecular mechanisms governing the reprogramming of SELENOP+ macrophages, which could unveil promising therapeutic targets within the TME.

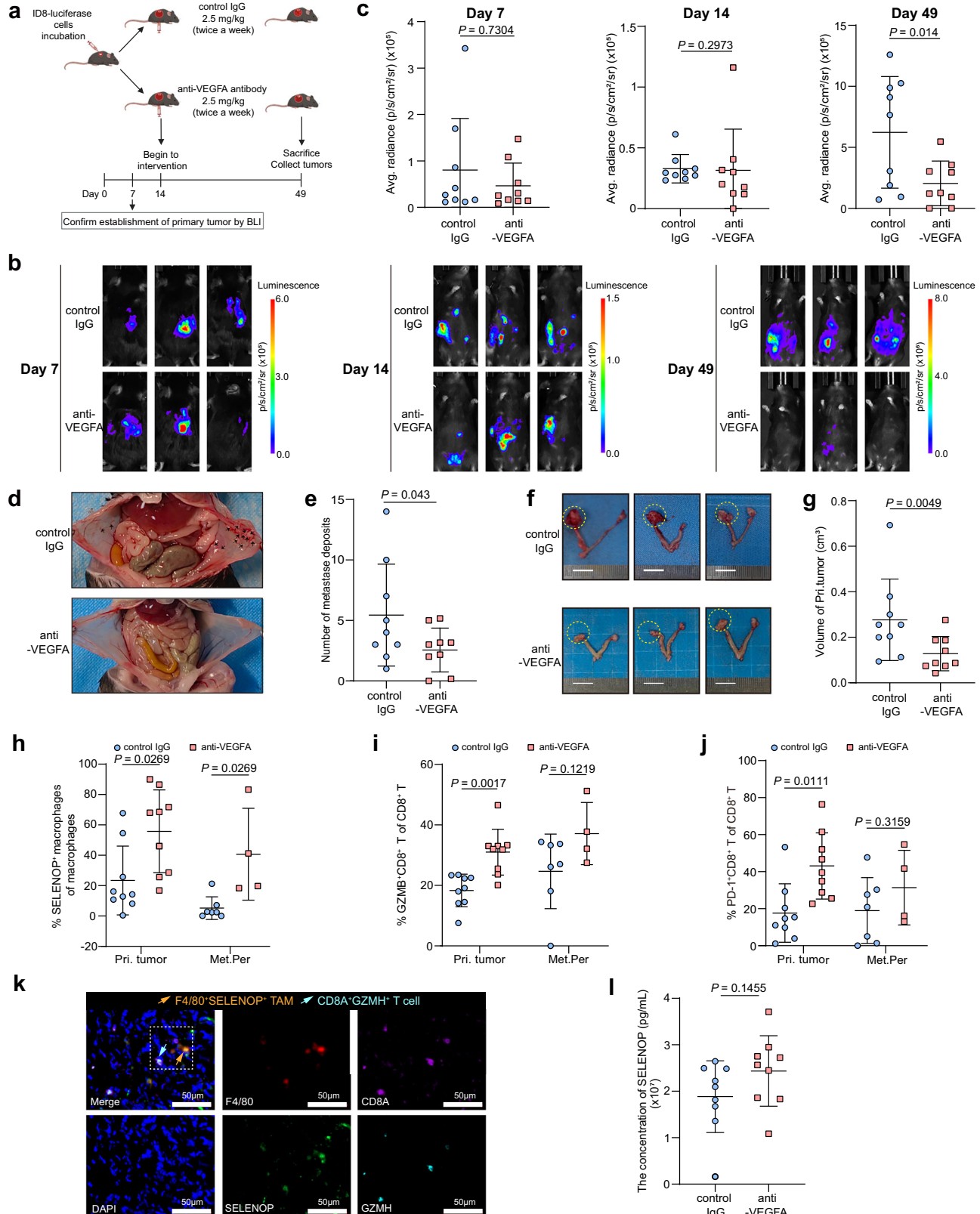

## Methods

This study was performed following the ethical guidelines of the Declaration of Helsinki and was approved by the Research Ethics Committee of Shengjing Hospital of China Medical University, the Research Ethics Committee of Tongji Hospital, Tongji Medical College, Huazhong University of Science and Technology, and the Research Ethics Committee of the First Hospital of China Medical University. Written informed consent was obtained from all patients involved in this study for the use of their tissue samples and clinical information and participant compensation was not implemented in this study. Sex was not considered a biological variable in the study, as ovarian cancer is a gender specific disease. Only female patients were included in our

**Fig. 7 | Exploration of upregulated antitumor response within ovarian tumor microenvironment by the administration of anti-VEGFA antibody in vivo.**
**a** Schematic diagram of the protocol for in vivo experiments. C57BL/6J (B6) mice inoculated with ID8-luciferase cells are treated with 2.5 mg/kg IgG (control mouse; $n = 9$) or 2.5 mg/kg anti-VEGFA antibody (anti-VEGFA treatment mouse; $n = 9$). Created in BioRender. Song, X. (2025) https://BioRender.com/1zggfuh. Representative images (**b**) and quantifications (**c**) of C57BL/6J (B6) mice detected at 7 d (left), 14 d (middle) and 49 d (right) by bioluminescence imaging after inoculation with ID8-luciferase cells. 49 d after inoculation with ID8-luciferase cells, mice in both groups were euthanized ($n = 9$ mice per group). Representative images showing characteristic (**d**) and quantifications (**e**) of peritoneal metastases (black arrows) in mice treated with control IgG (upper), and the absence of peritoneal metastases in mice treated with anti-VEGFA antibody (lower). Representative images (**f**) and quantification of tumor volumes (**g**) of primary tumors of mice treated

with control IgG or anti-VEGFA antibody. Bars, 10 mm. Fresh primary tumor and peritoneal metastasis tissues were digested and stained for flow cytometry analysis. Shown is the proportion of F4/80⁺CD11b⁺SELENOP⁺ macrophages (**h**), CD3⁺CD8⁺GZMB⁺ T cells (**i**), and CD3⁺CD8⁺PD-1⁺ T cells (**j**) across different sites of each group. Pri. Tumor, primary tumor of orthotopic ovarian cancer mouse model. Met.Per, peritoneal metastasis of orthotopic ovarian cancer mouse model.
**k** Representative immunofluorescence staining showing co-localization of F4/80 (red), SELENOP (green), CD8A (magenta), GZMH (cyan) and DAPI (blue) in C57BL/6 J (B6) mice samples. **l** Blood serum SELENOP concentrations of mice treated with IgG or anti-VEGFA antibody. For **c–j**, **l**, $n = 9$ for each group, biological replicates, control IgG, control mouse group; anti-VEGFA, anti-VEGFA treatment mouse group. For **c**, **e**, **g**, **h–j**, **l**, data represent the mean ± SD, points represent individual samples. $p$ values: two-sided unpaired Student's $t$ test for two groups, multiple $t$-test for multiple groups. ns not significant. Source data are provided as a Source Data file.

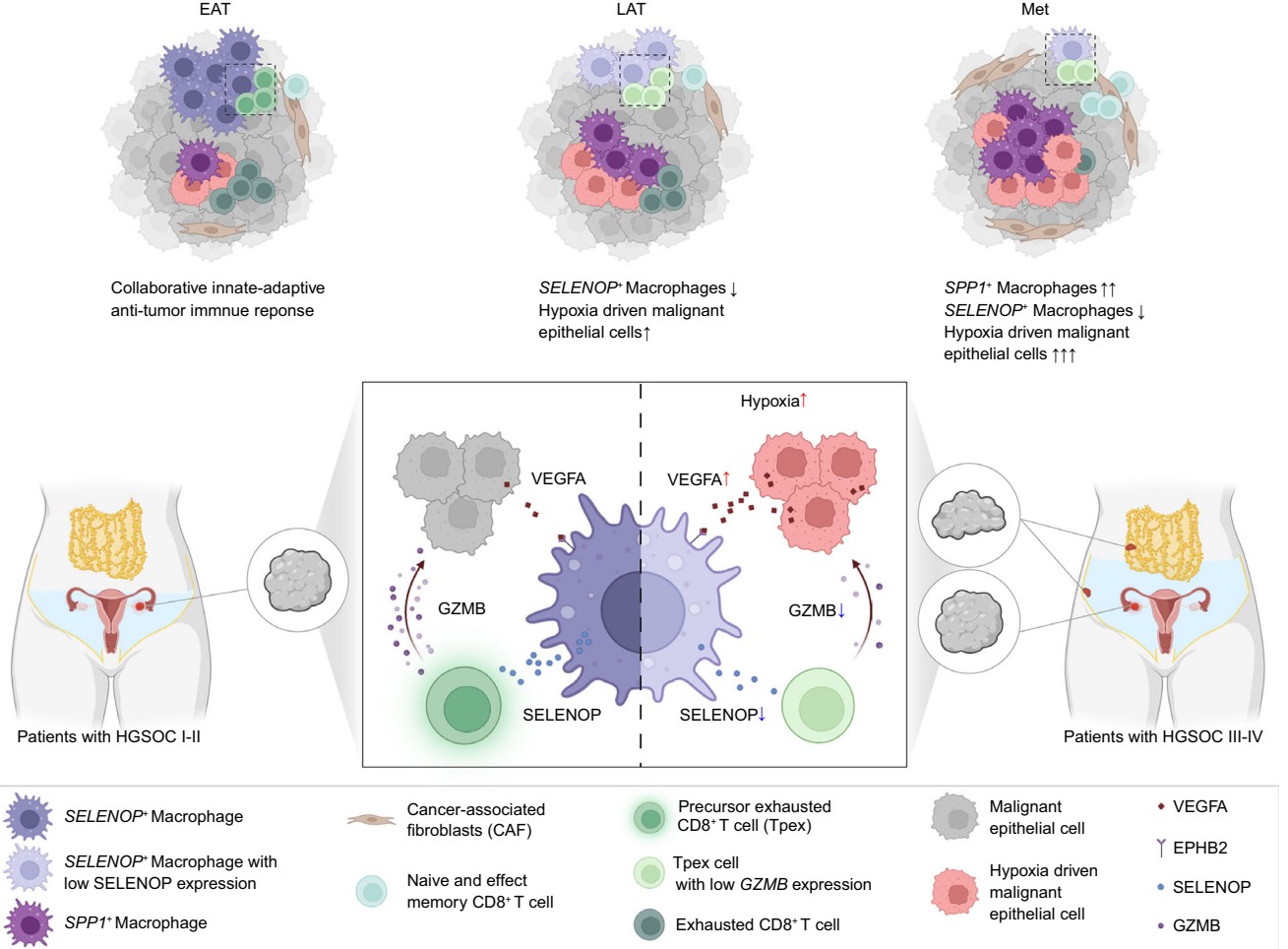

**Fig. 8 | Graphical summary of spatiotemporal heterogeneity in TME and model for the role of the malignant cells-macrophages-CD8⁺ T cells axis in immunosuppressive TME formation associated with HGSOC metastasis.** Schematic representation of the spatiotemporal heterogeneity of TME in cell type infiltration in the solid tumor sites of HGSOC, profiled by malignant epithelial cells, macrophages, and CD8⁺ T cells. Sketch map showing the proposed interaction model for

the role of the malignant cells-macrophages-CD8⁺ T cells axis in the immunosuppressive TME formation. Under hypoxic conditions, malignant epithelial cells derived VEGFA amplification, leading to macrophages reprogramming, especially the decreased expression of SELENOP, followed by impaired cytotoxic function of precursor exhausted CD8⁺ T cells, associated with HGSOC metastasis. Created in BioRender. Song, X. (2025) https://BioRender.com/zm3gt6g.

study. Patients' self-reported gender were used for the analysis. All animal studies were reviewed and approved by the Ethics Committee of the Laboratory Animal Department of China Medical University. Measures were taken to minimize animal distress. According to institutional policies on tumor production, the maximum allowable tumor diameter in any direction is 2.0 cm, and this limit was not exceeded in any of our experiments.

## Patient samples and ethics statement

The scRNA-seq cohort in our study included 34 fresh samples from 17 female patients with pathologically confirmed high-grade serous ovarian cancer (HGSOC) or uterine fibroids who underwent surgery at two medical centers: Shengjing Hospital of China Medical University and the First Hospital of China Medical University. The disease stages of the patients with HGSOC were classified according to the 2018 FIGO staging

system[82]. Specifically, the sample collection included three normal ovarian tissue samples from postmenopausal women (Nor.Ovr group) and six peritoneal lavage fluid samples (PLF.UF group) from nine patients diagnosed with uterine fibroids; two adnexal tumor lesion samples (EAT group) and two matched peritoneal lavage fluid samples (PLF.EOC group) from two patients with HGSOC at FIGO stage I–II; six adnexal tumor lesion samples (LAT group), six matched ascites samples (Ascites group) and nine matched metastases (Met group) including four omental metastatic lesion samples (Met.Ome group) and five peritoneal metastatic lesion samples (Met.Per group) from six patients with HGSOC at FIGO stage III–IV. Our scRNA-seq cohort included seven patients with HRD-positive HGSOC, comprising four with *BRCA1/2*-mutated HGSOC. The median age of the patients was 54 years. The spatial transcriptomics (ST) cohort in our study included 24 fresh samples from 6 female patients (median age: 51.5 years) diagnosed as HGSOC underwent surgeries at Tongji Hospital, Tongji Medical College, Huazhong University of Science and Technology, Wuhan, China. Fresh tumor samples were collected during surgeries and prepared for ST. None of these patients received chemotherapy, radiotherapy, or any other antitumor therapy before surgery. Clinical characteristics of the patients are summarized in Supplementary Data 1 and 2.

## Sample collection and single-cell suspension processing
Fresh samples including adnexal tumor, omental metastatic lesion, peritoneal metastatic lesion, ascites, peritoneal lavage fluid and normal ovarian tissue were obtained from the patients during surgery. Tissue samples were placed in MACS Tissue Storage Solution (Cat#130-100-008, Miltenyi Biotec). Briefly, fresh tissue samples were first washed with Phosphate-Buffered Saline (PBS), cut into small pieces (approximately 1 mm³) and dissociated in Human Tumor Dissociation kit (Cat#130-095-929, Miltenyi Biotec) in RPMI 1640 (Cat#C11875500BT, Gibco) for 40 min at 37 °C. Tissue samples after digestion or fluid samples were subsequently filtered using 70 μm cell strainers (Cat#352350, Corning) and centrifuged at 300 × g for 5 min. The supernatant was removed and the pelleted cells were suspended in red blood cell lysis buffer (Cat#R1010, Solarbio). Lysis was quenched by adding PBS and the cell suspension was centrifuged at 300 × g for 5 min. Then the remaining pellet was washed twice with PBS and resuspended in PBS containing 2% fetal bovine serum (FBS, Cat#FB65015, CLARK Bioscience). Dissociated single cells were then stained with acridine orange/propidium iodide (AO/PI) (Cat#CS2-0106-5ML, Nexcelom Bioscience) for viability assessment. When the viability of single cell suspensions was higher than 85%, the cell concentration was set according to 10x Genomics protocols (800–1200 cells/μL).

## Single-cell RNA library preparation and sequencing
Single-cell RNA library preparation and sequencing experiments were conducted according to the manufacturer's instructions of the 10x Genomics Chromium single cell 3′ platform (10x Genomics). In short, 10x Chromium Controller (Mode l# GCG-SR-1, 10x Genomics), Chromium Next GEM Single Cell 3′ Reagent Kit v3.1 (Cat# PN-1000123, 10x Genomics) and Chromium Next GEM Chip G Single Cell Kit (Cat#PN-1000120, 10x Genomics) were used to generate single-cell nanoliter-scale Gel Beads-in-emulsion (GEM) and barcoded RNAs. Reverse transcription was performed using a Touch Thermal Cycler (Model#9902, Applied Biosystems), the GEMs were programmed at 53 °C for 45 min, followed 85 °C for 5 min, and hold at 4 °C. The cDNA was generated and amplified, and the quality was assessed by the Agilent 4200 TapeStation system (Agilent Technologies). Libraries were sequenced on an Illumina Novaseq 6000 sequencer using a pair-end 150 bp (PE150) reading strategy (CapitalBio Technology, China).

## Single-cell RNA-seq data processing and quality control
Cell Ranger software (version 3.1.0) was used to process single-cell sequencing data and align it to the GRCh38 human reference genome.

For subsequent quality control and analysis, we utilized the filtered feature barcode expression matrix generated by Cell Ranger. Low-quality cells and genes were removed based on several criteria: cells with fewer than 200 genes, fewer than 800 UMI counts, top 1% UMI count, over 20% mitochondrial gene content, and genes present in fewer than 3 cells. The "DoubletFinder" R package (version 2.0.1) identified and removed potential doublets. Post quality control, the cell count matrix was normalized using the NormalizeData function of the Seurat package (version 4.3.0).

## Cell type identification and characterization
For each individual sample, the normalized expression matrix was subjected to dimensionality reduction and unsupervised clustering using the Seurat workflow. First, for the purpose of downstream analysis, a selection of 2000 highly variable genes was made utilizing the "FindVariableGenes" function. Next, the matrix was scaled using the "ScaleData" function. Each individual sample matrices were merged by sample and subsequently renormalized and scaled. Subsequently, principal component analysis (PCA) was conducted for dimensionality reduction on highly-variable genes using the RunPCA function from the Seurat R package, specifying npcs = 50 as the parameter. For primary analyses, unsupervised cell clusters were acquired by graph-based clustering approach using the "FindNeighbors" and "FindClusters" functions. For visualization of graph-based cell clustering, Uniform Manifold Approximation and Projection (UMAP) was applied with the "RunUMAP" function. The main cell populations were identified by first-run clustering and subsequently annotated according to the expression of canonical markers in previous literature. The cells expressing more than one major cell marker were considered doublets and removed from each cluster individually. Marker genes of each subcluster were identified by the "FindAllMarkers" function. Genes with adjusted $p$ value < 0.05 by Wilcoxon rank-sum test were defined as cluster-specific marker genes.

## Batch correction and integration
After the first round of dimensionality reduction and unsupervised cell clustering, the major cell types identified across samples were divided into ten supersets, with myeloid cells and T/NK cells being re-clustered for further analysis. For myeloid cells and T/NK cells superset, the R package harmony (version 1.0) was used for batch correction to account for sample-specific effects.

## PCA plot for pseudo-bulk RNA-seq
Pseudo-bulk RNA-seq analysis was executed by isolating the raw counts for each specimen using the "Aggregate Expression" function within the "Seurat" R package. The raw counts were consolidated for each specimen, effectively producing a collective gene expression profile from the single-cell data. Principal components for pseudo-bulk RNA-seq analysis were examined and visualized using "pcaExplorer" (version 2.14.2), and statistical analyses were performed using PERMANOVA.

## $R_{o/e}$ analysis
To assess tissue preference for each cluster, we calculated the ratio of observed to expected cell numbers ($R_{o/e}$) across various tissues, according to the following formula (Eq. (1)) described by Zhang et al.[23] and Cheng et al.[34]:

$$R_{o/e} = \frac{Observed}{Expected} \tag{1}$$

Expected cell counts for each cluster-tissue pair were determined using the chi-square test. A cluster was deemed enriched in a tissue if its $R_{o/e}$ ratio exceeded 1.

## Definition of gene signatures and cell function scores

The CD4[+] and CD8[+] T cells signature lists were obtained from previous studies[18,28,83–85]. The macrophages signature lists were acquired from previous publications by Huang et al.[86] and Zheng et al.[87]. Antigen processing and presentation signature, as well as ECM-receptor interaction pathways, were sourced from KEGG. For signature quantification, we utilized "UCell" (version 2.5.0) to assess the signature/pathway activity of each subset. The signature genes were showed in Supplementary Data 6.

## TCR analysis

The raw data of TCR sequences was obtained at GSA-Human under accession code HRA002184, following an application through the corresponding author, and processed using Cell Ranger (version 7.1.0). Only cells with both single-cell RNA sequencing and single-cell TCR sequencing data were retained, and cells containing only a single α or β chain were excluded from downstream analysis. We then presented two STARTRAC indices to analyze different aspects of T cells based on paired single-cell transcriptomes and TCR sequences using STARTRAC (version 0.1.0)[23]. STARTRAC-expa and STARTRAC-tran were used to measure the degree of clonal expansion and state transition of T cell clusters upon TCR tracking, respectively, while STARTRAC-migr was not used. Additionally, CD8[+]$GNLY$[+] NK-like cells (T05), CD8[+]$SLC4A10$[+] MAIT-like cells (T08), and CD8[+] $TRDV2$[+] γδ T cells (T09) were excluded from these analyses due to their distinct TCRs.

## Monocle2 trajectory inference analysis

Macrophages and CD8[+] T cells developmental trajectories were inferred using Monocle2 (version 2.30.0) with default parameters[88,89]. Count matrices of specific cell type were obtained from "Seurat". Subsequently, these matrices were used with "newCellDataSet" to construct a CellDataSet. Next, order genes were selected using the default parameters in "monocle". The DDRTree method was then employed for subsequent dimensionality reduction. After ordering the cells, a trajectory was obtained. The visualization function "plot_cell_trajectory" was used to plot each group along the trajectory. BEAM (Branched Expression Analysis Modeling) was employed to identify genes regulated in a branch-dependent manner. Finally, the function "plot_genes_branched_pseudotime" was used to depict gene expression across different branches over pseudotime.

## Survival analysis

The Cancer Genome Atlas Ovarian Cancer Cohort (TCGA OC) gene expression matrices and clinical data downloaded from UCSC Xena were used to evaluate the prognostic performance of gene sets. Previously reported blacklisted genes: ribosome-protein-coding genes, and tissue dissociation operation-induced genes, including heat shock protein encoding were removed[90]. To assess the enrichment of the gene signatures in each sample, we employed the single-sample gene set enrichment analysis (ssGSEA) method in the GSVA (version 1.40.0) package. Subsequently, we performed survival analyses utilizing the log-rank test method. To establish an optimal cutoff value and to generate Kaplan–Meier survival curves, we employed the "ggsurvplot" function available in the survminer R package (version 0.4.9). Significance was assessed by the log-rank test statistics ($p$ values) between two groups.

## BayesPrism deconvolution analysis

We employed the BayesPrism computational method (version 2.2.2) with default hyperparameters for deconvolution, aiming to gauge the cell-type abundances within bulk transcriptomic data from TCGA. Initially, we annotated our single-cell RNA sequencing data. To alleviate computational load, we subsampled 1000 cells for each subpopulation, subsequently generating reference profiles for all cell states. To avert sex-specific transcriptional biases, genes located on the sex chromosomes were excluded from these reference profiles. Furthermore, we removed ribosomal protein-coding genes and mitochondrial genes to mitigate batch effects. The cell type frequencies estimated by BayesPrism were then utilized as cell type fractions in the TCGA data for subsequent analyses.

## Spatial transcriptomic experiment

OCT-embedded tissue samples were prepared for spatial transcriptomic analysis, and RNA quality was assessed prior to Stereo-seq library construction and sequencing using BGI capture chips. Raw data were processed with the BGI Stereomics pipeline, followed by cell type spatial mapping with cell2location and spatial neighborhood analysis using MISTy to evaluate cell–cell abundance relationships (see Supplementary Methods for details).

## Similarity analysis of clusters

We employed the strategy published by Cheng et al.[34] to assess the similarity between cell subpopulations in our dataset and those in external datasets. Briefly, we trained a logistic regression model with elastic net regularization using the cv.glmnet function from the R glmnet package, performing 10-fold cross-validation for each model. By calculating predicted logit values for each cell in the test data and converting them to probabilities, we visualized the similarity of clusters from the test data to those from the training data.

## SCENIC analysis

We performed SCENIC analysis (pySCENIC) (version 0.12.1) to predict and validate the potential transcriptional regulatory network in cell subsets with default settings. The activity of the 215 regulons for each subset was calculated using the AUCell function. To pinpoint the key transcription factors (TFs) driving the unique characteristics of each cell subsets, we further analyzed the specific regulons in different cell types by calculating the regulon specificity score (RSS).

## Single-cell CNV and clonality analysis

Infer copy number variation (CNV) algorithm was applied to identify malignant cells with large-scale chromosomal CNV in the cell clusters labeled as epithelial cells. We filtered genes with an average read count <0.1 among reference cells. We used endothelial cells, SMC cells, and pericyte cells to serve as references. Malignant epithelial cells were identified using infer CNV R package (version 1.10.1). Malignant cells and non-malignant cells were defined as the same as Gavish et al.[51]. Subclone CNV architecture of each HGSOC patient were determined using the subcluster method of infer CNV R package and visualized by a phylogenetic tree plot to illustrate tumor clonality and evolution, as reported before[91]. Subclones with less than five cells were filtered out.

## Defining NMF programs and meta programs of malignant epithelial cells

Consensus Non-negative Matrix factorization (cNMF) process was carried out for each sample from patients with HGSOC to capture the intratumor heterogeneity by performing the computational scheme reported before[51]. Briefly, the steps were conducted as follows: (1) cNMF was run by using seven different parameter values ($K$ = 4, 5, 6, 7, 8, 9 and 10), producing 49 programs for each sample; (2) the cNMF programs were then defined by criteria involved the robust within the sample, robust across sample and non-redundant within the sample. We obtained a total of 980 robust cNMF programs, which were then selected by ranking. Sixty-seven robust cNMF program generated in this process were clustered based on Jaccard similarity; (3) we used the clustering method to further define clusters of cNMF programs, applying an alternative way proposed by Gavish et al.[51]. Robust cNMF programs were compared to assess overlapping genes. Programs sharing at least 13 genes were identified as potential initiators of new

clusters. If multiple cNMF programs overlapped, the program with the most genes in common with the founder was selected to join and create a cluster; (4) genes with the top cNMF scores in either program was picked to expand the model program; (5) the same methodology was applied to create additional clusters, and seven MPs were finally identified. Subsequently, their enrichment in gene sets with functional annotations was evaluated.

## PROGENy

PROGENy (version 1.20.0)[92] was used to calculate 14 signaling pathways, whose scores were computed by a weighted sum of the product from expression and the weight of footprint genes. We select the top 100 significant genes for each pathway or filter by significance in responding to the recommendation.

## Cell-cell interaction analysis

To evaluate the interaction strength between different cell types, we employed cellphoneDB (version 3.0.0) with database v.2.0.0, using default parameters. We calculated mean values (strength) and cell communication significance ($p$ value < 0.05). For the same ligand-receptor pair, interactions where the strength between malignant epithelial cells with high MP6 signature scores and *SELENOP*+ macrophages exceeded that of malignant epithelial cells with low MP6 signature scores and *SELENOP*+ macrophages were selected (malignant epithelial cells providing ligands). These interactions are thought to potentially play a key role in the hypoxia-driven remodeling of *SELENOP*+ macrophages.

## Multiplex immunohistochemistry

Ovarian cancer tissues were collected and sectioned into 4 μm paraffin sections. The sections were stained using the Goat Anti-Mouse/Rabbit Multiplex IHC Detection Kit (18006, zenbio) according to the manufacturer's instructions. The following antibodies were used: anti-CD8a Antibody (1:300, Cat#PAQ6570, Abmart), anti-PD1 antibody (1:300, Cat#PH9964, Abmart), anti-GZMH antibody (1:300, Cat#BS2543, Bioworld), anti-SPP1 antibody (1:200, Cat#ab63856, Abcam), anti-SELENOP antibody (1:200, Cat#PA5-112707, Invitrogen), anti-HIF-1α antibody (1:200, Cat#PU774605, Abmart), anti-CD68 antibody (1:500, Cat#ab283654, Abcam), anti-Panck antibody (1:10, Cat#RAB-0050, MXB Biotechnologies), anti-F4/80 antibody (1:200, Cat#30325T, Cell Signaling Technology). The sections were incubated at 4 °C for 10 h with the primary antibody. After staining, multispectral images were generated with the multichannel laser confocal microscope (Nikon AXR) and Microsystems (605025, Leica).

## Cell lines and cell culture

The human ovarian cancer cell line OVCAR3 and CAOV3 were purchased from Wuhan Pricella Biotechnology Co., Ltd (CL-0178). The human ovarian cancer cell line COV362 were purchased from GuangZhou Jennio Biotech Co., Ltd. The mouse ovarian cancer cell line ID8 was provided by X.C. (China Medical University). THP-1 cells were provided by Y.S. (China Medical University). All cell lines were cultured in RPMI 1640 medium with 10% FBS. Cells were incubated at 37 °C with 5% CO$_2$. The hypoxic condition was induced by adjusting the oxygen concentration to 2% for 24 h, 48 h, and 72 h, respectively. THP-1 cells were differentiated into M0 cells by incubating in 100 ng/mL phorbolmyristate acetate (PMA, Cat#P6741, Solarbio) for 72 h for further experiments. Bone marrow-derived macrophages (BMDMs) were isolated from C57BL/6J (B6) mice as follows[93]. Bone marrow cells were collected by flushing the femoral marrow cavity of mice, filtered through a 70-μm cell strainer (Cat#352350, Corning), and subsequently centrifuged at 300 × $g$ for 5 min at 4 °C. Red blood cells were lysed by treating the bone marrow cells with red blood cell lysis buffer (Cat#R1010, Solarbio). Subsequently, the bone marrow cells were cultured in a DMEM medium containing 10% FBS and 100 ng/mL

Macrophage-Colony Stimulating Factor (M-CSF, Cat#Z03275, Genscript) for 7 d, thereby obtaining BMDMs. Peripheral blood mononuclear cells (PBMCs) were isolated from peripheral blood of HGSOC patients using the human peripheral blood lymphocyte isolation solution, and were cultured in RPMI 1640 medium supplemented with 10% FBS and 25 ng/mL Recombinant Human M-CSF (Cat#CR97, Novoprotein). The BMDMs and PBMCs were differentiated for 7 days for further experiments.

## T cell isolation and SELENOP treatment assay

OT-1 T cells from the spleen cells of female OT-I mice and human CD8+ T cells of HGSOC patients were isolated by the MojoSort™ Mouse CD8 T Cell Isolation Kit (Cat#480035, Biolegend) and MojoSort™ Human CD8 T Cell Isolation Kit (Cat#480129, Biolegend), and subsequently stimulated with 2 μg/mL anti-mouse/human CD3 (Cat#100339, Cat#300331, Biolegend) and 0.5 μg/mL anti-mouse/human CD28 (Cat#102115, Cat#302933, Biolegend) antibodies for 2 days, respectively. OT-1 T cells and human CD8+ T cells were maintained in PRMI 1640 medium with 10% FBS and 30 U/mL mouse/human IL-2 (Cat#Eg0456, Cat#HZ-1015, Proteintech). Ovalbumin (OVA) peptide (257-264) (Cat#T510212, Sangon Biotech) -activated OT-1 T cells were stimulated with PBS (control) or 25 ng/mL, 50 ng/mL, 75 ng/mL, 100 ng/mL, 250 ng/mL and 500 ng/mL recombinant mouse SELENOP (Cat#PDMM100013, Elabscience) for 48 h. CD8+ T cells were collected and the expression of membrane and intracellular protein was detected by flow cytometry.

## T cell killing assay

The mouse ovarian cancer ID8-OVA-luciferase cell line with stable OVA expression was constructed using lentivirus transfection. The ID8-OVA-luciferase cells and OVCAR3 cells were co-cultured with OT-1 T cells and human CD8+ T cells for 24 h, respectively. Flow cytometry was used to detect the total apoptosis of the ID8-OVA-luciferase cells and OVCAR3 cells collected.

## Cell transfection via lentiviral vectors and CRISPR/Cas9 system

Cell transfection via lentiviral vectors and CRISPR/Cas9 system was conducted as described below[94]. All lentiviral vectors and CRISPR/Cas9 system were constructed by Hanbio Biotechnology Co., Ltd (RNA sequences can be found in Supplementary Data 10). For the generation of *SELENOP*-overexpressing or *EPHB2*-deficient THP-1 cell lines, THP-1 cells were infected by lentivirus at a multiplicity of infection (MOI) of 60, with 10 μg/mL polybrene (Cat#HB-PB-500, Hanbio) added to enhance transduction efficiency. Fresh medium was changed 48 h post infection, supplemented with 1 mg/mL puromycin (Cat# P8230, Solarbio) for selection. For the generation of *Selenop*-overexpressing, *Selenop*-deficient or *Ephb2*-deficient BMDMs, BMDMs were infected by lentivirus at a MOI of 50, with 2 μg/mL polybrene added to enhance transduction efficiency and the culture medium with M-CSF was replaced after 48 h. Cell transfection efficiency was detected by Western blot.

## Macrophages and CD8+ T cells coculture system

OT-1 T cells and human CD8+ T cells were coculture with medium from BMDMs or THP-1 for 48 h. The expression levels of GZMB and PRF1 in CD8+ T cells were analyzed by flow cytometry, and their tumor-specific cytotoxicity was evaluated using the T cell killing assay.

## Antigen presentation assay

BMDMs were collected on day 7 and cocultured with the OVA peptide (257-264) for 6 h. Cells were washed and cocultured with naive OT-1 T cells for 48 h. The cross-priming capacity of macrophages was then evaluated by flow cytometry as GZMB and PRF1 production by OT-1 T cells, and the T cell killing assay was performed to evaluate their tumor-specific cytotoxicity.

## Selenite pretreatment, IFNγ stimulation and VEGFA treatment assay in macrophages

THP-1 cells were pretreatment with 70 nM Na$_2$O$_3$Se (Cat#HY-W686381, MedChemExpress) or equal volume PBS for 72 h, followed by stimulated with 40 ng/mL recombinant human IFNγ protein (Cat#C014, Novoprotein) or equal volume PBS for 48 h. BMDMs/MDMs were also stimulated with 40 ng/mL recombinant mouse/human IFNγ protein (Cat#C746, Cat#C014, Novoprotein) or equal volume PBS for 48 h. SELENOP expression was detected by extracting cellular proteins for Western blot, while SELENOP secretion was measured from collected cell culture supernatants. THP-1 cells/ BMDMs were stimulated with 50 ng/mL recombinant human/mouse VEGFA (Cat#GMP-CR96, Cat#CX73, Novoprotein) or an equal volume of PBS for 48 h, respectively. CRISPR/Cas9-mediated *EPHB2* deficient THP-1 cells, as well as vector control cells, were stimulated with 50 ng/mL recombinant human VEGFA for 48 h. Similarly, *Ephb2* knockdown BMDMs and their corresponding negative control cells were treated with recombinant mouse VEGFA under the same conditions. SELENOP expression was analyzed by Western blot using protein extracted from cell lysates.

## Enzyme-linked immunosorbent assay (ELISA)

The concentrations of SELENOP secreted by THP-1 cells, MDMs and in the serum of mice were measured by ELISA kits (Cat#E-EL-H2177, Cat#E-EL-M3044, Elabscience). Briefly, cell culture supernatant was centrifuged at $800 \times g$ for 10 min at 4 °C to remove the cell pellet and the serum of mice were separated by centrifugation of whole blood at $1000 \times g$ for 10 min. The detailed experiment processes were performed according to the manufacturer's instructions of the ELISA kits. Finally, the concentrations of SELENOP were measured via microplate reader (Infinite M200Pro, TECAN) capable of measuring absorbance at 450 nm.

## Cancer cells and macrophages coculture system

The conditioned medium was collected from $5 \times 10^5$ OVCAR3, CAOV3, COV362, and ID8 cells cultured under normoxic or hypoxic (2% O$_2$) conditions for 48 h or 72 h, respectively. THP-1 cells and BMDMs were incubated with conditioned medium for 48 h, and rescue assays were subsequently conducted by supplementing the cultures with 50 μg/mL anti-human/mouse VEGFA neutralizing antibody (Cat#HY-P9906, MedChemExpress; Cat# 512810, BioLegend). Following incubation, cells were harvested and total protein was extracted for Western blot analysis.

## Patient-derived organoids (PDOs) culture

PDOs was derived from primary lesions of one patient with HGSOC. Fresh tumor tissues were minced into small pieces and incubated in 1 mg/mL collagenase type IV (Cat#17104019, Thermo) at 37 °C for 30 min. The digested cell solution was filtered with 70 μm cell filters (Cat#352350, Falcon) and centrifuged at $300 \times g$ for 5 min. The cell pellets were washed by DMEM with 5% FBS and 1% penicillin/streptomycin (Cat#15140163, Thermo). Subsequently, the cell pellets were suspended in 80% Matrigel (Cat#356231, Corning) and drops of Matrigel cell suspension were allowed to solidify on pre-warmed 24-well culture plates at 37 °C for 30 min. After the drops solidified, advanced DMEM/F12 (Cat#12634028, Invitrogen) containing 50% culture medium supernatant of L-WRN cell (Cat#CRL-3276, ATCC), 1×B27 (Cat#17504-044, Thermo), 50 ng/mL EGF (Cat#AF-100-15, PeproTech), 200 ng/mL FGF10 (Cat#100-26, PeproTech), 1.25 mM N-acetyl-L-cysteine (Cat#A9165, Sigma), 10 mM nicotinamide (Cat#N0636, Sigma), 0.5 μM A8301 (Cat#2939, Tocris), 10 μM forskolin (Cat#1099, R&D), 1×GlutaMAX (Cat#35050-061, Thermo), 1×Primocin (Cat#ant-pm1, Invivogen), 1×HEPES (Cat#15630080, Gibco) and 10 μM Y27632 (Cat#S1094, Selleck) were added to each well. Fresh medium was added every 3 days. TrypLE was used to passage cultures. For hypoxia culture, PDOs were incubated in normoxia (control) or hypoxia culture (2% O$_2$) incubator for 72 h or 96 h,

respectively. The protein of PDOs was collected using lysis buffer for Western blot analysis. The conditioned medium was collected from ~ 200 clones PDOs cultured under normoxic or hypoxic (2% O$_2$) conditions for 96 h. MDMs were then incubated with the conditioned medium for 48 h. Following incubation, cells were harvested and total protein was extracted for Western blot analysis.

## Mice

OT-1 (C57BL/6J (B6)-Tg(TcraTcrb)1100Mjb/J) and CD45.1 (B6.SJL-PtprcaPepcb/BoyJ) mice were generously provided by the First Hospital of China Medical University. C57BL/6J (B6) mice were obtained from GemPharmatech Co., Ltd. Age-matched (6–10 weeks) female mice were used in all mouse experiments. Female mice were selected because ovarian cancer is a gender specific disease. The number of animals (biological replicates) are indicated in figure legends. The female C57BL/6J (B6) mice were housed at the Experimental Animal Center of China Medical University, with commercial standard mouse diet containing 0.13 mg/kg Se (Available from: https://www.spfbiotech.com/report). All the mice were raised in an SPF environment with a temperature of 24 °C, a relative humidity of 50% to 60%, and a 12-h light/12-h dark cycle. All procedures on animals were performed under aseptic conditions and using sterile instruments.

## Tumor model and treatment

To generate orthotopic tumors, eight 10-week-old female C57BL/6J (B6) mice were anesthetized by inhalation of isoflurane (5% in oxygen) in an induction chamber and anesthesia maintained at 2.5–3.0% isoflurane delivered via nosecone during all procedures. A small incision was made at the dorso-medial position directly above the ovarian fat pad, with a secondary small incision through the peritoneal wall. The ovarian fat pad was externalized and stabilized with a bull clip. ID8-luciferase cells ($3 \times 10^6$ cells suspended in 15 μl sterile PBS) were injected underneath the right ovarian bursa. The peritoneal wall was sutured closed using 6/0 suture and closure of the incision with surgical staples[95]. The incisions were closed using sterile sutures. The general health of the mice was assessed daily, and the inoculated tumors were observed weekly using an in vivo imaging system. From day 14 after the initial injection, the mice were intraperitoneally administered a control treatment of IgG (2.5 mg/kg body weight, Cat#BE0089, Bioxcell) or experimental treatment of anti-VEGFA antibodies (2.5 mg/kg body weight, Cat#512810, BioLegend) twice a week for 5 weeks. Thereafter, the mice were sacrificed on day 49, and serum samples were collected for ELISA analysis as detailed in the Methods section Enzyme-linked immunosorbent assay (ELISA). For subcutaneous xenograft model, a total of $1 \times 10^7$ cells in 200 μL of PBS were subcutaneously injected into the right dorsal flank of 6-week-old female C57BL/6J (B6) mice. For ethical considerations, mice were sacrificed when the tumor volumes reached 1000 mm³. Tumor tissue was minced then digested in a digestion solution (1 M HEPES, 1 mg/mL collagenase I, 1 mg/mL collagenase IV, 0.2 mg/mL DNase I, 0.2 mg/mL Hyaluronidase, 1% Penicillin-streptomycin complex and 10% FBS). The dissociated cells were used for subsequent flow cytometry. For adoptive cellular transfer experiments, on day 14 after the establishment of the orthotopic ovarian cancer model, BMDMs or BMDMs-KD1-Selenop ($3 \times 10^6$) from female C57BL/6J (B6) mice (CD45.1) were administered via intraperitoneal injection. In the BMDMs & anti-VEGFA/BMDMs-KD1-Selenop & anti-VEGFA group, mice were administered with both BMDMs/BMDMs-KD1-Selenop ($3 \times 10^6$) and anti-VEGFA antibodies (2.5 mg/kg body weight, Cat#512810, BioLegend) on day 14, with dosing performed twice weekly for 2 weeks. Fourteen days after adoptive cellular therapy, mice were sacrificed, and orthotopic ovarian lesions as well as peritoneal metastases were collected. Tumor tissues were mechanically dissociated to obtain single-cell suspensions, and analyzed by flow cytometry.

## Statistical analysis

Statistical significance was defined as $p < 0.05$ after Benjamini-Hochberg (BH) correction. Analyses were performed using R (v4.2.2), Python (v3.9.0) or GraphPad Prism (v8.0.2). Group comparisons involved Student's $t$ tests, Wilcoxon rank-sum tests, Kruskal–Wallis tests, chi-square tests, one-way ANOVA or two-way ANOVA. Survival analyses used log-rank tests, concentrating on the proportions of $SELENOP^+$ and $SPP1^+$ macrophage subclusters. The ggsurvplot function from the survminer package was employed to determine cutoff values and generate Kaplan–Meier survival curves. Further details on statistical methods can be found in the results section and figure legends. Owing to double-precision numerical constraints, $p$ values less than $2.2 \times 10^{-16}$ are assigned a standardized value of $2.2 \times 10^{-16}$, as previously reported[96].

## Reporting summary

Further information on research design is available in the Nature Portfolio Reporting Summary linked to this article.

## Data availability

The scRNA-seq, WES and ST data generated in this study have been deposited in the Genome Sequence Archive for Human (GSA-Human) database under accession codes HRA006423 and HRA003236. The scRNA-seq, WES and ST data are available under restricted access for reasons related to human genetic resources; access can be obtained by contacting the corresponding author (X.Z., xzhou@cmu.edu.cn and C.S., suncydoctor@gmail.com). Raw data access requests will be processed within 3 months by following the guidelines for Genome Sequence Archive for noncommercial use. Data access will be granted for 1 year. The raw scRNA-seq, WES and ST data are protected and are not available due to data privacy laws. The detailed data generated in this study are provided in the Supplementary Information/Source Data file. The previously published data used in this study are available in the NCBI Gene Expression Omnibus (https://www.ncbi.nlm.nih.gov/gds/) under accession no. GSE184880, GSE203612, GSE266577; GSA-Human (https://ngdc.cncb.ac.cn/gsa-human/) under accession code HRA002184, HRA002767; Synapse (https://www.synapse.org/msk_spectrum) under accession number syn52458609; and https://doi.org/10.6084/m9.figshare.22147103. Source data are provided with this paper.

## Code availability

No novel algorithms were created for this study. All code used for analysis is available in GitHub (https://github.com/NCarticle2025/new).

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

## Acknowledgements

This project was supported by funding from the National Natural Science Foundation of China (grant no. 82072885 to X.Z.), Noncommunicable Chronic Diseases-National Science and Technology Major Project (grant no. 2025ZD0545600 to X.Z. and C.Y.S.), Science and technology plan joint plan of Liaoning Province (grant no. 2023JH2/101700193 to X.Z.), 345 Talent Project of Shengjing Hospital of China Medical University (X.Z.), Science and technology plan joint plan of Liaoning Province (grant no. 2023JH2/101700132 to Q.L.), the National Natural Science Foundation of China (grant nos. U25A20118 and 82303372 to X.Z.H.), the Department of Science & Technology of Liaoning Province (grant no. 2024JH3/10200031 to X.Z.H.), Open Research Fund of NHC Key Laboratory of Prevention and Treatment of Central Asia High Incidence Diseases (grant no. KF202503 to C.Z.F.). We acknowledge the developers of OpenAI's ChatGPT for language assistance and BioRender for providing the illustration tools used in the preparation of this manuscript.

## Author contributions

Conceptualization and methodology: X.Z., C.Y.S., and X.Z.H.; sample and patients' clinical information collection: Q.L., S.Y.L., B.Y., and S.M.Z.; investigation and formal analysis: T.H.W., S.Y.Z., Q.Z., X.Y.W., Q.L., C.Z.F., X.Y.S., Y.L.Z., and L.R.Q.; writing: Q.L., C.Z.F., T.H.W., S.Y.Z., X.Y.W., and J.X.L.; revision: X.Z., C.Y.S., C.Z.F., and X.Z.H.; supervised the project and provided valuable critical discussion: X.Z., C.Z.F., C.Y.S., Q.L., Q.Z., and G.C.; funding acquisition: X.Z. and Q.L.

## Competing interests

The authors declare no competing interests.
