## [Transparent Peer Review file · Nature Communications]

Hypoxia-Driven Remodeling of SELENOP⁺ Macrophages Shapes T Cell Dynamics and Promotes Ovarian Cancer Metastasis

Corresponding Author: Professor Xin Zhou

This manuscript has been previously reviewed at another journal. This document only contains information relating to versions considered at Nature Communications. Parts of this Peer Review File have been redacted as indicated to maintain the confidentiality of unpublished data.

Version 0:

Reviewer comments:

Reviewer #1

(Remarks to the Author)

The study of Liu et al., provides a comprehensive analysis of the tumor microenvironment (TME) in high-grade serous ovarian cancer (HGSOC) metastasis using advanced techniques like single-cell RNA sequencing and spatial transcriptomics. The authors identified critical cellular and molecular mechanisms influencing tumor progression and immune suppression. The SELENOP⁺ macrophages were found to enhance antitumor immunity by activating CD8⁺ T cells, whereas SPP1⁺ macrophages promoted tumor metastasis. Hypoxia-driven malignant epithelial cells were shown to reprogram SELENOP⁺ macrophages through VEGFA-EPHB2 signaling, diminishing the immune response. This interaction was associated with HGSOC metastasis. Then, anti-VEGFA therapy increased the proportion of SELENOP⁺ macrophages and CD8⁺ T cells, highlighting its potential for improving therapeutic outcomes. The study also revealed that BRCA-mutated HGSOC tumors exhibit a more active immune response compared to BRCA wild-type, suggesting the need for targeted immunotherapeutic strategies.

Key findings of the study include:

- Identification of two distinct macrophage subsets: SELENOP⁺ macrophages (M06) with antigen-processing and presentation capabilities, and SPP1⁺ macrophages (M09) involved in extracellular matrix remodeling. Also, M06 macrophages are associated with better patient outcomes, whereas M09 macrophages are linked to poor outcomes.
- Demonstration that SELENOP⁺ macrophages possess immunostimulatory properties, potentially enhancing the cytotoxicity of CD8⁺ T cells via SeP. This is in contrast with pro-tumorigenic effects of SPP1⁺ macrophages.
- SELENOP⁺ macrophages are enriched in adnexal tumors, but their presence decreases as the tumor metastasizes.
- Identification of seven consensus metaprograms (MPs) within malignant epithelial cells, each with distinct gene signatures
- Hypoxia-driven malignant epithelial cells reprogram SELENOP⁺ macrophages via VEGFA-EPHB2, as well as Hypoxia and FTL promote metastasis and therapy resistance. The study shows that VEGFA inhibition increases SELENOP⁺ macrophage proportions in primary tumors and peritoneal metastasis.
- The study also characterizes the heterogeneity of CD8⁺ T cells and their functional states in different anatomical sites.

Although the study is nicely presented as such, several major concerns should be addressed before the publication:

1) Limited Sample Size: The authors acknowledge that the study may be limited by a relatively small sample size, especially in some groups such as EAT. This may reduce the significance of the findings. We suggest to increase the sample size, at least for profiling the macrophage subsets.

2) Lack of Treatment Response Data: While the study provides an in-depth analysis of the TME, it does not incorporate the effects of chemotherapy or other treatments on the TME or link molecular findings to clinical outcomes like platinum-free interval (PFI). All the patient samples investigated by the authors are from treatment-naive cohorts. I suggest that the authors should investigate:

- what is the effect of chemotherapy on macrophages distribution and characterization in HGSOC?
- how does the chemotherapy affect the functionality of M06 and M09 populations, in terms of immunostimulatory properties?

3) The authors demonstrate that hypoxia-driven malignant epithelial cells reprogram M06 macrophages, and for this they did experiment using only OVCAR3 cell-derived conditioned media. OVCAR3 is not the best representative of HGSOC cell lines. I suggest the authors should use at least 2 more cell lines, and choose some BRCA1 positive and some BRCA1 negative (See Domcke et al, 2013, <https://www.nature.com/articles/ncomms3126>)

4) The study does not focus on the detailed interaction of CD8+ T cells with myeloid cells, particularly after chemotherapy, as is found in recent study (<https://doi.org/10.1016/j.ccell.2024.11.005>). The study also lacks focus on Myeloid cells. The authors should address these in treatment-naive and post-chemotherapy.

5) Address the relevance of treatment impact on macrophage subsets in the discussion as well. It will be interesting to see whether chemotherapy positively or negatively influences M06 or M09 enrichment and their functional roles.

Reviewer #2

(Remarks to the Author)

Previous work from this group showed that CD8+ T cells were tumour-specific in adnexal tumours but functioned as bystanders in HGSOC metastasis, which involved paired scRNA seq and scTCR-seq analyses. The intention of this manuscript is to explore the dynamic changes in CD8+ T cells during HGSOC metastasis, i.e., alteration to functional and differentiation states. The manuscript is a complicated paper full of bioinformatic data analysis (densely populated figures that are enriched with complex data). The information shown supports the narrative of the paper, showing intratumoral heterogeneity and complex epigenetic changes that causes remodelling of the tumour microenvironment in HGSOC. The finding that hypoxia-driven malignant epithelial cells could reprogram SELENOP+ macrophages is backed up with additional data. The addition of the mouse work adds weight to this study and a mechanism of tumour progression/survival/reduced immune surveillance, showing that VEGFA signalling is a key component and highlights that anti-angiogenic therapies could be used as a treatment these tumours. This work is an important contribution to our understanding of HGSOC and is a resource for future work.

1) Last paragraph of the introduction is very long and acts as an abstract of the full manuscript. This is inappropriate, as the introduction should rather summarise the intention/rationale of the study and study approach.

2) Should consider moving some methods into supplementary to reduce the length of the manuscript.

3) Figure 2B, the dark colour of the extreme -1 and +1 are heavily blackened, which results in not knowing what end of the colour the T06_CD8T-CXCL13 cell population is for the exhausted and TEX boxes, either very dark/blackened red (+1) or very dark/blackened blue (-1). I suggest taking out the black shading from these boxes. While not as bad, same could be said for Figure 2A with several dark blue spots that could be misinterpreted in the figure as being over-expressed (especially as the spots are very small). I suggest to make all the dots larger, as there is room in the figure for this. Same can be said for Figure 3A and 3D.

4) Line 178: 'Subcluster T06_CD8T-CXCL13, defined by the expression of CXCL13, CTLA4, and TIGIT, exhibited a transcriptional signature indicative of exhaustion, thus classified as exhausted T cells (Tex) (Fig. 2a,b).' Please add a sentence to explain why this expression signature is indicative of exhaustion and reference a paper if possible to help readers that might not be as knowledgeable.

5) Figure 2D, in annotation 'expansion index' is spelt wrong. In this figure there should also be axis annotations that have been omitted.

6) Figure 2F: in result text, lack of annotation in the figure and a different description in the figure legend, i.e., what is 'Ro/e'? Currently, it is ambiguous to what the data represents. Can the authors improve the labelling of the figure and description in the text and figure legend to make this result clearer.

7) Line 595: Figure 6h, siEPHB is not effective at knocking down siEPHB. I would suggest using other siRNA that are more effective and employ EPHB drug inhibitors to add validity to this work.

8) Line 1191: Amplicon lengths should be provided here, and whether dissociation curves were also run routinely to ensure specificity of reactions.

Reviewer #3

(Remarks to the Author)

This study reveals a novel cross-talk between macrophages, T cells and epithelial cells during the progression of high-grade serous ovarian cancer which centres around SELENOP+ macrophages. This cell population was identified by an extensive single cell RNAseq approach considering both the temporal aspect of tumor development as well as different anatomical sites. While this first screening approach is very convincing and well justified to identify the relevant cellular players the mechanistic part analysed by cell culture and mouse experiments remains rather unclear and needs to be substantially improved and extended to support the conclusions drawn by the authors.

The identification of the proposed cross-talk within the tumor microenvironment might not only be relevant for ovarian cancer but could extend to additional cancer types given that this SELENOP+ macrophage population has also been identified in lung cancer before. Therefore, the manuscript would be of great interest to the broad readership of Nature Communications. However, in my view, the manuscript still lacks mechanistic details and controls and should be thoroughly revised to address the following comments.

The whole manuscript should be more clearly structured according to the information provided in the abstract. So far, it is very complicated for the reader to have data provided as figures, extended data figures, and supplementary figure. In my

view, a fundamental restructuring of the data is really needed to support the story described in the abstract. Thus, later sections e.g. about immune checkpoints or the section starting at line 436 about dendritic cells are not really related to this story and should be excluded. Also the comparison between different anatomical sides within one patient could be left out. At least it is not getting clear how this relates to the main story line.

Based on the abstract, the title could also more specifically describe the novel interactions identified.

Throughout the manuscript: Please introduce the genes/proteins that were discussed in more detail. This is mandatory especially when addressing a broader readership.

Introduction, line 63: Please elaborate on the results gained in the previous publication (Ref 17) a bit more. Which questions remained unanswered after this publication? Why did you now follow up an alternative approach?

Introduction, lines 72-94: This section intensively summarizes the generated results but I miss a clear aim of the study.

Results, line 114: The authors conclude that the anatomical site was the primary driver of heterogeneity. How does this relate to the following results which mainly take the tumor progression / stage into account? Here it would have been also helpful for my understanding to comment on the dependency of samples because different anatomical samples are taken from one patient (dependent variables) while different stages are analysed in independent samples from different patients. When looking at progression, the comparison between tumor and fluids is not conclusive. Why was this incorporated into the analyses? Effects of anatomical localization and tumor progression should be more clearly separated from each other (or even leave out anatomical localization results).

Figure 1 and 2 extensively characterize the samples analysed. Those results could be narrowed down a bit to provide a clearer focus.

Figure 3: the SELENOP+ macrophage cluster was identified. It is very important to keep in mind that this is only gene expression data which does not necessarily relate to protein expression because selenoprotein synthesis is strictly dependent on selenium availability which, under conditions with limited supply, overwrites a transcriptional regulation. Please provide data showing the specificity of the SELENOP immunohistochemistry staining and Western Blot results (e.g. by staining cells cultured under conditions with and without selenium or by using specific blocking peptides). Unfortunately, there are many unspecific SELENOP antibodies available.

As only very shortly stated in line 334-336, SELENOP is secreted by hepatocytes but also other cell types (including macrophages?) and can be taken up by other cells via the receptor LRP8. SELENOP is broken down to release selenium which is used for the synthesis of other selenoproteins such as GPX or TXNRD. Those parameters definitely need to be analysed to show that SELENOP is taken up by T cells after culturing them with recombinant SELENOP. I would recommend to analyse GPX1 protein expression but other selenoproteins will work as well. Alternatively, the intracellular selenium concentration could be measured.

For all cell culture experiments, it is mandatory to know the basal selenium concentration in FCS and as a proof of concept, cells should be also treated +/- selenite. Using THP1 macrophages, the authors should show that macrophages indeed secrete SELENOP. To strengthen the results, the authors could perform this experiment also with monocytes isolated from patient buffy coats. The conditioned medium from THP1 or other monocytes/macrophages could be used in addition to recombinant SELENOP to treat T cells. Also treating THP1 cells with IFN γ should be performed in cells +/- selenite pretreatment for 72 h.

Section on epithelial cells (starting at line 461): The rationale for the transition from immune cells to epithelial cells is not getting clear. This section should be shortened. For example, the link to ferritin is not supporting the story line. This is also not mentioned later in the discussion again.

Line 587: Briefly introduce the VEGFA-EPHB2 pathway.

The knockdown efficiency of EPHB2 is quite poor. Is there an alternative way of inhibiting the pathway? A knockout would be more conclusive to show a dependency.

Line 594: If the effect of VEGFA treatment should be reversed by the EPHB2 knockdown, please combine the VEGFA treatment with the knockdown to draw this conclusion.

Using patient-derived organoids is an interesting alternative approach but as stated above this should be combined with primary human monocytes/macrophages and not again with the THP1 cell line.

Line 613, the in vivo model: The SELENOP+ macrophages at this stage really express more SELENOP protein as this has been characterized by FACS, in contrast to the RNAseq data which only rely on mRNA data (which might be quite irrelevant for selenoprotein expression). Please use another term for these macrophages to make this difference clearer to the reader. In addition, the authors should provide information about the selenium concentration in the mouse chow. In addition, it would be very interesting to also measure SELENOP concentrations in the serum of mice to exclude a systemic effect of VEGF on SELENOP secreted from the liver.

Please quantify primary tumor, omental metastasis and peritoneal metastasis formation and not only provide fluorescence images and FACS results.

To really show that macrophage-derived SELENOP is needed to modulate the VEGF effect on tumor metastasis in vivo, the authors should use SELENOP knockout mice with a macrophage specific SELENOP knockout (SELENOP Δ Mye mice).

Line 621: The last section of the results is again out of the focus of the manuscript and should be clearly shortened.

The model provided in Fig. 8 is very interesting but it is not really supported by the data provided in this manuscript (see comment above).

Discussion

The discussion of the novel crosstalk should be extended. There is much more literature on SELENOP and its mode of action which should be discussed here. In addition, a putative role of the trace element selenium could be discussed as well. For the discussion of hypoxia-induced effects on SELENOP refer to Becker et al., 2014, Hypoxia reduces and redirects selenoprotein biosynthesis.

The second part of the discussion starting in line 688 is again out of topic and has not much to do with the results described in the abstract.

Minor comments:

Abstract, line 37: malignant epithelial cells

Please use SELENOP as abbreviation for selenoprotein P as this is the official nomenclature.

Line 297: I would mention here again that “elevated phagocytosis function” is another result of the KEGG pathway analysis. Otherwise, one could expect a functional readout.

Line 334: Please delete “theoretically”.

Fig. 6i is not very informative for the reader.

Line 661, Ref 66-68: Only Ref 66 is supporting the respective information about the macrophage subset in lung cancer.

Lines 805-806: specify the title of the section

Line 1225: Is the amount injected (15 μ l) correct?

Line 1238: statistical analysis One-Way ANOVA is missing (is written in figure legends)

Reviewer #4

(Remarks to the Author)

This work by Liu et al. uses multiple state of the art single-cell multi-omics and spatial transcriptomics methods to map the spatiotemporal heterogeneity of HGSOc metastasis and tumor microenvironment (TME) and identifies two major populations of tumor-associated macrophages (TAMs): SELENOP⁺ and SPP1⁺, respectively, which undergo dynamic transition in the setting of hypoxic malignant ovarian cells to subvert immune-mediated tumor control, thus facilitating HGSOc metastasis. Some of the findings are consistent with what others have found in various types of cancers but not all of them have been functionally and definitively validated in patients or in vivo model systems. Given the limitation, it is important that, at a minimum, the pro-metastatic role of SPP1⁺ macrophages, as postulated in this study, should be unequivocally demonstrated by taking at least one of the suggested approaches as follows:

1. SPP1-targeting antibodies: use neutralizing antibodies against SPP1 to block its activity and downstream signaling.
2. SPP1 knockout: Global and conditional SPP1 KO mice are commercially available.
3. Receptor Blockade: SPP1 interacts with integrins (e.g., α 3 β 3, α 3 β 5) and CD44. One could use receptor-blocking antibodies or peptides (e.g., RGD peptides for integrins).

There are also concerns regarding the claim of direct involvement of SELENOP⁺ macrophages in priming T cells, as there is no direct evidence showing that SELENOP⁺ macrophages can activate CD8⁺ T cells. In contrast, previous studies (PMID: 18424738, 20530259) show that SELENOP is highly upregulated in M2 macrophages, which are generally considered more pro-tumorigenic than M1 macrophages.

Some additional specific issues of concern:

1. Fig. 2- Out of on the scRNA-seq data (Extended Data Fig. 2a), only CD8⁺ T cells are analyzed in detail (Fig. 2). However, CD4⁺ T cells are also abundant in the tumor. Why no further analysis on the CD4⁺ T cell population? Are regulatory T cells enriched in late state tumors?

2. Fig. 3h- The resolution of spatial transcriptomics data is too low. T04_CD8T-GZMH gene expression doesn't show clear cluster on the spatial transcriptomics data. It appears that the gene expression is mapped to the whole slide. CD8⁺ T cell clusters can be observed on the immunofluorescence staining slides (Fig. 3j). However, based on the Visium data (Extended Data Fig. 3e), the correlation between SELENOP⁺ macrophage and T04_CD8T-GZMH abundance is very weak (R = 0.11, Extended Data Fig. 3e).

3. Fig. 3k and Extended Data Fig. 3i- To study the effect of SELENOP⁺ macrophages on CD8⁺ T cell function, the authors use SeP protein to stimulate the OT1 cells. There is no dose-dependent change on the GZMB and PRF1. Can SELENOP⁺ macrophages directly prime CD8⁺ T cells? To verify the role of SELENOP⁺ macrophages, co-culturing of OT1-specific SLINFEKL peptide-loaded-SELENOP⁺ macrophages with OT1 cells should be performed. If it is not feasible to directly isolate SELENOP⁺ macrophages, the SELENOP protein can be overexpressed in macrophages, which can then be used to prime CD8⁺ T cells.

4. Fig.6f- The experiment should be performed with purified macrophages from the spleen of mice or bone marrow derived macrophages for more physiological relevance.

5. Fig.6g, h- The EPHB2 knockdown efficiency is very low. However, the SELENOP expression increases dramatically in the siEPHB2 cells. Is it due to off-target effect of siEPHB2? Anti-VEGF neutralizing antibody can be used to verify the role of VEGF signaling on macrophages. The result should be validated using purified macrophages or bone marrow derived macrophages.

6. Fig.7b- On day 14, the mouse in the anti-VEGF group seems to show stronger luminescence intensity than the control mouse, suggesting that anti-VEGF may not efficiently suppress primary tumor growth. Quantification of luminescence intensity from multiple mice per group should be provided with statistics.

7. Fig.7c- How are SELENOP⁺ macrophages identified using flow cytometry? The gating strategy should be provided. In this mouse model, do SELENOP⁺ macrophages also co-localize with CD8⁺ T_{pex}? What is the phenotype of this SELENOP⁺ macrophage population? Are they similar to M1 or M2 macrophages? Since the SELENOP⁺ macrophages can be found in this mouse model, those macrophages can be used to test their capacity to prime CD8⁺ T cells in vitro and in vivo.

Reviewer #5

(Remarks to the Author)

****REVIEW \- Site-specific Tumor Microenvironment Remodeling associated with HGSOC metastasis****

The paper maps the full spatiotemporal progression of high-grade serous ovarian cancer (HGSOC). Thus it improves over the current literature, which predominantly focuses on the tumor microenvironment (TME) of advanced HGSOC. The authors employ multimodal profiling techniques including scRNA-seq, spatial transcriptomics, scTCR-seq, RNA-seq, organoids, and orthotopic syngeneic models. They report spatial proximity between SELENOP+ macrophages and precursor exhausted CD8+ T cells. Their findings suggest that hypoxia-driven remodeling of the TME (via the malignant epithelial cells) plays a critical role in facilitating metastasis, and indicate the potential for combining anti-VEGFA therapy with immunotherapy.

The question addressed by this work is important, and overall its execution is promising. I think this will be a useful contribution to scientific progress in this field. I particularly appreciate that the authors chose to validate some of their findings using publicly available TCGA data (Fig. 3f). I have a few comments that might help improve the impact of this work and its presentation.

1. ****Ratio of observed to expected:**** The usage of the ratio of observed to expected seems inconsistent. For example, in Fig. 2f, each row sums up to 1, whereas in Fig. 3b, this is not the case. This should either be investigated, or the methodology should be explained more clearly.
2. ****Methodology of NMF to define programs:**** the way that the metaprograms are defined seems unnecessarily convoluted. I would rather start with the GO approach to get initial cluster annotations, and highlight the similarities with existing signatures afterwards.
3. ****Grammatical errors and typos:**** there are some minor grammatical errors within the text (e. g. lines 31, 57, 115, 318, 320, 370, 444, 585, 630). Fig. 2b refers to precursor exhausted T cells as Texp, whereas Tpex is used everywhere else. Fig. 2d should be "expansion index". Line 130: "mere" seems like the wrong word here.
4. ****Presentation of expression data:**** the authors use a rainbow-style colormap to show expression values in Fig. 2a, 3a, 4j. There is however no standardization as to which color represents 0. I would instead encourage the use of a divergent color palette with a clear center, such as the coolwarm palette already used in Fig. 2b. The labels of the colorbar in Fig. 3a are not aligned properly.
5. ****Fig. 2g-i:**** it is not clear why clusters T03 and T07 were excluded here.
6. ****Fig. 4d:**** Once again, the choice of only showing clusters M05, M06, and M09 is not clearly explained in the text. I would also appreciate it if the same analysis was performed on all macrophage clusters (or at least the TAMs) and put in the supplement (same for the previous point).
7. ****Fig. 4e:**** would benefit from adding plot titles directly in the figure, showing which enrichment analysis belongs to which state.
8. ****Fig. 6c:**** there should be a distinction between 0 and negative importances.
9. ****Extended Data Fig. 4:**** in every other Ro/e plot, the numbers are shown, but not in this one.
10. ****Code Availability:**** the authors claim that the code is available (Line 1254), however there is no link to a GitHub repository (or equivalent).

Reviewer #6

(Remarks to the Author)

Liu et al. characterized the spatial heterogeneity of HGSOC across different progression stages and tissue sites using scRNA-seq and spatial transcriptomics. Their study identifies the co-occurrence and spatial colocalization of SELENOP+ macrophages and precursor exhausted CD8+ T (Tpex) cells, revealing that SELENOP+ macrophages promote CD8+ T cell activation in early tumors. They further showed that as the tumor progresses and metastasizes, SELENOP+ macrophages are reprogrammed into SPP1+ macrophages by hypoxia-induced malignant epithelial cells through VEGFA-EPHB2 signaling. Notably, anti-VEGFA therapy restores SELENOP+ cells in ovarian cancer cell lines and mouse models. While the study provides insights into the role of macrophage subsets in HGSOC progression, it would benefit from a more rigorous analysis. The key findings are not effectively emphasized, some results are overly descriptive, and the mechanistic validation requires stronger supporting evidence.

Here are my major concerns:

1. The authors highlight the inclusion of early-stage patients as a strength, but similar comparisons have been made in previous studies (such as PMID35675036, 38278958). While one is cited in the discussion, these works should also be discussed. Additionally, the authors should compare their findings with these studies to clarify how their results align or differ from existing literature.
2. The authors repeatedly use the term "significant" without indicating statistical tests, such as in the Ro/e analysis (Fig. 1f, Extended Fig. 2i, j). They should either remove or revise their claims or perform appropriate statistical tests. Specifically, as the Ro/e analysis is throughout the study and underpins several conclusions, a chi-square test should be applied to assess tissue preference. Additionally, the equation used for Ro/e analysis should be explicitly provided.
3. Although Tregs are not the focus of this study, their absence is notable. The authors should address their relevance in HGSOC progression. Additionally, since T cells and NK cells were initially grouped together, separating them into distinct clusters would allow for more precise characterization.
4. Quantifying the spatial distribution of macrophage and Tpex is valuable. However, the authors should include zoom-in images to visually support the computationally inferred spatial relationships between SELENOP+ or SPP1+ macrophage and Tpex.
5. The SELENOP+ macrophages appear abundant in scRNA-seq and Visium data but are scarce in IF staining (Fig. 3j and 4b). The authors should provide an explanation for this discrepancy, such as differences in detection sensitivity, technical

limitations, or biological factors affecting protein versus transcript expression.

6. In the BRCA analysis, the BRCAmut group includes two EAT samples, comprising half of the total samples, while the wt group consists only of late-stage samples. This raises the concern that the observed high proportions of T_{pex}, T_{ex}, and SELENOP⁺ macrophages in BRCAmut may be driven by differences in sample composition rather than the BRCA mutation itself. The authors should clarify how they account for these confounding factors.

7. Given the significant growth differences between PDOs and the OVACR3 tumor cell line, both models were subjected to 48-hour normal and hypoxic cultures. Have alternative hypoxic culture durations been explored? Additionally, the authors should specify the cell numbers of PDOs and OVACR3 used to prepare the conditioned medium.

8. The authors show that SELENOP⁺ macrophages activate CD8⁺ T cells via SeP, but validation is limited. While exogenous SeP increased GZMB and PRF1 expression and co-culture assays assessed cytotoxicity, these alone are insufficient. Blocking SeP secretion in macrophages would strengthen mechanistic validation.

9. The bioluminescence images show only one representative mouse per group. Can the authors confirm whether the same mouse was imaged on days 7, 14, and 42? To enhance data reliability, it is recommended that images of at least three mice per group be included.

Minor points:

1. The authors should provide the full name of ISG and PARPi and others if they are first mentioned.
2. The authors should indicate the cell fate on top of the corresponding barplot in Fig4e for better readability.
3. In the first paragraph of the first results section, line 12, there is an extra "and" at the end of the sentence.

Reviewer #7

(Remarks to the Author)

Reviewer #8

(Remarks to the Author)

Reviewer #9

(Remarks to the Author)

Version 1:

Reviewer comments:

Reviewer #1

(Remarks to the Author)

The authors addressed all my concerns. I am particularly please that they validated their findings using more cell lines (beside just OVCAR3) and also, addressed the effect of chemotherapy on the different populations of macrophages. Also, the cohort size was improved. The article should be accepted for publication.

Reviewer #2

(Remarks to the Author)

The authors have addressed all six reviewers' constructive concerns and suggestions, substantially improving the manuscript. The revisions have enhanced the overall flow and narrative, and the figures have been improved to resolve earlier shortcomings. I have no specific concerns. The study advances our understanding of ovarian cancer progression, highlighting the role of hypoxia signaling in driving SELENOP⁺ macrophages and their interface with T cells. This work will be of considerable interest to the readership of Nature Communications and the broader research community, with clear clinical implications.

Reviewer #3

(Remarks to the Author)

The authors have made a great effort to revise their manuscript and provide many novel data. Overall, I think the manuscript has been significantly improved. My comments were satisfactorily addressed.

Regarding the selenium concentration in FCS the authors answered:

The basal selenium concentration in fetal calf serum (FCS), measured using a selenium ion detection kit (Cat#abs580188, absin), was 1.693 mmol/L.

This cannot be possible. Usually, the selenium concentration in FCS is in the nano molar range. We treat cells with 50-100 nM selenium to increase seleoprotein expression. Such a high selenium concentration in the mM range would be close to toxicity for most cells.

This needs to be clarified.

Reviewer #4

(Remarks to the Author)

The authors have satisfactorily addressed all our concerns and issues.

Reviewer #6

(Remarks to the Author)

The authors have adequately addressed my previous comments regarding both the analyses and the experimental validation. The manuscript has been improved accordingly.

Reviewer #7

(Remarks to the Author)

Reviewer #8

(Remarks to the Author)

Reviewer #9

(Remarks to the Author)

My established referee and I have discussed the revised manuscript. We are both satisfied with the revision and have no additional comments. However, since my established referee is travelling and is unable to upload the report, I am submitting the report instead.

We have addressed each comment and point-by-point responses are provided below:

EDITORS' Comment:	1
REVIEWER 1:	3
REVIEWER 2:	10
REVIEWER 3:	16
REVIEWER 4:	35
REVIEWER 5:	48
REVIEWER 6:	55

In our letter, the reviewer's comments are shown in *blue italics*, all the responses are shown in black Arial. We have inserted some of the revised and newly added figures with their legends, and the corresponding revised text excerpts are provided (**section name, line numbers**) from the clean version of the manuscript for ease of accessibility.

Also, in the revised version of the manuscript, the removed parts from the original paper are shown in double strikethrough and all the changes and additions are highlighted in blue Times New Roman.

Detailed response to the editors' comment as following:

EDITORS' Comment:

"In particular, we would expect your revision to address Reviewer #1's concern on the lack of sample size and treatment response data, Reviewer #3's concern on a lack of mechanistic details and controls to support the proposed crosstalk, Reviewer #4's concern on a lack of direct evidence to support the proposed direct involvement of SELENOP+ macrophages in activating CD8+ T cells, and Reviewer #6's concerns on the relevance of Tregs on HGSOC progression, discrepancy on the SELENOP+ macrophage abundance inferred from the two datasets, and the potential sample composition effect on the BRCA analysis. That is not to say that we consider any of the remaining reviewer concerns to be any less important, and we would expect your revision to address them in full."

Response and revision in the manuscript:

First, we again thank the editors for giving us the option of responding to reviewer comments and revising the manuscript for *Nature Communications*. It has taken us nearly 6 months to complete the studies owing to the technically demanding and time-consuming nature of the *in vivo* experiments. We thank the editorial team for their patience.

(1) To address concerns on the lack of sample size and treatment response data raised by Reviewers #1, we expand sample size by integrating our single-cell RNA sequencing (scRNA-seq) data with a publicly available dataset GSE184880, and the results regarding macrophages remained consistent with our original observations. We also analyzed the GSE266577, which included paired samples from chemo-naive and post-neoadjuvant chemotherapy (interval debulking surgery, IDS) patients. The results revealed a reduced interaction potential between *SELENOP*⁺ macrophages and *GZMH*⁺ precursor exhausted CD8+ T cells following neoadjuvant chemotherapy (NACT).

(2) In response to Reviewer #3, we conducted a comprehensive set of additional experiments, including SELENOP antibody validation assay, Western blotting of GPX1 in CD8+ T cells, ELISA for SELENOP, selenite pretreatment assay, macrophage-derived CM and CD8+ T cell coculture system, and CRISPR-Cas9-mediated EPHB2 deficiency, serum SELENOP concentrations in orthotopic ovarian cancer mice, adoptive cellular transfer assay *in vivo*, along with several other assays suggested by Reviewer #3. These experiments have refined mechanistic details and controls to support the proposed crosstalk.

(3) To address Reviewer #4's comments, we performed targeted experiments that directly support the role of *SELENOP*⁺ macrophages in CD8+ T cells activation, including antigen presentation assay *in vitro* and adoptive cellular transfer assay *in vivo*.

(4) In response to Reviewer #6, we analyzed CD4+ T cell heterogeneity using our scRNA-seq data and identified a potential involvement of regulatory T cells (Tregs) in the metastatic progression of HGSOC. We explained the discrepancy in *SELENOP*⁺ macrophage abundance between different detection methods. This methodological

difference would not affect data explanation. In addition, we incorporated two previously unassigned LAT samples, classified as *BRCA* wild-type based on newly generated whole-exome sequencing (WES) results, into the subsequent analysis. The results remained consistent with our original observations.

(5) Furthermore, as recommended by Reviewers #2 and #5, we clarified methodological details and improved figure quality and presentation throughout the manuscript. To strengthen the reported data and conclusions, we now include the relevant statistical information in figures and their legends, and have generated a new Supplementary Table (**Source Data**) detailing every result for statistical analysis.

We believe that the revisions made in response have adequately addressed the reviewers' concerns, substantially improving the quality of the manuscript. We are confident that the revised manuscript offers valuable insights into HGSOC, with implications for future research, the development of effective therapeutic strategies, and potential translational applications.

POINT-BY-POINT RESPONSE:

REVIEWER 1:

"The study of Liu et al., provides a comprehensive analysis of the tumor microenvironment (TME) in high-grade serous ovarian cancer (HGSOC) metastasis using advanced techniques like single-cell RNA sequencing and spatial transcriptomics. The authors identified critical cellular and molecular mechanisms influencing tumor progression and immune suppression. The SELENOP⁺ macrophages were found to enhance antitumor immunity by activating CD8⁺ T cells, whereas SPP1⁺ macrophages promoted tumor metastasis. Hypoxia-driven malignant epithelial cells were shown to reprogram SELENOP⁺ macrophages through VEGFA-EPHB2 signaling, diminishing the immune response. This interaction was associated with HGSOC metastasis. Then, anti-VEGFA therapy increased the proportion of SELENOP⁺ macrophages and CD8⁺ T cells, highlighting its potential for improving therapeutic outcomes. The study also revealed that BRCA-mutated HGSOC tumors exhibit a more active immune response compared to BRCA wild-type, suggesting the need for targeted immunotherapeutic strategies.

Key findings of the study include:

- Identification of two distinct macrophage subsets: SELENOP⁺ macrophages (M06) with antigen-processing and presentation capabilities, and SPP1⁺ macrophages (M09) involved in extracellular matrix remodeling. Also, M06 macrophages are associated with better patient outcomes, whereas M09 macrophages are linked to poor outcomes.*
- Demonstration that SELENOP⁺ macrophages possess immunostimulatory properties, potentially enhancing the cytotoxicity of CD8⁺ T cells via SeP. This is in contrast with pro-tumorigenic effects of SPP1⁺ macrophages.*
- SELENOP⁺ macrophages are enriched in adnexal tumors, but their presence decreases as the tumor metastasizes.*
- Identification of seven consensus metaprograms (MPs) within malignant epithelial cells, each with distinct gene signatures*
- Hypoxia-driven malignant epithelial cells reprogram SELENOP⁺ macrophages via VEGFA-EPHB2, as well as Hypoxia and FTL promote metastasis and therapy resistance. The study shows that VEGFA inhibition increases SELENOP⁺ macrophage proportions in primary tumors and peritoneal metastasis.*
- The study also characterizes the heterogeneity of CD8⁺ T cells and their functional states in different anatomical sites.*

Although the study is nicely presented as such, several major concerns should be addressed before the publication:"

General response: We thank the reviewer for the constructive summary of our study. We appreciate that the key findings and overall significance of the work were acknowledged, which was noted as "*nicely presented*". Below, we provide a point-by-point response to the reviewer's concerns.

“1) *Limited Sample Size: The authors acknowledge that the study may be limited by a relatively small sample size, especially in some groups such as EAT. This may reduce the significance of the findings. We suggest to increase the sample size, at least for profiling the macrophage subsets.*”

Response and revision in the manuscript: We thank the reviewer for this insightful comment. We integrated our scRNA-seq dataset with GSE184880, resulting in a combined dataset including seven EAT, eight LAT and nine Met samples. By analyzing the inferred macrophage subpopulations in the integrated dataset, we confirmed and further substantiated our original observations: a progressive decrease in *SELENOP*⁺ macrophages and a concomitant increase in *SPP1*⁺ macrophages from EAT to LAT and ultimately to Met (**Extended Data Fig. 4d**) (**Results, lines 295-299**).

Extended Data Fig. 4d. Tissue preference of each macrophage clusters estimated by R_{o/e} (integrated GSE184880 with our scRNA-seq data) ($P < 2.2e-16$). P -values: chi-squared test. $n = 24$ samples, $n = 7$ EAT, $n = 8$ LAT, $n = 9$ Met.

“2) *Lack of Treatment Response Data: While the study provides an in-depth analysis of the TME, it does not incorporate the effects of chemotherapy or other treatments on the TME or link molecular findings to clinical outcomes like platinum-free interval (PFI). All the patient samples investigated by the authors are from treatment-naive cohorts. I suggest that the authors should investigate:*

- *what is the effect of chemotherapy on macrophages distribution and characterization in HGSOc?*
- *how does the chemotherapy affect the functionality of M06 and M09 populations, in terms of immunostimulatory properties?”*

Response and revision in the manuscript: We sincerely thank the reviewer for these thoughtful and important suggestions. We have expanded our analysis using GSE266577 including chemo-naive and respective IDS patients following NACT. Our new findings are summarized as below:

(1) The abundance of macrophages within the myeloid cells decreased following NACT, accompanied a shift from anti-inflammatory to pro-inflammatory states (**Extended Data**

Fig. 8a-c). Inferred *SELENOP*⁺ macrophages showed reduced *SELENOP* expression and lower antigen processing and presentation signature scores, while *SPP1*⁺ macrophages exhibited decreased matrix remodeling signature scores following NACT, with no significant enrichment between the two groups (**Extended Data Fig. 8d-g**) (**Results, lines 491-498**).

Extended Data Fig. 8a-g. **a**, Tissue preference of myeloid clusters estimated by R_{oe} between chemo-naive and post-neoadjuvant chemotherapy (interval debulking surgery, IDS) groups in GSE266577 ($P < 2.2e-16$). Violin plots comparing the proinflammatory (**b**) and anti-inflammatory function scores (**c**) between groups. **d**, Tissue preference of inferred *SELENOP*⁺ and *SPP1*⁺ macrophages estimated by R_{oe} between chemo-naive and IDS groups in GSE266577. Violin plots comparing the *SELENOP* expression (**e**), antigen processing and presentation of *SELENOP*⁺ macrophages (**f**) and matrix remodeling signature scores of *SPP1*⁺ macrophages (**g**) between groups. For **a-g**, a total of $n = 18$ HGSOc paired samples, including $n = 9$ chemo-naive, $n = 9$ IDS. For **a, d**, P -values: chi-squared test. For **b, c, e-g**, box, median \pm interquartile range. P -values: two-sided Wilcoxon test.

(2) Clinically, in our scRNA-seq cohort, *SELENOP*⁺ macrophages were enriched in non-relapsed patients, while *SPP1*⁺ macrophages predominated in relapsed ones. In GSE266577, patients with long PFIs exhibited enrichment of *SELENOP*⁺ macrophages along with higher *SELENOP* expression, whereas the proportion of *SPP1*⁺ macrophages was more abundant in short PFI group, with higher matrix remodeling signature scores (**Extended Data Fig. 8j-m**). The finding related to *SELENOP*⁺ macrophages was validated by mIHC in an independent cohort (**Extended Data Fig. 8n and Supplementary Table 7**). These results expanded our knowledge on the impact of chemotherapy on macrophages in HGSOc (**Results, lines 509-516**).

Extended Data Fig. 8j-n. **j**, Tissue preference of *SELENOP*⁺ and *SPP1*⁺ macrophages estimated by $R_{o/e}$ between groups in our scRNA-seq dataset ($P < 2.2e-16$). **k**, Tissue preference of inferred *SELENOP*⁺ and *SPP1*⁺ macrophages estimated by $R_{o/e}$ between short and long platinum free interval (PFI) groups in GSE266577 ($P < 2.2e-16$). Violin plots comparing the *SELENOP* expression of *SELENOP*⁺ macrophages (**l**), and matrix remodeling signature scores of *SPP1*⁺ macrophages (**m**) between groups. **n**, Representative image of ovarian tumor stained by multiplex immunohistochemistry (left), scale bar, 100 μ m and the quantification plots (right). Data represent the mean \pm SD. Two-sided unpaired Student's t test, $n = 10$. * $P < 0.05$. For **j**, $n = 7$ solid sites samples in non-replased group, $n = 8$ solid sites samples in replsed group. For **k-m**, a total of $n = 22$ HGSOC samples, including $n = 6$ long PFI, $n = 16$ short PFI. For **k**, P -values: chi-squared test. For **l, m**, box, median \pm interquartile range. P -values: two-sided Wilcoxon test.

“3) The authors demonstrate that hypoxia-driven malignant epithelial cells reprogram M06 macrophages, and for this they did experiment using only OVCAR3 cell-derived conditioned media. OVCAR3 is not the best representative of HGSOC cell lines. I suggest the authors should use at least 2 more cell lines, and choose some BRCA1 positive and some BRCA1 negative (See Domcke et al, 2013, <https://www.nature.com/articles/ncomms3126>).”

Response and revision in the manuscript: We thank the reviewer for this important and constructive suggestion. We extended our *in vitro* experiments to include two additional HGSOC cell lines besides OVCAR3 (*BRCA1* wild-type) with distinct *BRCA1* statuses: CAOV3 (*BRCA1* wild-type) and COV362 (*BRCA1* mutant)¹. Different hypoxic culture durations were tested in these cell lines (24 h, 48 h and 72 h) (**Fig. 6e**). Conditioned media (CM) derived from both CAOV3 and COV362 recapitulated the reprogramming effects on THP-1 cells as OVCAR3, characterized by reduced *SELENOP* expression cocultured with the hypoxic CM, which were reversed by VEGFA neutralization. Parallel experiments using hypoxia-CM from ID8 and BMDMs yielded consistent results (**Fig. 6f**). These findings support the robustness and consistency of our observations across multiple genetically diverse ovarian cancer models (**Results, lines 435-443**).

Fig. 6e,f. **e**, Western blot images and quantifications of VEGFA protein expression in cell lines under normoxic or hypoxic conditions for 24 h, 48 h, and 72 h. Two-sided unpaired Student's t test. **f**, SELENOP expression in THP-1 cells or BMDMs after cultured with conditioned medium (CM) from cell lines under normoxic or hypoxic conditions for 48 h or 72 h, with or without VEGFA antibody (VEGFA Ab). One-way ANOVA with Bonferroni post hoc test. Data represent the mean \pm SD. $n = 3$. ns: not significant, $*1D < 0.05$, $**1D < 0.01$, $***1D < 0.001$.

“4) The study does not focus on the detailed interaction of CD8⁺ T cells with myeloid cells, particularly after chemotherapy, as is found in recent study (<https://doi.org/10.1016/j.ccell.2024.11.005>). The study also lacks focus on Myelons. The authors should address these in treatment-naive and post-chemotherapy.”

Response and revision in the manuscript: We thank the reviewer for this highly insightful comment.

(1) We performed the intercellular communication analysis on the dataset GSE266577. Our findings reveal a reduction in the interaction potential between *SELENOP*⁺ macrophages and *GZMH*⁺ CD8⁺ T cells following NACT (**Extended Data Fig. 8h**). Further analysis using t-CyCIF-guided GeoMX data indicate no changes in spatial proximity between these two populations (**Extended Data Fig. 8i**), suggesting a disruption of the immunoregulatory axis between *SELENOP*⁺ macrophages and *GZMH*⁺ CD8⁺ T cells following NACT, which may be attributable, at least in part, to a reduction in their functional interaction potential rather than the changes in spatial proximity (**Results, lines 498-508**).

Extended Data Fig. 8h, i. **h**, Heatmaps illustrating the cell-cell interaction patterns in chemo-naive (upper) and IDS samples (lower). **i**, Boxplots showing the coexistence score in stromal (upper) or tumor (lower) areas of interest of *SELENOP*⁺ macrophages and *GZMH*⁺ precursor exhausted CD8⁺ T cells for between chemo-naive and IDS groups. For **h**, total $n = 46$, including $n = 26$ chemo-naive, $n = 20$ IDS. For **i** (upper), total $n = 52$, including $n = 27$ Chemo-naive, $n = 25$ IDS. For **i** (lower), total $n = 52$, including $n = 26$ Chemo-naive, $n = 26$ IDS. For **i**, box, median \pm interquartile range. P -values: two-sided Wilcoxon test.

(2) Myelonets are spatial structures that form interconnected myeloid networks². As suggested, we collected ten paired omental metastatic lesions samples from HGSOC patients of chemo-naïve and IDS to analyze the Myelonets involved in *SELENOP*⁺ macrophage. The results from mIHC showed no change of the proportion of *SELENOP*⁺ macrophages clustered together in 100 μ m circle in total macrophages, suggesting no detectable effect of chemotherapy on the Myelonets consist of *SELENOP*⁺ macrophage (**Supporting Fig. 1a-c**).

Supporting Fig. 1. Representative examples of ovarian tumor stained by multiplex immunohistochemistry in chemo-naive group (a) and in IDS group (b), scale bar, 100 μ m and the quantification plots (c). Two-sided paired Student test, n =10. ns: not significant.

“5) Address the relevance of treatment impact on macrophage subsets in the discussion as well. It will be interesting to see whether chemotherapy positively or negatively influences M06 or M09 enrichment and their functional roles.”

Response and revision in the manuscript: We thank the reviewer for this valuable suggestion. In the revised manuscript, we have added relevant content in the **Discussion** section (**lines 598-612**).

REVIEWER 2:

“Previous work from this group showed that CD8⁺ T cells were tumour-specific in adnexal tumours but functioned as bystanders in HGSO17 metastasis, which involved paired scRNA seq and scTCR-seq analyses. The intention of this manuscript is to explore the dynamic changes in CD8⁺ T cells during HGSO17 metastasis, i.e., alteration to functional and differentiation states. The manuscript is a complicated paper full of bioinformatic data analysis (densely populated figures that are enriched with complex data). The information shown supports the narrative of the paper, showing intratumoral heterogeneity and complex epigenetic changes that causes remodelling of the tumour microenvironment in HGSO17. The finding that hypoxia-driven malignant epithelial cells could reprogram SELENOP⁺ macrophages is backed up with additional data. The addition of the mouse work adds weight to this study and a mechanism of tumour progression/survival/reduced immune surveillance, showing that VEGFA signalling is a key component and highlights that anti-angiogenic therapies could be used as a treatment these tumours. This work is an important contribution to our understanding of HGSO17 and is a resource for future work.”

General response: We thank the reviewer for the detailed and thoughtful evaluation of our manuscript as *“an important contribution to our understanding of HGSO17 and is a resource for future work”*. We appreciate the recognition of the study's contribution to understanding the tumor microenvironment and immune dynamics in HGSO17, as well as the potential translational relevance of our findings.

“1) Last paragraph of the introduction is very long and acts as an abstract of the full manuscript. This is inappropriate, as the introduction should rather summarise the intention/rationale of the study and study approach.”

Response and revision in manuscript: We thank the reviewer for the helpful suggestion. In response, we have revised the final paragraph of the **Introduction (lines 78-93)**.

“2) Should consider moving some methods into supplementary to reduce the length of the manuscript.”

Response and revision in the manuscript: Thank you for this valuable suggestion. We have moved the methods related to Spatial transcriptomic experiment, WES, Flow cytometry, Western blot and RT-qPCR to the **Supplementary Methods** section in the **Supplementary Materials**.

“3) Figure 2B, the dark colour of the extreme -1 and +1 are heavily blackened, which results in not knowing what end of the colour the T06_CD8T-CXCL13 cell population is for the exhausted and TEX boxes, either very dark/blackened red (+1) or very dark/blackened blue (-1). I suggest taking out the black shading from these boxes. While not as bad, same could be said for Figure 2A with several dark blue spots that could be

misinterpreted in the figure as being over-expressed (especially as the spots are very small). I suggest to make all the dots larger, as there is room in the figure for this. Same can be said for Figure 3A and 3D.”

Response and revision in the manuscript: We appreciate the reviewer’s helpful feedback. In response, we have revised the figures accordingly.

Fig. 2a,b. **a**, UMAP plots depicting the clusters of CD8⁺ T cells, colored by cluster. Conventional and unconventional CD8⁺ T cells are highlighted with black dotted lines, respectively (left). Dot plot showing marker genes of conventional CD8⁺ T clusters (right). Dot size representing percent of expressing cells in each cluster and color represents z-score of normalized mean expression level of selected genes. **b**, Heatmap indicating the expression levels of gene signatures of previously reported Texp and Tpep cell states, and exhaustion signature in conventional CD8⁺ T subtypes. For **a**, **b**, n = all 34 scRNA-seq cohort samples.

Fig. 3a. **a**, UMAP projection of 13 myeloid clusters colored by clusters (left). Dot plot showing expression patterns of selected genes across indicated clusters. Dot size represents percent of expressing cells in each cluster and color represents z-score of normalized mean expression level of selected genes (right). For **a**, n = all 34 scRNA-seq cohort samples.

d

Fig. 3d. Enriched pathways of each macrophage subset in KEGG databases. n = all 34 scRNA-seq cohort samples.

“4) Line 178: ‘Subcluster T06_CD8T-CXCL13, defined by the expression of CXCL13, CTLA4, and TIGIT, exhibited a transcriptional signature indicative of exhaustion, thus classified as exhausted T cells (Tex) (Fig. 2a,b).’ Please add a sentence to explain why this expression signature is indicative of exhaustion and reference a paper if possible to help readers that might not be as knowledgeable.”

Response and revision in the manuscript: We sincerely thank the reviewer for the careful and insightful comments on our manuscript. We have added a brief explanation in the revised manuscript (**Results, lines 136-138**).

“5) Figure 2D, in annotation ‘expansion index’ is spelt wrong. In this figure there should also be axis annotations that have been omitted.”

Response and revision in the manuscript: We thank the reviewer for pointing this out. We updated Figure 2D (now **Extended Data Fig. 2d**) in the revised manuscript accordingly.

Extended Data Fig. 2d. Clonal expansion of the clusters of conventional CD8⁺ T cells quantified by STARTRAC indices (PRJCA005422). Each dot represents a patient. Box, median ± interquartile range. n = 14 HGSOC samples. P-values are calculated by Kruskal–Wallis test.

“6) Figure 2F: in result text, lack of annotation in the figure and a different description in the figure legend, i.e., what is ‘ $R_{o/e}$ ’? Currently, it is ambiguous to what the data represents. Can the authors improve the labelling of the figure and description in the text and figure legend to make this result clearer.”

Response and revision in the manuscript: We sincerely appreciate the reviewer’s careful evaluation of Figure 2f.

(1) In our analysis, the ratio of observed to expected cell numbers ($R_{o/e}$) was calculated using the formula: $R_{o/e} = \text{Observed} / \text{Expected}$, as described by Zhang *et al*³ and Chen *et al*⁴, to assess tissue preference of specific cell clusters. A cluster was deemed enriched in a tissue if its $R_{o/e}$ ratio exceeded 1. We have included a detailed description in the **Methods** section to ensure clarity for readers unfamiliar with this metric (**Methods, lines 736-744**).

(2) We have provided a clear explanation of $R_{o/e}$ at its first mention (**Supplementary Note 1, lines 13-17**) and in the figure legend of **Extended Data Figure. 2c** (Figure. 2f in the original manuscript).

Extended Data Fig. 2c. Tissue preference of each CD8⁺ T cluster, estimated by $R_{o/e}$ (the ratio of observed to expected cell numbers) ($P < 2.2e-16$, left) ($P = 3.512e-15$, right). P -values are calculated by chi-squared test.

“7) Line 595: Figure 6h, siEPHB is not effective at knocking down siEPHB. I would suggest using other siRNA that are more effective and employ EPHB drug inhibitors to add validity to this work.”

Response and revision in the manuscript: We thank the reviewer for this valuable and constructive comment. We employed the CRISPR-Cas9 system and successfully generated *EPHB2*-deficient THP-1 cells (**Supplementary Fig. 2c**) (**Methods, lines 935-945**). Treatment of THP-1 cells with exogenous recombinant VEGFA protein for 48 h resulted in a decrease in SELENOP expression and secretion, which could be reversed by knockdown of *EPHB2* expression, further supporting that VEGFA-mediated

suppression of SELENOP is largely dependent on EPHB2 signaling (**Fig. 6g,h**) (**Results, lines 443-445**).

Supplementary Fig. 2c. Western blot analysis confirmed CRISPR/Cas9-mediated EPHB2 deletion in THP-1 cells. sgNC: empty vector for negative control of sgRNA. Data represent the mean \pm SD. Two-sided unpaired Student's t test for two groups, $n = 3$. * $P < 0.05$.

Fig. 6g, h. Western blot images and quantifications of SELENOP protein expression after knocking down *EPHB2* by CRISPR-Cas9 in THP-1 cells (**g**) with 50 ng/mL rhVEGAF treated for 48h. sgNC, empty vector for negative control of sgRNA, rh VEGAF: recombinant human VEGFA protein. **h**, ELISA showing SELENOP level in the culture supernatants in **g**. For **g, h**, one-way ANOVA with Bonferroni post hoc test, $n = 3$. ns: not significant, * $P < 0.05$, *** $P < 0.001$.

“8) Line 1191: Amplicon lengths should be provided here, and whether dissociation curves were also run routinely to ensure specificity of reactions.”

Response and revision in the manuscript: We thank the reviewer for this helpful suggestion. In the revised manuscript, we have added the amplicon lengths and clarified that dissociation curves were routinely performed to ensure the specificity of the reactions in the **Supplementary Materials (Supplementary Methods, line 278-290)**. We also determined the amplified FTL (left) and ACTB (right) on 1.5% TBE agarose gels (**Supplementary Fig. 8g**). Comparative dissociation curves are presented in **Supporting Figure 2a** below. In addition, the RT-qPCR products were sequenced and verified by BLAST analysis to further confirm the specificity of amplification (**Supporting Figure 2b**).

Extended Data Fig. 8g. Size and purity determination of amplified *FTL* (upper) and *ACTB* (lower) on 1.5% TBE agarose gels. Amplified products of RT-qPCR from COV362 cell line cultured under normoxic (Line 1, 3, 5) and hypoxic (Line 2, 4, 6) conditions.

Supporting Fig. 2. Evaluation of the specificity of real-time RT-qPCR in quantifying FTL expression. (a) Comparative dissociation curves of amplified *FTL* (left) and *ACTB* (right). (b) BLAST analysis for *FTL* (left) and *ACTB* (right) sequence.

REVIEWER 3:

“This study reveals a novel cross-talk between macrophages, T cells and epithelial cells during the progression of high-grade serous ovarian cancer which centres around SELENOP⁺ macrophages. This cell population was identified by an extensive single cell RNAseq approach considering both the temporal aspect of tumor development as well as different anatomical sides. While this first screening approach is very convincing and well justified to identify the relevant cellular players the mechanistic part analysed by cell culture and mouse experiments remains rather unclear and needs to be substantially improved and extended to support the conclusions drawn by the authors.

The identification of the proposed cross-talk within the tumor microenvironment might not only be relevant for ovarian cancer but could extend to additional cancer types given that this SELENOP⁺ macrophage population has also been identified in lung cancer before. Therefore, the manuscript would be of great interest to the broad readership of Nature Communications. However, in my view, the manuscript still lacks mechanistic details and controls and should be thoroughly revised to address the following comments.”

General response: We sincerely thank the reviewer for the thoughtful and encouraging comments that underscore the significance and broader relevance of our findings as *“of great interest to the broad readership of Nature Communications”*. We fully agree that the mechanistic aspects of the study required further investigation. To address this, the mechanistic analyses have been substantially improved and extended through additional *in vitro* and *in vivo* experiments to support the proposed cross-talk among SELENOP⁺ macrophages, T cells, and epithelial cells during HGSOC progression.

“The whole manuscript should be more clearly structured according to the information provided in the abstract. So far, it is very complicated for the reader to have data provided as figures, extended data figures, and supplementary figure. In my view, a fundamental restructuring of the data is really needed to support the story described in the abstract. Thus, later sections e.g. about immune checkpoints or the section starting at line 436 about dendritic cells are not really related to this story and should be excluded. Also the comparison between different anatomical sides within one patient could be left out. At least it is not getting clear how this relates to the main story line.”

Response and revision in the manuscript: We sincerely thank the reviewer for the constructive comments regarding the manuscript structure. As suggested, they have been removed.

“Based on the abstract, the title could also more specifically describe the novel interactions identified.”

Response and revision in the manuscript: We appreciate the reviewer’s thoughtful suggestion regarding the title. In response, we have revised the title. The new title is “Hypoxia-Driven Remodeling of SELENOP⁺ Macrophages Shapes T Cell Dynamics and

Promotes Ovarian Cancer Metastasis". We hope this revised title more accurately captures the scope and significance of our study.

"Throughout the manuscript: Please introduce the genes/proteins that were discussed in more detail. This is mandatory especially when addressing a broader readership."

Response and revision in the manuscript: We thank the reviewer for this important comment. Accordingly, we have expanded the introduction of SELENOP⁵⁻⁷ (**Results, lines 245-247 and Discussion, lines 555-566**) and VEGFA-EPHB2⁸ (**Results, lines 431-435**).

"Introduction, line 63: Please elaborate on the results gained in the previous publication (Ref 17) a bit more. Which questions remained unanswered after this publication? Why did you now follow up an alternative approach?"

Response and revision in the manuscript: We thank the reviewer for this helpful comment. We have added relevant content in the **Introduction** section (**lines 69-93**).

"Introduction, lines 72-94: This section intensively summarizes the generated results but I miss a clear aim of the study."

Response and revision in the manuscript: We thank the reviewer for pointing this out. We have clarified the aim of our study at this section to better guide the reader through the subsequent summary of our findings (**Introduction, lines 78-93**).

"Results, line 114: The authors conclude that the anatomical site was the primary driver of heterogeneity. How does this relate to the following results which mainly take the tumor progression / stage into account? Here it would have been also helpful for my understanding to comment on the dependency of samples because different anatomical samples are taken from one patient (dependent variables) while different stages are analysed in independent samples from different patients."

Response and revision in manuscript: We appreciate the reviewer's insightful comment. (1) Ovarian cancer (OC) patients are staged based on the locations of different peritoneal metastasis sites according to the FIGO staging system⁹. Studies have shown that different peritoneal metastasis sites are closely related to ovarian cancer staging, treatment resistance, and even prognosis¹⁰⁻¹². Thus, metastasis sites are indicators of progression/stage. We have added this biological background in the revised manuscript (**Introduction, lines 61-66**).

(2) We would like to clarify that the original statement "*anatomical site was the primary driver of heterogeneity*" referred specifically to the distinction between solid and fluid sites-derived samples, rather than to differences among anatomical locations such as adnexal

tumors, peritoneal metastases, or omental lesions. We agree that the phrasing could be misleading, and in the revised manuscript, we have updated the statement for greater clarity as follows:

“To assess global transcriptomic variation across all samples from scRNA-seq cohort, we performed Principal Component Analysis (PCA). The results revealed that the primary source of heterogeneity in the HGSOC tumor microenvironment was the distinction between solid and fluid sites, as samples from these two compartments were clearly separated.” We believe this revised wording more accurately reflects our findings and avoids potential misinterpretation (**Results, lines 109-114**).

(3) In addition, we would like to clarify that the samples included in the PCA were obtained from different individuals rather than from multiple sites in a single patient. We have updated the PCA plot by labeling each data point with the corresponding patient ID (**Fig. 1b**), demonstrating that the observed clustering is not primarily driven by inter-patient variation, but instead reflects site-associated transcriptional differences.

Fig. 1b. The Principal Component Analysis (PCA) plot showing the PC1 vs PC2 projection of all samples at the pseudo-bulk RNA-seq level. For **b**, $n =$ all 34 scRNA-seq cohort samples.

“When looking at progression, the comparison between tumor and fluids is not conclusive. Why was this incorporated into the analyses? Effects of anatomical localization and tumor progression should be more clearly separated from each other (or even leave out anatomical localization results).”

Response and revision in manuscript: We sincerely thank the reviewer for this valuable comment.

(1) Unlike other solid tumors, OC is marked by extensive transcoelomic dissemination and the accumulation of ascites^{13,14}. Peritoneal fluids in patients with OC reflect the FIGO stage IC3, serving as an essential part of the tumor ecosystem⁹. We have added this biological background in the revised manuscript to facilitate readers’ interpretation (**Introduction, lines 61-66**).

(2) We have removed the anatomical localization results (i.e., the comparison between the Met.Per and Met.Ome groups) from the revised manuscript to maintain a clearer and more focused narrative.

“Figure 1 and 2 extensively characterize the samples analysed. Those results could be narrowed down a bit to provide a clearer focus.”

Response and revision in the manuscript: We appreciate the reviewer’s constructive suggestion. In response, we have removed **Fig. 1f** (right), **Extended Data Fig. 1c**, **Extended Data Fig. 2c** in the original manuscript. In addition, **Figure. 2f** has been relocated to **Extended Data Fig. 2c** in the revised manuscript.

“Figure 3: the SELENOP⁺ macrophage cluster was identified. It is very important to keep in mind that this is only gene expression data which does not necessarily relate to protein expression because selenoprotein synthesis is strictly dependent on selenium availability which, under conditions with limited supply, overwrites a transcriptional regulation.”

Response and revision in the manuscript: We thank the reviewer for this important comment. We confirmed SELENOP expression at protein level in macrophages using multiple approaches, such as Western blot, ELISA and flow cytometry analyses, which are specified in the response to *“For all cell culture experiments,...pretreatment for 72 h.”*

“Please provide data showing the specificity of the SELENOP immunohistochemistry staining and Western Blot results (e.g. by staining cells cultured under conditions with and without selenium or by using specific blocking peptides). Unfortunately, there are many unspecific SELENOP antibodies available.”

Response and revision in the manuscript: We thank the reviewer for this important comment. SELENOP is a glycoprotein that typically appears as bands around ~55–60 k Da or at approximately 46 and 52 k Da in Western blot analyses, depending on the tissue type¹⁵. We tested three commercially available anti-SELENOP antibodies (**Supporting Table 1**) by Western blot using lysates from THP-1 cells, THP-1 cells cultured with selenium supplementation, and HEK293 cells (used as a negative control, as previously described¹⁵), respectively. Among these, Antibody 3 demonstrated superior specificity (**Supporting Fig. 3a**). We also assessed antibody 1 and 3 for IHC, via staining human normal gastric tissue adjacent to gastric cancer and normal lymphoid tissue, which are known to express high and low levels of SELENOP, respectively, according to the Human Protein Atlas (HPA). Antibody 3 also showed superior specificity (**Supporting Fig. 3b**). Thus, Antibody 3 was selected for all subsequent experiments to ensure the reliable detection of SELENOP protein expression throughout this study.

Supporting Table 1. Three antibodies for specificity validation.

ID	Company	Catalog number	Applications
Antibody 1	Santa Cruz	sc-376858	WB, IP, IF, IHC, ELISA
Antibody 2	abcam	ab277526	ICC/IF, Flow Cyt, WB, IP
Antibody 3	Invitrogen	PA5-112707	WB, IHC, ELISA

Supporting Fig. 3. Validation of SELENOP Antibodies Specificity. (a) Western blot: the first and last lanes show the molecular weight marker (red line indicates 70 k Da). The second lane: HEK293 cells. The third and fourth lanes contain protein lysates from THP-1 cells and THP-1 cells cultured with selenium supplementation, respectively. (b) Immunohistochemistry staining: SELENOP expression in representative normal human gastric tissue (left panel) and normal human lymphoid tissue (right panel), scale bar, 100 μm. Two-way ANOVA test.

“As only very shortly stated in line 334-336, SELENOP is secreted by hepatocytes but also other cell types (including macrophages?) and can be taken up by other cells via the receptor LRP8. SELENOP is broken down to release selenium which is used for the synthesis of other selenoproteins such as GPX or TXNRD. Those parameters definitely need to be analysed to show that SELENOP is taken up by T cells after culturing them with recombinant SELENOP. I would recommend to analyse GPX1 protein expression but other selenoproteins will work as well. Alternatively, the intracellular selenium concentration could be measured.”

Response and revision in the manuscript: We thank the reviewer for this insightful and constructive suggestion. In response, we examined GPX1 expression in CD8⁺ T cells after culturing them with recombinant SELENOP. Our results showed that treatment with 200 ng/mL recombinant SELENOP significantly increased the protein levels of GPX1 in OT-1 T cells (**Extended Data Fig. 3e**). This serves as mechanistic detail supporting the proposed cross-talk described in our study (**Results, lines 254-256**).

quantifications of GPX1 expression in OT-1 cells following treated with rm SELENOP or an equivalent volume of PBS as control for 48 h. Data represent the mean ± SD. One-way ANOVA with Bonferroni post hoc test for multiple groups, n = 3. * $P < 0.05$.

“For all cell culture experiments, it is mandatory to know the basal selenium concentration in FCS and as a proof of concept, cells should be also treated +/- selenite. Using THP1 macrophages, the authors should show that macrophages indeed secrete SELENOP. To strengthen the results, the authors could perform this experiment also with monocytes isolated from patient buffy coats. The conditioned medium from THP1 or other monocytes/macrophages could be used in addition to recombinant SELENOP to treat T cells. Also treating THP1 cells with IFN γ should be performed in cells +/- selenite pretreatment for 72 h.”

Response and revision in manuscript: We appreciate the reviewer’s thoughtful and detailed suggestions.

(1) The basal selenium concentration in fetal calf serum (FCS), measured using a selenium ion detection kit (Cat#abs580188, absin), was 1.693 mmol/L (**Supporting Table 2 and Supporting Fig. 4**).

Supporting Table 2. Selenium concentration in fetal calf serum (FCS)

Samples	FCS (test 1)	FCS (test 2)	FCS (test 3)
Sample OD	0.261	0.279	0.311
Blank OD	0.255	0.255	0.255
Corrected OD	0.009	0.015	0.027
Diluted Concentration (mmol/L)	0.006	0.024	0.056
Original Concentration (mmol/L)	0.899	1.531	2.651
Average Concentration (mmol/L)	1.693		

Supporting Fig. 4. Standard Curve for calculation.

(2) While assessing SELENOP protein levels in THP-1 cells and macrophages differentiated from patient-derived human peripheral blood mononuclear cells (hPBMCs) (MDMs) cells using Western blot, we also performed ELISA to evaluate SELENOP secretion (**Methods, lines 978-986**). The results showed that macrophages do secrete SELENOP, with increased levels upon IFN γ stimulation (**Fig. 4k,i** and **Extended Data Fig. 4f,g**). These results demonstrate that macrophages are indeed capable of synthesizing and secreting SELENOP (**Results, lines 326-328**).

Fig. 4k, I, k, Western blot images (left) and quantification (right) of the levels of *SELENOP* proteins in THP-1 cells under control conditions or after treatment with 70 nM Se, 40 ng/mL IFN γ , or their combination. Se: Na₂Se₃. I, ELISA showing SELENOP level in the culture supernatants in k. For k

(right), **I**, data represent the mean \pm SD. One-way ANOVA with Bonferroni post hoc test, $n = 3$, $*P < 0.05$, $**P < 0.01$, $***P < 0.001$.

Extended Data Fig. 4f,g. Western blot analysis and quantification of SELENOP expression in MDMs (**f**) after treatment with PBS as control or IFN γ . **g**, ELISA showing SELENOP level in the corresponding culture supernatants in **f**. For **f** (right), **g**, data represent the mean \pm SD. For **f**, **g**, two-sided unpaired Student's t test, $n = 3$. $*P < 0.05$, $**P < 0.01$.

(3) Macrophage-derived conditioned medium (CM) and CD8 $^+$ T cell coculture system were established to assess the role of macrophage-derived SELENOP (**Fig. 3k**) (**Methods, lines 947-951**). CM from *SELENOP*-overexpressing BMDMs modulated via lentiviral transduction (**Methods, lines 935-945**), significantly enhanced the tumor cell killing capacity of OT-1 T cells (**Fig. 3l**), which was accompanied by increased GZMB and PRF1 expression (**Extended Data Fig 3f, g**). Conversely, *Selenop*-knockdown BMDMs exhibited the opposite effects, which could be reversed by exogenous supplementation of recombinant SELENOP (**Extended Data Fig. 3h-j**). Parallel experiments using CM from THP-1 cells overexpressed SELENOP and human CD8 $^+$ T cells of HGSOc patients yielded consistent results (**Extended Data Fig. 3k-m**). Indeed, we further performed antigen presentation assay by directly coculturing OVA-loaded BMDMs with OT-1 T cells (**Extended Data Fig. 3n**) (**Methods, lines 953-958**). The results were also consistent with those observed in the macrophage-CM and CD8 $^+$ T cell co-culture system, suggesting the direct priming capacity of *SELENOP* $^+$ macrophages on CD8 $^+$ T cells (**Extended Data Fig. 3o-t**) (**Results, lines 257-277**).

Fig. 3k,l. k, (3) Macrophage-derived conditioned medium (CM) and CD8⁺ T cell coculture system. I, The total apoptosis rate of ID8-OVA cells induced by corresponding CD8⁺ T cells. Data represent the mean \pm SD, two-sided unpaired Student's t test, n = 3, *****P* < 0.0001.

Extended Data Fig. 3f-t. The proportions of CD3⁺CD8⁺GZMB⁺, CD3⁺CD8⁺PRF1⁺ T cells of OT-1 T cells or human CD8⁺ T cells co-cultured with CM from BMDMs-puroNC and BMDMs-OE-Selenop (**f, g**); BMDMs-BSDNC, BMDMs-KD1-Selenop, BMDMs-KD2-Selenop, BMDMs-KD1-Selenop and BMDMs-KD2-Selenop supplemented with rmSELENOP (**h, i**); THP-1-puroNC and THP-1-OE-SELENOP (**k, l**), respectively, and the total apoptosis rate of ID8-OVA cells (**j**) or OVCAR3 (**m**) induced by corresponding CD8⁺ T cells. **n**, Antigen presentation assay. The proportions of CD3⁺CD8⁺GZMB⁺, CD3⁺CD8⁺PRF1⁺ T cells of OT-1 T cells cocultured directly with BMDMs-puroNC and BMDMs-OE-Selenop (**o, p**); BMDMs-BSDNC, BMDMs-KD1-Selenop, BMDMs-KD2-Selenop, BMDMs-KD1-

Selenop and BMDMs-KD2-Selenop supplemented with rmSELENOP (**r, s**), respectively, and the total apoptosis rate of ID8-OVA cells (**q, t**) induced by corresponding CD8⁺ T cells. BMDMs, bone marrow derived macrophages; BMDMs-puroNC, BMDMs transfected with negative control lentivirus; BMDMs-OE-Selenop, BMDMs overexpressing *Selenop* after lentiviral transfection. BMDMs-BSDNC, BMDMs transfected with negative control shRNA; BMDMs-KD1-Selenop, BMDMs with *Selenop* knockdown after transfected with sh*Selenop*-1. BMDMs-KD2-Selenop, BMDMs with *Selenop* knockdown after transfected with sh*Selenop*-2. THP-1-puroNC, THP-1 transfected with control negative lentivirus; THP-1-OE-SELENOP, THP-1 overexpressing *SELENOP* after lentiviral transfection. For **f-m, o-t**, data represent the mean \pm SD. One-way ANOVA with Bonferroni post hoc test for multiple groups, two-sided unpaired Student's t test for two groups, n = 3. ns, not significant, * $P < 0.05$, ** $P < 0.01$, *** $P < 0.001$, **** $P < 0.0001$.

(4) Compared to the untreated controls, pretreatment of THP-1 cells with sodium selenite for 72 h significantly upregulated SELENOP expression and secretion under both IFN γ -stimulated and unstimulated conditions, highlighting the dependence of SELENOP synthesis on selenium availability in macrophages (**Fig. 4k, l** presented in (2)) (**Methods, lines 960-976**) (**Results, lines 328-330**). This result also serves as a proof of concept, aligning with the reviewer's suggestion that "...cells should also be treated \pm selenite."

"Section on epithelial cells (starting at line 461): The rationale for the transition from immune cells to epithelial cells is not getting clear. This section should be shortened. For example, the link to ferritin is not supporting the story line. This is also not mentioned later in the discussion again."

Response and revision in manuscript: We appreciate the reviewer's insightful comment.

(1) We have added a paragraph emphasizing the importance of malignant cells in metastasis, based on the "seed and soil" theory (**Results, lines 345-349**).

(2) As suggested by the reviewer, we have shortened the content related to ferritin in the section on epithelial cells (**Results, lines 389-393**).

"Line 587: Briefly introduce the VEGFA-EPHB2 pathway."

Response and revision in the manuscript: We thank the reviewer for the suggestion. We have added a brief introduction to the VEGFA-EPHB2 pathway, highlighting its relevance in cell-cell communication (**Results, lines 431-435**).

"The knockdown efficiency of EPHB2 is quite poor. Is there an alternative way of inhibiting the pathway? A knockout would be more conclusive to show a dependency."

Response and revision in the manuscript: We appreciate the reviewer's thoughtful and constructive comments. We employed the CRISPR-Cas9 system and successfully generated *EPHB2*-deficient THP-1 cells (**Supplementary Fig. 2c**) (**Methods, lines 935-945**). Treatment of THP-1 cells with exogenous VEGFA protein for 48 h resulted in a decrease in SELENOP expression and secretion, which could be reversed by knockdown

of *EPHB2* expression, further supporting that VEGFA-mediated suppression of SELENOP is largely dependent on *EPHB2* signaling (**Fig. 6g,h**) (**Results, lines 443-445**).

Supplementary Fig. 2c. Western blot analysis confirmed CRISPR/Cas9-mediated *EPHB2* deletion in THP-1 cells. sgNC: empty vector for negative control of sgRNA. Data represent the mean \pm SD. Two-sided unpaired Student's t test for two groups, $n = 3$. * $P < 0.05$.

Fig. 6g, h. Western blot images and quantifications of SELENOP protein expression after knocking down *EPHB2* by CRISPR-Cas9 in THP-1 cells (**g**) with 50 ng/mL rhVEGFA treated for 48h. sgNC, empty vector for negative control of sgRNA, rh VEGFA: recombinant human VEGFA protein. **h**, ELISA showing SELENOP level in the culture supernatants in **g**. One-way ANOVA with Bonferroni post hoc test, $n = 3$. ns: not significant, * $P < 0.05$, *** $P < 0.001$.

*“Line 594: If the effect of VEGFA treatment should be reversed by the *EPHB2* knockdown, please combine the VEGFA treatment with the knockdown to draw this conclusion.”*

Response and revision in the manuscript: We thank the reviewer for the insightful suggestion. We performed experiments combining VEGFA treatment with *EPHB2* knockdown in **Fig. 6g, h** presented in the previous response.

“Using patient-derived organoids is an interesting alternative approach but as stated above this should be combined with primary human monocytes/macrophages and not again with the THP1 cell line.”

Response and revision in the manuscript: We thank the reviewer for this valuable suggestion. In response, we combined CM from patient-derived cancer organoid (PDOs) under normoxic or hypoxic conditions with MDMs. CM from hypoxia-treated PDOs for 96 h significantly suppressed SELENOP expression in MDMs (**Fig. 6k**), supporting the role of hypoxia-driven epithelial signals in modulating macrophage states at the individual patient level (**Results, lines 447-448**).

Fig. 6k. SELENOP protein expression in MDMs after cultured with CM from organoids (HGSOC-12) under normoxic or hypoxic conditions (96 h) for 48 h (**k**). Data represent the mean \pm SD. Two-sided unpaired Student's t test, $n = 3$. * $P < 0.05$.

“Line 613, the in vivo model: The SELENOP⁺ macrophages at this stage really express more SELENOP protein as this has been characterized by FACS, in contrast to the RNAseq data which only rely on mRNA data (which might be quite irrelevant for selenoprotein expression). Please use another term for these macrophages to make this difference clearer to the reader.”

Response and revision in the manuscript: We thank the reviewer for this important clarification. We have used the term “F4/80⁺CD11b⁺/SELENOP⁺ macrophages” to refer specifically to the macrophage population phenotypically defined by FACS (**Results, lines 460, 476, and 479**).

“In addition, the authors should provide information about the selenium concentration in the mouse chow. In addition, it would be very interesting to also measure SELENOP concentrations in the serum of mice to exclude a systemic effect of VEGF on SELENOP secreted from the liver.”

Response and revision in manuscript: We appreciate the reviewer’s insightful comments. (1) We have added information regarding the selenium concentration (0.13 mg/kg) in the mouse chow in the revised manuscript (**Supporting Figure 5**) (**Methods, lines 1029-1031**).

Supporting Fig. 5. a, Cover page of the test report for the mouse chow. **b**, The selenium concentration in the mouse chow determined by this test (upper) and English translation (lower).

(2) We also measured SELENOP concentrations in the serum of mice (**Methods, 978986**). The results showed that VEGFA inhibition did not significantly alter circulating SELENOP levels, thereby excluding a substantial systemic effect of VEGF on SELENOP secreted from the liver in this murine model (**Fig. 7I**) (**Results, lines 467-470**).

Fig. 7I. Blood serum SELENOP concentrations of mice treated with IgG or anti-VEGFA antibody. Data represent the mean \pm SD. *P*-values: two-sided unpaired Student's t test. ns, not significant.

“Please quantify primary tumor, omental metastasis and peritoneal metastasis formation and not only provide fluorescence images and FACS results.”

Response and revision in manuscript: We thank the reviewer for the helpful suggestion. In the revised manuscript, we have quantified the formation of primary tumors and peritoneal metastases (**Fig. 7d-g**) (**Results, lines 456-459**). Although we initially intended to evaluate omental metastases, no detectable lesions were observed in the omenta of the mice in this murine model.

Fig. 7d-g. Representative images showing characteristic (d) and quantifications (e) of peritoneal metastases (black arrows) in mice treated with IgG (upper), and the absence of peritoneal metastases in mice treated with anti-VEGFA antibody (lower). Representative images (f) and quantification of tumor volumes (g) of primary tumors of mice treated with IgG or anti-VEGFA antibody. Bars, 10 mm. For e, g, data represent the mean \pm SD. *P*-values: two-sided unpaired Student's *t* test. ns, not significant. **P* < 0.05, ***P* < 0.01.

“To really show that macrophage-derived SELENOP is needed to modulate the VEGF effect on tumor metastasis in vivo, the authors should use SELENOP knockout mice with a macrophage specific SELENOP knockout (SELENOPdeltaMye mice).”

Response and revision in the manuscript: We appreciate the reviewer's valuable suggestion.

(1) While we fully agree that a macrophage-specific *SELENOP* knockout mouse model would be an ideal tool for mechanistic studies, it poses inherent limitations, including long time required for model construction, a relatively low success rate and the potential for developmental compensations associated with germline knockouts. Upon thorough evaluation, we employed an established adoptive cellular transfer system, which has been widely used to investigate macrophage-specific functions in tumor immunity¹⁶⁻¹⁸, to address this concern, as illustrated in **Extended Data Fig. 7a** (**Methods, lines 1057-1066**).

(2) Then we found that, compared with the BMDMs group, the proportion of CD45.2+CD3+CD8+GZMB+ T cells decreased, accompanied by reduced frequencies of CD45.1+F4/80+CD11b+SELENOP+ macrophages in the primary tumors of the BMDMs-KD1-Selenop group. Meanwhile, VEGFA inhibition led to a concomitant reduction in CD45.2+CD3+CD8+GZMB+ T cells and CD45.1+F4/80+CD11b+SELENOP+ macrophages in the primary tumors of mice injected with BMDMs-KD1-Selenop cells, but had no significant effect in mice injected with BMDMs (**Extended Data Fig. 7b,c**). These findings support the role of SELENOP+ macrophages in priming CD8+ T cell cytotoxicity within the HGSOc tumor microenvironment *in vivo*, and further emphasize a VEGFA-mediated regulatory mechanism that appears to be more pronounced in macrophages with low SELENOP expression (**Results, lines 471-485**).

(3) In addition, together with the observed reduction in tumor metastasis in our another orthotopic models after VEGFA blockade for 35 days (**Fig 7d-g** as presented in the previous response), these findings suggest a potential requirement for macrophage-derived SELENOP in modulating the effects of VEGFA on tumor metastasis (**Results, lines 456-459**).

Extended Data Fig. 7a-c. **a**, Schematic diagram of the protocol for adoptive cellular transfer experiments. Fresh primary tumor tissues were digested and stained for flow cytometry analysis. Shown is the proportion of CD45.1⁺F4/80⁺CD11b⁺SELENOP⁺ cells (**b**) and CD45.2⁺CD3⁺CD8⁺GZMB⁺ cells (**c**) across different sites of each group. Pri. tumor: primary tumor of orthotopic ovarian cancer mouse model; Met.Per: peritoneal metastasis of orthotopic ovarian cancer mouse model. *P*-values are calculated by two-way ANOVA with Tukey post hoc test. *n* = 5 in BMDMs group, *n* = 5 in BMDMs-KD1-Selenop group, *n* = 4 in BMDMs combined with anti-VEGFA group, *n* = 5 in BMDMs-KD1-Selenop combined with anti-VEGFA group. ns: not significant, **P* < 0.05, ***P* < 0.01, *****P* < 0.0001.

“Line 621: The last section of the results is again out of the focus of the manuscript and should be clearly shortened.”

Response and revision in the manuscript: We appreciate the reviewer’s insightful comment. In response, we have substantially shortened the subsection “Distinct Immune Cell Subpopulations Enriched in Solid Sites Across Different *BRCA* Statuses” (**Results, lines 518-530**).

“The model provided in Fig. 8 is very interesting but it is not really supported by the data provided in this manuscript (see comment above).”

Response and revision in the manuscript: We thank the reviewer for this important comment. In the revised manuscript, we have incorporated additional experiments as suggested above by reviewer. These new data provide stronger support for the conceptual framework illustrated in **Figure 8** in the revised manuscript.

Fig. 8. Graphical summary of spatiotemporal heterogeneity in TME and model for the role of the malignant cells-macrophages-CD8⁺ T cells axis in immunosuppressive TME formation associated with HGSOC metastasis.

“Discussion.

The discussion of the novel crosstalk should be extended. There is much more literature on SELENOP and its mode of action which should be discussed here. In addition, a putative role of the trace element selenium could be discussed as well.”

Response and revision in the manuscript: We appreciate the reviewer’s valuable suggestion. We have added relevant content in the **Discussion** section (lines 555-566).

“For the discussion of hypoxia-induced effects on SELENOP refer to Becker et al., 2014, Hypoxia reduces and redirects selenoprotein biosynthesis.”

Response and revision in the manuscript: In response, we have added the relevant content in the **Discussion** section (**lines 587-590**).

“The second part of the discussion starting in line 688 is again out of topic and has not much to do with the results described in the abstract.”

Response and revision in manuscript: We appreciate the reviewer’s comment. In the revised manuscript, we have removed the content related to immune checkpoints from the **Discussion** section.

*“Minor comments:
Abstract, line 37: malignant epithelial cells.”*

Response and revision in manuscript: Thank you for your suggestion. We have updated the manuscript to change "hypoxia malignant cells" to "malignant epithelial cells", now in **line 37** of the **Abstract** section.

“Please use SELENOP as abbreviation for selenoprotein P as this is the official nomenclature.”

Response and revision in the manuscript: Thank you for your suggestion. We have updated the manuscript to use "SELENOP" as the abbreviation for selenoprotein P, in accordance with the official nomenclature throughout the revised manuscript.

“Line 297: I would mention here again that “elevated phagocytosis function” is another result of the KEGG pathway analysis. Otherwise, one could expect a functional readout.”

Response and revision in manuscript: Thank you for pointing this out. We have removed this sentence to avoid any potential misunderstanding.

“Line 334: Please delete “theoretically”.”

Response and revision in the manuscript: Thank you for the suggestion. We have removed the word “theoretically” in the revised version (**Results, line 246-247**).

“Fig. 6i is not very informative for the reader.”

Response and revision in manuscript: Thank you for the suggestion. We have added the representative images of PDOs under hypoxic and normoxic conditions, respectively (**Fig. 6I**). We observed a decrease trend in the PDOs volume under hypoxic conditions (**Results, lines 448-449**).

Fig. 6I. Representative images of PDOs under normoxic (upper) or hypoxic conditions (lower).

“Line 661, Ref 66-68: Only Ref 66 is supporting the respective information about the macrophage subset in lung cancer.”

Response and revision in the manuscript: Thank you for pointing out this point. We have removed Ref 67 and 68 and kept Ref 66, now as Ref 72, which specifically supports the information about the macrophage subset in lung cancer.

“Lines 805-806: specify the title of the section.”

Response and revision in manuscript: We appreciate the reviewer’s suggestion. In response, we have revised the section title to "Cell Type Identification and Characterization" in order to more accurately and concisely reflect the content of this section. The change has been made accordingly in the revised manuscript (**Methods, line 701**).

“Line 1225: Is the amount injected (15 µl) correct?”

Response and revision in the manuscript: Thank you for your comment. We confirm that the injection volume of 15 µl is correct. This protocol was adopted following the methodology described in the published study and has been consistently applied in our

orthotopic implantation experiments¹⁹. We have cited this reference in the revised manuscript as Ref 98.

“Line 1238: statistical analysis One-Way ANOVA is missing (is written in figure legends).”

Response and revision in the manuscript: Thank you for pointing out this question. We have included the use of one-way ANOVA in the revised manuscript (**Methods, lines 1071-1072**).

REVIEWER 4:

“This work by Liu et al. uses multiple state of the art single-cell multi-omics and spatial transcriptomics methods to map the spatiotemporal heterogeneity of HGSOC metastasis and tumor microenvironment (TME) and identifies two major populations of tumor-associated macrophages (TAMs): SELENOP⁺ and SPP1⁺, respectively, which undergo dynamic transition in the setting of hypoxic malignant ovarian cells to subvert immune-mediated tumor control, thus facilitating HGSOC metastasis. Some of the findings are consistent with what others have found in various types of cancers but not all of them have been functionally and definitively validated in patients or in vivo model systems. Given the limitation, it is important that, at a minimum, the pro-metastatic role of SPP1⁺ macrophages, as postulated in this study, should be unequivocally demonstrated by taking at least one of the suggested approaches as follows:

- 1. SPP1-targeting antibodies: use neutralizing antibodies against SPP1 to block its activity and downstream signaling.*
- 2. SPP1 knockout: Global and conditional SPP1 KO mice are commercially available.*
- 3. Receptor Blockade: SPP1 interacts with integrins (e.g., α v β 3, α v β 5) and CD44. One could use receptor-blocking antibodies or peptides (e.g., RGD peptides for integrins).”*

General reponse: We thank the reviewer for the constructive and thoughtful assessment of our study. We acknowledge that these approaches could reveal the pro-metastatic role of SPP1⁺ macrophages, however, we consider that this is beyond the scope of this study. Nonetheless, we would like to provide supporting **Fig. 6**, which is under review for another manuscript by our group, to appreciate the reviewer’s taste. As suggested by the reviewer, we conducted *in vivo* experiments, which showed that CD44 blockade (Angstrom6, **Supporting Fig. 6A-C**) and SPP1 antibody (**Supporting Fig. 6D-F**) markedly suppressed ovarian metastasis in mice.

[Redacted]

[Redacted]

“There are also concerns regarding the claim of direct involvement of SELENOP⁺ macrophages in priming T cells, as there is no direct evidence showing that SELENOP⁺ macrophages can activate CD8⁺ T cells. In contrast, previous studies (PMID: 18424738, 20530259) show that SELENOP is highly upregulated in M2 macrophages, which are generally considered more pro-tumorigenic than M1 macrophages.”

Response: We appreciate the reviewer's insightful comments.

(1) Indeed, SELENOP⁺ macrophages have been identified in lung cancer, exhibiting antigen-presenting capabilities with proposed antitumor function associated with CD8⁺ T cells^{20,21}, but without direct evidence. We conducted both *in vitro* and *in vivo* target experiments, which provided direct evidence supporting a role for SELENOP⁺ macrophages in promoting CD8⁺ T cell activation.

(2) Although SELENOP has previously been identified as part of M2-associated gene signatures^{22,23}, numerous studies have demonstrated that tumor-associated macrophages (TAMs) within the TME exhibit remarkable plasticity and heterogeneity, thereby challenging the traditional M1/M2 polarization paradigm. We characterized the phenotype of SELENOP⁺ macrophages in the context of HGSOC.

These results above are detailed in our response to Comment #3 and #7.

“Some additional specific issues of concern:

1. Fig. 2- Out of on the scRNA-seq data (Extended Data Fig. 2a), only CD8⁺ T cells are analyzed in detail (Fig. 2). However, CD4⁺ T cells are also abundant in the tumor. Why no further analysis on the CD4⁺ T cell population? Are regulatory T cells enriched in late state tumors?”

Response and revision in the manuscript: We thank the reviewer for raising this important point. We have extended our analysis to include CD4⁺ T cell heterogeneity, with a specific focus on regulatory T cells (Tregs). We identified a FOXP3⁺ CD4⁺ Treg subpopulation that is enriched in solid tumor regions and shows a progressive increase from early to late stages (**Extended Fig. 2g-j**). These findings indicate a potential involvement of FOXP3⁺ CD4⁺ Tregs in the metastatic progression of HGSOC (**Results, lines 190-199**).

Extended Data Fig. 2g-j. h, g, UMAP plots depicting the clusters of all CD4⁺T cells, colored by cluster. **h**, Dot plot showing marker genes of CD4⁺T clusters. **i**, Dot plot showing expression patterns of functional signatures across indicated clusters. **j**, Tissue preference of each CD4⁺T clusters, estimated by $R_{0/e}$ ($P < 2.2e-16$). P -values are calculated by chi-squared test. For **g, h, i**, $n = 34$ scRNA-seq cohort samples. For **j** (left), $n = 25$ scRNA-seq cohort samples except for Nor.Ovr and PLF.UF. For **j** (right), total $n = 17$ scRNA-seq cohort solid site samples.

“2.Fig. 3h- The resolution of spatial transcriptomics data is too low. T04_CD8T-GZMH gene expression doesn’t show clear cluster on the spatial transcriptomics data. It appears that the gene expression is mapped to the whole slide. CD8⁺ T cell clusters can be observed on the immunofluorescence staining slides (Fig. 3j). However, based on the Visium data (Extended Data Fig. 3e), the correlation between SELENOP⁺ macrophage and T04_CD8T-GZMH abundance is very weak ($R = 0.11$, Extended Data Fig. 3e).”

Response and revision in the manuscript: We thank the reviewer for this constructive comment.

(1) We have improved the resolution of the spatial transcriptomics data to optimized the visualization of spatial transcriptomics data. T04_CD8T-GZMH gene expression show clear cluster in **Fig. 3h** in the revised manuscript.

(2) We agree with the reviewer that this correlation appears modest, which may be partially attributed to the limited spatial resolution and mixed-cell capture of the Visium platform. Nevertheless, our integrated analyses, including spatial transcriptomics (**Fig. 3h**, presented in (1)) and immunofluorescence staining (**Fig. 3j**), consistently support the spatial proximity between these two cell populations, providing spatial support for the proposed role of SELENOP⁺ macrophages in promoting CD8⁺ T cell activation.

Fig. 3h. Representative spatial co-localizations of *SELENOP*⁺ macrophage with T04_CD8T-GZMH in tumor spots of slide from adnexal sites (left) and zoom-in image (right) in our spatial transcriptomics cohort.

Fig. 3j. Representative immunofluorescence staining showing co-localization of CD68 (orange), SELENOP (green), CD8A (red), GZMH (magenta) and DAPI (blue) in HGSOE samples. Scale bars of each group, 100 µm (left) and 50 µm (right). The white arrow points to the CD8⁺GZMH⁺ cell, and the yellow arrow points to the CD68⁺SELENOP⁺ TAM.

“3.Fig. 3k and Extended Data Fig. 3i- To study the effect of SELENOP⁺ macrophages on CD8⁺ T cell function, the authors use SeP protein to stimulate the OT1 cells. There is no dose-dependent change on the GZMB and PRF1. Can SELENOP⁺ macrophages directly prime CD8⁺ T cells? To verify the role of SELENOP⁺ macrophages, co-culturing of OT1-specific SIINFEKL peptide-loaded-SELENOP⁺ macrophages with OT1 cells should be performed. If it is not feasible to directly isolate SELENOP⁺ macrophages, the SELENOP protein can be overexpressed in macrophages, which can then be used to prime CD8⁺ T cells.”

Response and revision in the manuscript:

We sincerely thank the reviewer for the valuable suggestions.

(1) We treated OT-1 T cells with a refined concentration gradient of recombinant SELENOP (**Extended Data Fig. 3b**). The results demonstrated that GZMB and PRF1 expression in OT-1 T cells increased in a dose-dependent manner within the range of 0 to 100 ng/mL of SELENOP (**Extended Data Fig. 3c**), thereby supporting the biological relevance of SELENOP-mediated T cell activation within a defined concentration range (**Results, lines 247-252**).

Extended Data Fig. 3b,c. **b**, CD8⁺ T cell cytotoxicity followed by recombinant SELENOP treatment. **c**, OVA-specific OT-1 T cells were treated with recombinant murine SELENOP or an equivalent volume of PBS as control for 48 h. The proportions of CD3⁺CD8⁺GZMB⁺ or CD3⁺CD8⁺PRF1⁺ T cells of OT-1 T cells were analyzed using flow cytometry. Data represent the mean \pm SD. Brown-Forsythe and Welch ANOVA test. *, $D < 0.05$, **, $D < 0.01$, ***, $D < 0.001$.

(2) As suggested, we performed the antigen presentation assay (**Extended Data Fig. 3n**) (**Methods, lines 953-958**). SELENOP was first overexpressed (BMDMs-OE-Selenop) or knocked down (BMDMs-KD-Selenop) in BMDMs via lentiviral vectors (**Methods, lines 935-945**). The results showed that OVA-loaded BMDMs-OE-Selenop significantly enhanced GZMB expression, as well as tumor-specific cytotoxic activity in CD8⁺ T cells, compared to the control BMDMs. Meanwhile, OVA-loaded *Selenop*-knockdown BMDMs elicited a significantly weaker cytotoxic response effect, with reduced GZMB and PRF1 expression, as well as the cytotoxic activity of CD8⁺ T cells (**Extended Data Fig. 3o-t**). These findings support the role of SELENOP⁺ macrophages in directly priming CD8⁺ T cells *in vitro* (**Results, lines 272-277**).

Extended Data Fig. 3n-t. **n**, Antigen presentation assay. The proportions of CD3⁺CD8⁺GZMB⁺, CD3⁺CD8⁺PRF1⁺ T cells of OT-1 T cells cocultured directly with BMDMs-puroNC and BMDMs-OE-Selenop (**o**, **p**); BMDMs-BSDNC, BMDMs-KD1-Selenop, BMDMs-KD2-Selenop, BMDMs-KD1-Selenop and BMDMs-KD2-Selenop supplemented with rmSELENOP (**r**, **s**), respectively, and the total apoptosis rate of ID8-OVA cells (**q**, **t**) induced by corresponding CD8⁺ T cells. BMDMs, bone marrow derived macrophages; BMDMs-puroNC, BMDMs transfected with negative control lentivirus; BMDMs-OE-Selenop, BMDMs overexpressing *Selenop* after lentiviral transfection. BMDMs-BSDNC, BMDMs transfected with negative control shRNA; BMDMs-KD1-Selenop, BMDMs with *Selenop* knockdown after transfected with sh*Selenop*-1. BMDMs-KD2-Selenop, BMDMs with *Selenop* knockdown after transfected with sh*Selenop*-2. Data represent the mean \pm SD. One-way ANOVA with Bonferroni post hoc test for multiple groups, two-sided unpaired Student's t test for two groups, $n = 3$. ns, not significant, * $P < 0.05$, ** $P < 0.01$, *** $P < 0.001$.

“4.Fig.6f- The experiment should be performed with purified macrophages from the spleen of mice or bone marrow derived macrophages for more physiological relevance.”

Response and revision in the manuscript: We thank the reviewer for this important suggestion. In response, we repeated the experiment using BMDMs. The results also showed that CM from hypoxia-treated ID8 cells markedly suppressed the expression of SELENOP in BMDMs compared to the normoxic controls, an effect reversed by VEGFA neutralization, as in THP-1 cells co-cultured with CM from ovarian cancer cell lines, supporting the physiological relevance of our conclusions (**Fig. 6f**) (**Results, lines 441-443**).

Fig. 6f. SELENOP expression in THP-1 cells or BMDMs after cultured with conditioned medium (CM) from cell lines under normoxic or hypoxic conditions for 48 h, with or without VEGFA antibody (VEGFA Ab) (f). One-way ANOVA with Bonferroni post hoc test, n = 3. ns: not significant, * $P < 0.05$, ** $P < 0.01$, *** $P < 0.001$.

“5.Fig.6g, h- The *EPHB2* knockdown efficiency is very low. However, the SELENOP expression increases dramatically in the siEPHB2 cells. Is it due to off-target effect of siEPHB2? Anti-VEGF neutralizing antibody can be used to verify the role of VEGF signaling on macrophages. The result should be validated using purified macrophages or bone marrow derived macrophages.”

Response and revision in the manuscript: We appreciate the reviewer’s thoughtful and constructive comments.

(1) We appreciate the reviewer’s thoughtful and constructive comments. We employed the CRISPR-Cas9 system and successfully generated *EPHB2*-deficient THP-1 cells (**Supplementary Fig. 2c**) (**Methods, lines 953-945**). Treatment of THP-1 cells with exogenous VEGFA protein resulted in a decrease SELENOP expression and secretion, which could be reversed by knockdown of *EPHB2* expression, further supporting that VEGFA-mediated suppression of SELENOP is largely dependent on *EPHB2* signaling (**Fig. 6g,h**). As suggested, parallel experiments using BMDMs with *Ephb2* knocked down via lentiviral transduction (**Supplementary Fig. 2d**) (**Methods, lines 953-945**) yielded consistent results (**Fig. 6i**) (**Results, lines 443-445**).

(2) Anti-VEGF neutralizing antibody has been used to verify the role of VEGF signaling on macrophages in **Fig. 6f** presented in Comment #4.

Supplementary Fig. 2c, d. **c**, Western blot analysis confirmed CRISPR/Cas9-mediated *EPHB2* deletion in THP-1 cells. sgNC: empty vector for negative control of sgRNA. **d**, Quantification of *EPHB2* by Western blot in BMDMs after *Ephb2* knockdown by sh*Ephb2*-1 (KD1) or sh*Ephb2*-2 (KD2). Data represent the mean \pm SD. Two-sided unpaired Student's t test for two groups, one-way ANOVA with Bonferroni post hoc test for multiple groups, $n = 3$. * $P < 0.05$, *** $P < 0.001$.

Fig. 6g-i. **g**, Western blot images and quantifications of SELENOP protein expression after knocking down *EPHB2* by CRISPR-Cas9 in THP-1 cells (**g**) or by shRNA in BMDMs (**i**) with 50 ng/mL rVEGAF treated for 48h. sgNC, empty vector for negative control of sgRNA, rh (m) VEGAF: recombinant human (mouse) VEGFA protein. **h**, ELISA showing SELENOP level in the culture supernatants in **g**. One-way ANOVA with Bonferroni post hoc test, $n = 3$. ns: not significant, * $P < 0.05$, *** $P < 0.001$.

“6.Fig.7b- On day 14, the mouse in the anti-VEGF group seems to show stronger luminescence intensity than the control mouse, suggesting that anti-VEGF may not efficiently suppress primary tumor growth. Quantification of luminescence intensity from multiple mice per group should be provided with statistics.”

Response and revision in the manuscript:

We appreciate the reviewer's thoughtful comments. We have now included the corresponding luminescence data at days 7, 14, and 49 post-inoculation for both the

control and anti-VEGF groups, with no statistically significant differences in baseline luminescence intensity (days 7 and 14). This design allows for an accurate evaluation of anti-VEGF efficacy *in vivo*. We finally observed decreased luminescence intensity and tumor burden in the anti-VEGF group, demonstrating the antitumor effect of VEGFA inhibition. (Fig. 7a-g) (Results, lines 453-459).

Fig. 7a-g. **a**, Schematic diagram of the protocol for *in vivo* experiments. C57BL/6 mice inoculated with ID8-luciferase cells are treated with 2.5 mg/kg IgG (control mouse; n = 9) or 2.5 mg/kg anti-VEGFA antibody (anti-VEGFA treatment mouse; n = 9). Representative images (**b**) and quantifications (**c**) of C57BL/6 mice detected at 7d (left), 14d (middle) and 49d (right) by bioluminescence imaging after inoculation with ID8-luciferase cells. 49d after inoculation with ID8-luciferase cells, mice in both groups were euthanized (n = 9 mice per group). Representative images showing characteristic (**d**) and quantifications (**e**) of peritoneal metastases (black arrows) in mice treated with control IgG (upper), and the absence of peritoneal metastases in mice treated with anti-VEGFA antibody (lower). Representative images (**f**) and quantification of tumor volumes (**g**) of primary tumors of mice treated with control IgG or anti-VEGFA antibody. Bars, 10 mm. For **c**, **e**, **g**, data represent the mean ± SD. *P*-values: two-sided unpaired Student's *t* test for two groups. ns, not significant. **P* < 0.05, ***P* < 0.01.

“7.Fig.7c- How are SELENOP⁺ macrophages identified using flow cytometry? The gating strategy should be provided. In this mouse model, do SELENOP⁺ macrophages also co-localize with CD8⁺ T_{pex}? What is the phenotype of this SELENOP⁺ macrophage population? Are they similar to M1 or M2 macrophages? Since the SELENOP⁺ macrophages can be found in this mouse model, those macrophages can be used to test their capacity to prime CD8⁺ T cells in vitro and in vivo.”

Response and revision in the manuscript: We sincerely thank the reviewer for the insightful and constructive comments.

(1) We have clarified that the gating strategy used to identify SELENOP⁺ macrophages by flow cytometry was based on CD45⁺/F4/80⁺CD11b⁺/SELENOP⁺ expression in **Source data (sheet Fig. 7)**, presented as below.

(2) We performed multiplex immunohistochemistry in the orthotopic ovarian cancer model. The results also revealed a close colocalization between SELENOP⁺ macrophages and CD8⁺GZMH⁺ T cells, similar to the spatial distribution observed in human HGSOc tumors (**Fig. 7k**) (**Results, lines 465-467**).

Fig. 7k. Representative immunofluorescence staining showing co-localization of F4/80 (red), SELENOP (green), CD8A (magenta), GZMH (cyan) and DAPI (blue) in C57BL/6 mice samples.

(3) We analyzed the correlation of the M1 score and M2 score of SELENOP⁺ macrophages in our scRNA-seq data, which revealed the coexistence of both functional phenotypes (**Supporting Fig. 7a**). Then we established a subcutaneous tumor model in female C57BL/6 mice and performed flow cytometric analysis on tumor-infiltrating macrophages. Following identification of these macrophages, we further stratified them into four subpopulations based on surface expression of the canonical M1 marker CD86 and the M2 marker CD206. Notably, we observed an enrichment of a CD86^{hi}_{g^h}/CD206^{hi}_{g^h}

subset in tumors, suggesting a noncanonical, intermediate polarization state (**Supporting Fig. 7b**).

Supporting Fig. 7. Profile of SELENOP⁺ macrophages. **a**, Scatterplot showing the Spearman correlation of the M1 score and M2 score of *SELENOP*⁺ macrophages in our scRNA-seq cohort (total n = 34 samples). **b**, FACS profiles illustrating the percentage of F4/80⁺CD11b⁺SELENOP⁺CD86⁺CD206⁺ macrophages in subcutaneous tumors in female C57BL/6 mice (n=3).

(4) The proportion of SELENOP⁺ macrophages obtained by FACS (without permeabilization and stimulation) was low, below 10% (see **Supporting Fig. 8**), rendering their use in subsequent *in vivo* and *in vitro* experiments impractical. Therefore, we followed the reviewer's valuable suggestion in Comment #3 and evaluated their functional capacity both *in vitro* and *in vivo* using *Selenop*-overexpressing macrophages. The *in vitro* experiments to validate the capacity of *SELENOP*⁺ macrophages to prime CD8⁺ T cells were presented in the response to Comment #3.

Supporting Fig. 8. Proportion of SELENOP⁺ macrophages detected by FACS.

(3) The *in vitro* experiments were performed as below (**Extended Data Fig. 7a**). Then we found that, compared with the BMDMs group, the proportion of CD45.2⁺CD3⁺CD8⁺GZMB⁺ T cells decreased, accompanied by reduced frequencies of CD45.1⁺F4/80⁺CD11b⁺SELENOP⁺ macrophages in the primary tumors of the BMDMs-KD1-Selenop group. Meanwhile, VEGFA inhibition led to a concomitant reduction in CD45.2⁺CD3⁺CD8⁺GZMB⁺ T cells and CD45.1⁺F4/80⁺CD11b⁺SELENOP⁺ macrophages in the primary tumors of mice injected with BMDMs-KD1-Selenop cells, but had no significant effect in mice injected with BMDMs (**Extended Data Fig. 7b,c**). These findings support the role of SELENOP⁺ macrophages in priming CD8⁺ T cell cytotoxicity within the HGSOc tumor microenvironment *in vivo*, and further emphasize a VEGFA-mediated regulatory mechanism that appears to be more pronounced in macrophages with low SELENOP expression (**Results, lines 471-485**).

Extended Data Fig. 7. a, Schematic diagram of the protocol for adoptive cellular transfer experiments. Fresh primary tumor tissues were digested and stained for flow cytometry analysis. Shown is the proportion of $CD45.1^+F4/80^+CD11b^+SELENOP^+$ cells (**b**) and $CD45.2^+CD3^+CD8^+GZMB^+$ cells (**c**) across different sites of each group. Pri. tumor: primary tumor of orthotopic ovarian cancer mouse model; Met.Per: peritoneal metastasis of orthotopic ovarian cancer mouse model. *P*-values are calculated by two-way ANOVA with Tukey post hoc test. $n = 5$ in BMDMs group, $n = 5$ in BMDMs-KD1-Selenop group, $n = 4$ in BMDMs combined with anti-VEGFA group, $n = 5$ in BMDMs-KD1-Selenop combined with anti-VEGFA group. ns: not significant, $*P < 0.05$, $**P < 0.01$, $****P < 0.0001$.

REVIEWER 5:

“The paper maps the full spatiotemporal progression of high-grade serous ovarian cancer (HGSOC). Thus it improves over the current literature, which predominantly focuses on the tumor microenvironment (TME) of advanced HGSOC. The authors employ multimodal profiling techniques including scRNA-seq, spatial transcriptomics, scTCR-seq, RNA-seq, organoids, and orthotopic syngeneic models. They report spatial proximity between SELENOP⁺ macrophages and precursor exhausted CD8⁺ T cells. Their findings suggest that hypoxia-driven remodeling of the TME (via the malignant epithelial cells) plays a critical role in facilitating metastasis, and indicate the potential for combining anti-VEGFA therapy with immunotherapy.

The question addressed by this work is important, and overall its execution is promising. I think this will be a useful contribution to scientific progress in this field. I particularly appreciate that the authors chose to validate some of their findings using publicly available TCGA data (Fig. 3f). I have a few comments that might help improve the impact of this work and its presentation.”

General response: We thank the reviewer for the positive and constructive feedback as *“The question addressed by this work is important, and overall its execution is promising. I think this will be a useful contribution to scientific progress in this field.”*. We are encouraged by the recognition of our study’s significance and its potential contribution to advancing the understanding of HGSOC progression. In the following, we provide a point-by-point response to the reviewer’s comments regarding the clarity and consistency of data presentation and methodology.

*“1. **Ratio of observed to expected:** The usage of the ratio of observed to expected seems inconsistent. For example, in Fig. 2f, each row sums up to 1, whereas in Fig. 3b, this is not the case. This should either be investigated, or the methodology should be explained more clearly.”*

Response and revision in the manuscript: We appreciate the reviewer’s careful observation. We would like to clarify that there is no inconsistency in the methodology regarding the ratio of observed to expected. In both **Fig. 2f** (now **Extended Data Fig. 2c**, left) and **Fig. 3b**, we applied the same calculation method for calculating the ratio of observed to expected cell numbers (R_{ote}) as mentioned in Method section “ R_{ote} analysis” (**Methods, lines 736-744**). The fact that in **Fig. 2f** (now **Extended Data Fig. 2c**, left) the values in each row appear to sum to approximately 2 is purely coincidental due to rounding to two decimal places for visualization purposes. The raw values do not strictly sum to 2, as presented as **Supporting Table 3** below.

Supporting Table 3. Raw $R_{o/e}$ values of CD8⁺ T cell subclusters across solid and fluid compartments shown in Extended Data Fig. 2c, left.

CD8 ⁺ subclusters	Solid sites	Fluid sites
CD8 ⁺ TRDV2 ⁺ $\gamma\delta$ T (T09)	0.7584212	1.2423671
CD8 ⁺ SLC4A10 ⁺ MAIT (T08)	0.6207193	1.3805184
CD8 ⁺ CXCL13 ⁺ Tex (T06)	1.3975388	0.6011639
CD8 ⁺ IFIT3 ⁺ Tisg (T07)	1.5190566	0.4792496
CD8 ⁺ GNLY ⁺ NK-like (T05)	0.7069565	1.2939998
CD8 ⁺ GZMH ⁺ Tpex (T04)	1.0398623	0.9600077
CD8 ⁺ XCL1 ⁺ Trm (T03)	0.8697651	1.1306599
CD8 ⁺ GZMK ⁺ Tem (T02)	1.0334714	0.9664194
CD8 ⁺ CCR7 ⁺ Tn (T01)	0.8561904	1.1442788

*“2. **Methodology of NMF to define programs:** the way that the metaprograms are defined seems unnecessarily convoluted. I would rather start with the GO approach to get initial cluster annotations, and highlight the similarities with existing signatures afterwards.”*

Response and revision in the manuscript: As recommended, we first employed Gene Ontology (GO) analysis to obtain initial cluster annotations. The corresponding modifications have also been made in the revised manuscript (**Results, lines 356-359**).

*“3. **Grammatical errors and typos:** there are some minor grammatical errors within the text (e. g. lines 31, 57, 115, 318, 320, 370, 444, 585, 630). Fig. 2b refers to precursor exhausted T cells as Texp, whereas Tpex is used everywhere else. Fig. 2d should be “expansion index”. Line 130: “mere” seems like the wrong word here.”*

Response and revision in the manuscript: We thank the reviewer for pointing out these language issues.

(1) Corresponding revisions of grammatical errors within the text have been made in the revised manuscript (**lines 29-32, 57-60, 109-114, 225-229, 229-230, 307-309**).

(2) Additionally, we corrected **Fig. 2b** and **Fig. 2d** (now **Extended Data Fig. 2d**). We also have removed the word “mere” (**Results, lines 118-122**).

Fig. 2b. Heatmap indicating the expression levels of gene signatures of previously reported Tpex and Tex cell states, and exhaustion signature in conventional CD8⁺ T subtypes.

Extended Data Fig. 2d. Clonal expansion of the clusters of conventional CD8⁺ T cells quantified by STARTRAC indices (PRJCA005422). Each dot represents a patient. Box, median \pm interquartile range. *P*-values: Wilcoxon Rank-Sum test. For **d**, a total of *n* = 14 HGSOC samples (PRJCA005422).

*“4. ****Presentation of expression data:**** the authors use a rainbow-style colormap to show expression values in Fig. 2a, 3a, 4]. There is however no standardization as to which color represents 0. I would instead encourage the use of a divergent color palette with a clear center, such as the coolwarm palette already used in Fig. 2b. The labels of the colorbar in Fig. 3a are not aligned properly.”*

Response and revision in the manuscript: We thank the reviewer for this helpful suggestion. In response, we have revised the figures.

Fig. 3a. UMAP plots depicting the clusters of CD8⁺ T cells, colored by cluster. Conventional and unconventional CD8⁺ T cells are highlighted with black dotted lines, respectively (left). Dot plot showing marker genes of conventional CD8⁺ T clusters (right). Dot size representing percent of expressing cells in each cluster and color represents z-score of normalized mean expression level of selected genes. For **a**, n = all 34 scRNA-seq cohort samples.

Fig. 4a. UMAP projection of 13 myeloid clusters colored by clusters (left). Dot plot showing expression patterns of selected genes across indicated clusters. Dot size represents percent of expressing cells in each cluster and color represents z-score of normalized mean expression level of selected genes (right). For **a**, n = all 34 scRNA-seq cohort samples.

Fig. 4j. Heatmap showing the RNA expression of transcription factors along the pseudotime trajectory. n = 17 scRNA-seq cohort solid site samples.

“5. **Fig. 2g-i:** it is not clear why clusters T03 and T07 were excluded here.”

Response and revision in the manuscript: We appreciate the reviewer’s insightful comment.

(1) In our work, we focused the pseudotime analysis on CD8⁺ T cell subsets with cytotoxic potential that are responsive to TCR-mediated stimulation in solid tumors, rather than interferon signaling. ISG⁺CD8⁺ TILs exhibit transcriptional inertness to TCR-based stimulation²⁴, and therefore we excluded this population from pseudotime inference in the main analysis. Additionally, subcluster T03 (Trm) was excluded due to its predominant enrichment in ascites, making it less representative of the solid tumor microenvironment (**Results, lines 164-173**).

(2) In parallel, we also expanded the pseudotime analysis to include all conventional CD8⁺ T cell clusters (**Supporting Figure 9**). This comprehensive analysis also revealed two distinct differentiation trajectories. In contrast to the original model—where one trajectory ended in a precursor exhausted/memory state (T02, T04) and the other in terminal exhaustion (T06)—the full-model trajectories culminated in T06 (exhausted) and T07 (ISG⁺) cells, respectively. This reflects the potential different differentiation pathway of CD8⁺ T cells between interferon and TCR-mediated stimulation, in line with previous studies^{24,25}.

Supporting Figure 9. The trajectory plot of all conventional CD8⁺ T cell clusters of tumor sites along pseudotime inferred by Monocle2. Pseudotime of six conventional CD8⁺ T clusters. Each point corresponds to a single cell. Clusters information are shown. The pie charts showing the percentage of clusters in each state.

“6. **Fig. 4d:** Once again, the choice of only showing clusters M05, M06, and M09 is not clearly explained in the text. I would also appreciate it if the same analysis was performed on all macrophage clusters (or at least the TAMs) and put in the supplement (same for the previous point).”

Response and revision in the manuscript: We thank the reviewer for this valuable comment. In Fig. 4d, we focused on clusters M05, M06, and M09 for the pseudotime analysis because our primary aim was to investigate the tumor-associated macrophages

in the solid TME. Among these, M06 and M09 were the two dominant populations specifically enriched in solid tumor sites, while M05 represented monocyte-like precursors that likely give rise to these subsets. As suggested, we have performed additional pseudotime analyses incorporating all monocyte-macrophage clusters. These results showed a similar bifurcating pattern, with two distinct differentiation trajectories culminating in *SELENOP*⁺ and *SPP1*⁺ macrophage-enriched endpoints, supporting our original findings (**Supporting Figure 10**) (**Results, lines 304-309**).

Supporting Figure 10. The trajectory plot of all monocyte macrophage clusters of tumor sites along pseudotime inferred by Monocle2. Pseudotime of five monocyte macrophage clusters. Each point corresponds to a single cell. Clusters information are shown. The pie charts showing the percentage of clusters in each state.

“7. **Fig. 4e:** would benefit from adding plot titles directly in the figure, showing which enrichment analysis belongs to which state.”

Response and revision in the manuscript: We appreciate the reviewer’s suggestion and have revised **Fig. 4e** accordingly to improve its readability and interpretability.

Fig. 4e. Pathway enrichment analysis of the differential genes of cells in different state. n = 17 scRNA-seq cohort solid site samples.

“8. **Fig. 6c:** there should be a distinction between 0 and negative importances.”

Response and revision in the manuscript: We thank the reviewer for pointing this out and have revised **Fig. 6c** accordingly to enhance the interpretability of the results.

Fig. 6c. Heatmap showing the importances among all cell types signature scores in spots of spatial transcriptomics data. Data were summarized from $n = 24$ spatial RNA cohort samples.

“9. ****Extended Data Fig. 4:**** in every other $R_{o/e}$ plot, the numbers are shown, but not in this one.”

Response and revision in the manuscript: We thank the reviewer for noticing this inconsistency. We have now revised **Extended Data Fig. 4c**.

Extended Data Fig. 4c. Tissue preference of each macrophage clusters in adnexal sites and peritoneal foci in cohort (syn52458609), estimated by the $R_{o/e}$ analysis ($P < 2.2e-16$). P -values are calculated by chi-squared test.

“0. ****Code Availability:**** the authors claim that the code is available (Line 1254), however there is no link to a GitHub repository (or equivalent).”

Response and revision in the manuscript: We thank the reviewer for pointing this out. We have now uploaded the code to a publicly accessible GitHub repository. The link to the repository has been added to the **Code Availability** section of the manuscript (**lines 1090-1092**).

REVIEWER 6:

“Liu et al. characterized the spatial heterogeneity of HGSOC across different progression stages and tissue sites using scRNA-seq and spatial transcriptomics. Their study identifies the co-occurrence and spatial colocalization of SELENOP⁺ macrophages and precursor exhausted CD8⁺ T (Tpex) cells, revealing that SELENOP⁺ macrophages promote CD8⁺ T cell activation in early tumors. They further showed that as the tumor progresses and metastasizes, SELENOP⁺ macrophages are reprogramed into SPP1⁺ macrophages by hypoxia-induced malignant epithelial cells through VEGFA-EPHB2 signaling. Notably, anti-VEGFA therapy restores SELENOP⁺ cells in ovarian cancer cell lines and mouse models. While the study provides insights into the role of macrophage subsets in HGSOC progression, it would benefit from a more rigorous analysis. The key findings are not effectively emphasized, some results are overly descriptive, and the mechanistic validation requires stronger supporting evidence.”

General response: We thank the reviewer for the constructive and encouraging comments that highlight the significance of our study. In response to the concerns raised, we have carefully revised the manuscript to incorporate a more rigorous analysis, better highlight the key findings, present the results in a more objective and concise manner, and provide additional supporting evidence to strengthen the mechanistic validation. Please find our detailed point-by-point responses below.

“Here are my major concerns:

1. The authors highlight the inclusion of early-stage patients as a strength, but similar comparisons have been made in previous studies (such as PMID35675036, 38278958). While one is cited in the discussion, these works should also be discussed. Additionally, the authors should compare their findings with these studies to clarify how their results align or differ from existing literature.”

Response and revision in the manuscript: We thank the reviewer for the valuable comment. We have carefully reviewed the two referenced studies^{26,27}. Although our findings, like those of the aforementioned studies, identified alterations in specific immune cell subsets associated with ovarian cancer progression, our work goes a step further by simultaneously examining both tumor and immune cell compartments within the ovarian cancer microenvironment. Importantly, we revealed a novel cross-talk among macrophages, T cells, and tumor cells associated with the progression of HGSOC. Such mechanistic insights have not been systematically addressed in previous studies (**Discussion, lines 591-597**).

“2. The authors repeatedly use the term “significant” without indicating statistical tests, such as in the R_{0/e} analysis (Fig. 1f, Extended Fig. 2i, j). They should either remove or revise their claims or perform appropriate statistical tests. Specifically, as the R_{0/e} analysis is throughout the study and underpins several conclusions, a chi-square test should be

applied to assess tissue preference. Additionally, the equation used for R_{ote} analysis should be explicitly provided.”

Response and revision in the manuscript: We thank the reviewer for this insightful comment. In response, we have performed chi-square tests to support our R_{ote} analyses for assess tissue preference. The statistical methods and corresponding P values have been added to the relevant figures’ legends in the revised manuscript (**Fig. 1e, Fig. 3b, Extended Data Fig. 2c,j, Extended Data Fig. 4a,c,d, Extended Data Fig. 8a,d,j,k and Supplymentary Fig. 8a,c**). In addition, the equation used for the R_{ote} analysis is explicitly provided in the revised manuscript in the **Methods** section (**lines 736-744**). We also removed the term "significant" without indicating statistical tests throughout the revised manuscript.

“3. Although Tregs are not the focus of this study, their absence is notable. The authors should address their relevance in HGSOC progression. Additionally, since T cells and NK cells were initially grouped together, separating them into distinct clusters would allow for more precise characterization.”

Response and revision in the manuscript: We thank the reviewer for this valuable suggestion.

(1) We extended our analysis to include $CD4^+$ T cell heterogeneity, with a specific focus on regulatory T cells (Tregs). We identified a $FOXP3^+$ $CD4^+$ Treg subpopulation that is enriched in solid tumor regions and shows a progressive increase from early to late stages (**Extended Data Fig. 2g-j**) (**Results, lines 190-199**).

(2) We separated T cells and NK cells into distinct clusters in our analysis to enable more precise characterization (**Fig. 1c-f**) (**Supplementary Note 1, lines 4-6**).

Extended Data Fig. 2g-j. **h, g**, UMAP plots depicting the clusters of all CD4⁺T cells, colored by cluster. **h**, Dot plot showing marker genes of CD4⁺ T clusters. **i**, Dot plot showing expression patterns of functional signatures across indicated clusters. **j**, Tissue preference of each CD4⁺T clusters, estimated by $R_{o/e}$ ($P < 2.2e-16$). P -values are calculated by chi-squared test. For **g, h, i**, $n = 34$ scRNA-seq cohort samples. For **j** (left), $n = 25$ scRNA-seq cohort samples except for Nor.Ovr and PLF.UF. For **j** (right), total $n = 17$ scRNA-seq cohort solid site samples.

Fig. 1c-f. **c**, Uniform Manifold Approximation and Projection (UMAP) plots of the clustering of main cell types from all samples of scRNA-seq cohort. Cell lineages are highlighted with black dotted lines. Individual cells (dots) are colored by clusters. NK, natural killer; CAF, cancer-associated fibroblasts; SMC, smooth muscle cells. **d**, Track plot indicating selected marker genes in each cell type. **e**, Tissue preference of each major cluster across solid and fluid sites (left), EAT, LAT and Met groups (right), estimated by $R_{o/e}$ ($P < 2.2e-16$). P -values are calculated by chi-squared test. **f**, Cell proportions of major cell types in different groups, colored by corresponding cell type colors in **c**. For **c, d, f**, $n =$ all 34 scRNA-seq cohort samples. For **e**, $n = 25$ scRNA-seq cohort samples except for Nor.Ovr and PLF.UF.

“4. Quantifying the spatial distribution of macrophage and T_{pex} is valuable. However, the authors should include zoom-in images to visually support the computationally inferred spatial relationships between SELENOP⁺ or SPP1⁺ macrophage and T_{pex}.”

Response and revision in the manuscript: We thank the reviewer for this helpful suggestion. In response, we made the corresponding improvements. The updated images are presented in **Fig. 3h** of the revised manuscript.

Fig. 3h. Representative spatial co-localizations of *SELENOP*⁺ macrophage with T04_CD8T-GZMH in tumor spots of slide from adnexal sites (left) and zoom-in image (right) in our spatial transcriptomics cohort.

*“5. The *SELENOP*⁺ macrophages appear abundant in scRNA-seq and Visium data but are scarce in IF staining (Fig. 3] and 4b). The authors should provide an explanation for this discrepancy, such as differences in detection sensitivity, technical limitations, or biological factors affecting protein versus transcript expression.”*

Response and revision in the manuscript: We thank the reviewers for this crucial comment. The discrepancy in *SELENOP*⁺ macrophage abundance was primarily attributed to the different underlying principles of the detection methods. RNA expression was quantified using methods that yield continuous variables, whereas protein detection by immunofluorescence is generally interpreted in an all-or-none manner at the single-cell level. This methodological difference likely accounts for the apparent discrepancy in data representation²⁸⁻³⁰.

*“6. In the BRCA analysis, the BRCAmut group includes two EAT samples, comprising half of the total samples, while the wt group consists only of late-stage samples. This raises the concern that the observed high proportions of *Tpex*, *Tex*, and *SELENOP*⁺ macrophages in BRCAmut may be driven by differences in sample composition rather than the BRCA mutation itself. The authors should clarify how they account for these confounding factors.”*

Response and revision in the manuscript: We thank the reviewer for this insightful comment. We have removed the two EAT samples from the BRCA mut group to eliminate potential bias related to sample composition. In addition, we performed whole-exome sequencing (WES) on two samples (OCL3 and OCL6) for which *BRCA* mutation status had not been previously determined (**Supplementary Methods, lines 213-233**); both were inferred to be *BRCA* wild-type and subsequently incorporated into the BRCAw group (**Supplementary Table 9**), thereby expanding and balancing the cohort. Importantly,

even after these adjustments, the enrichment of T_{pex}, T_{ex}, and *SELENOP*⁺ macrophages in BRCAmut group remained evident (**Supplementary Fig. 8a,c**). We also observed consistently elevated cytotoxicity scores in CD8⁺CXCL13⁺ T_{ex} (T06) and CD8⁺GZMH⁺ T_{pex} (T04) cells, as well as higher proinflammatory scores in *SELENOP*⁺ macrophages (M06) in the BRCAmut group compared to BRCAw (**Supplementary Fig. 8b,d**). We believe these revisions directly address the reviewer's concern and further reinforce the robustness of our conclusions (**Results, lines 518-530 and Supplementary Note 6, lines 97-113**).

Supplementary Fig. 8. a, Tissue preference of conventional CD8⁺ T subpopulation between BRCAmut and BRCAw groups, estimated by R_{oe} ($P = 9.725e-10$). P -values are calculated by chi-squared test. **b**, Violin plots showing tumor specific and cytotoxic signature scores of T04_CD8T-GZMH (left) and T06_CD8T-CXCL13 (right) between BRCAmut and BRCAw groups. P values calculated by two-sided Wilcoxon tests. BRCAmut, BRCA mutant group. BRCAw, BRCA wild-type group; **c**, Tissue preference of macrophage subpopulation between BRCAmut and BRCAw groups, estimated by R_{oe} ($P = 0.00000971$). P -values are calculated by chi-squared test. **d**, Violin plots showing proinflammatory (left) and antigen processing and presentation (right) signature scores of *SELENOP*⁺ macrophages between BRCAmut and BRCAw groups. P values calculated by two-sided Wilcoxon tests. For **a-d**, data were summarized from all $n = 6$ samples. BRCAw: $n = 4$, BRCAmut: $n = 2$. Box of violin plot represents median \pm interquartile range.

“7. Given the significant growth differences between PDOs and the OVACR3 tumor cell line, both models were subjected to 48-hour normal and hypoxic cultures. Have

alternative hypoxic culture durations been explored? Additionally, the authors should specify the cell numbers of PDOs and OVACR3 used to prepare the conditioned medium.”

Response and revision in the manuscript: We appreciate the reviewer’s thoughtful question.

(1) We assessed VEGFA expression in different ovarian cancer cell lines exposed to hypoxia for 24 h, 48 h, and 72 h, and in patient-derived organoids (PDOs) for 72 h and 96 h. VEGFA expression in OVCAR3 cells and PDOs showed the most pronounced upregulation under hypoxic conditions at 48 and 96 h, respectively, compared to the normoxic controls (**Fig. 6e, j**). In addition, we also assessed VEGFA expression in CAOV3, COV362 and ID8 exposed to hypoxia for different durations, with VEGFA expression most significantly upregulated in hypoxic CAOV3 at 48 h, and in hypoxic COV362 and ID8 at 72 h, compared to normoxia (**Fig. 6e**) (**Results, lines 435-440, 446-447**).

(2) We observed that SELENOP expression in the THP-1 cells were markedly reduced following co-culture with hypoxia-conditioned media (CM) from OVCAR3 (48 h), compared to normoxic controls. This effect could be rescued by VEGFA neutralization (**Fig. 6f**). CM collected from the 96-h hypoxia-treated PDOs also significantly suppressed SELENOP expression in MDMs (**Fig. 6k**). Parallel experiments using hypoxia-conditioned media from CAOV3, COV362 and ID8 yielded consistent results (**Fig. 6f**). Also, we observed a decrease trend in PDOs volume under hypoxic conditions (**Fig. 6l**) (**Results, lines 440-449**).

(4) We also specified the cell numbers of OVACR3 (5×10^5) and PDOs (~ 200 clones) used to prepare the CM (**Methods, lines 989-991, lines 1016-1018**).

Fig. 6e,f, j-l. Western blot images and quantifications of VEGFA in cell lines under normoxic or hypoxic conditions for 24 h, 48 h, and 72 h (e). SELENOP in THP-1 or BMDMs after cultured with conditioned medium (CM) from cell lines under normoxic or hypoxic conditions for 48 h, with or without VEGFA antibody (VEGFA Ab) (f). Western blot images and quantifications of VEGFA in PDOs under normoxic or hypoxic conditions for 72 h and 96 h (j). SELENOP in MDMs after cultured with CM from PDOs under normoxic or hypoxic conditions (96 h) (k). MDMs, human monocyte-derived macrophages. l. Representative images of PDOs under normoxic (upper) or hypoxic conditions (lower). Data represent the mean \pm SD. For e, j, k, two-sided unpaired Student's t test, n = 3. For f, one-way ANOVA with Bonferroni post hoc test, n = 3. ns: not significant, * $P < 0.05$, ** $P < 0.01$, *** $P < 0.001$.

“8. The authors show that SELENOP⁺ macrophages activate CD8⁺ T cells via SeP, but validation is limited. While exogenous SeP increased GZMB and PRF1 expression and co-culture assays assessed cytotoxicity, these alone are insufficient. Blocking SeP secretion in macrophages would strengthen mechanistic validation.”

Response and revision in the manuscript: We thank the reviewer for this insightful suggestion. In response, we generated the *Selenop* knockdown BMDMs by lentivirus transfection, with reduced SELENOP expression (**Supplementary Fig. 2a**) (**Methods, lines 935-945**). Macrophage-derived CM and CD8⁺ T cell coculture system were established as illustrated in **Fig 3k** (**Methods, lines 947-951**). CM from *Selenop*-knockdown BMDMs (KD1 and KD2) induced a weaker cytotoxic response effect, with reduced both GZMB and PRF1 expression, as well as the cytotoxic activity of CD8⁺ T cells, which could be reversed by supplementation of rmSELENOP (**Extended Data Fig. 3h-j**). Meanwhile, we also established *Selenop*-overexpressing macrophages (BMDMs-OE-Selenop and THP-1-OE-SELENOP) (**Supplementary Fig. 2a,b**) (**Methods, lines 935-945**). The CM from these cells enhanced the cytotoxic activity of CD8⁺ T cells (**Fig. 3l** and **Extended Data Fig. 3f,g,k-m**). In fact, we also performed the antigen presentation assay, with OVA-loaded BMDMs cocultured directly with OT-1 T cells (**Extended Data Fig. 3n**) (**Methods, lines 953-958**). The results were also consistent with those observed in the macrophage-conditioned medium and CD8⁺ T cell co-culture system. Taken

together, these results support the ability of *SELENOP*⁺ macrophages to activate CD8⁺T cells via *SELENOP* (**Extended Data Fig. 3o-t**) (**Results, lines 257-277**).

Supplementary Fig. 2a,b. **a**, Quantification of *SELENOP* by Western blot in BMDMs after infected with a lentivirus encoding *Selenop* (**left**), or after *Selenop* knockdown by *shSelenop-1* (KD1) or *shSelenop-2* (KD2) (**right**). **b**, Quantification of *SELENOP* by Western blot in THP-1 after infected with a lentivirus encoding *SELENOP*. Data represent the mean \pm SD. Two-sided unpaired Student's t test for two groups, one-way ANOVA with Bonferroni post hoc test for multiple groups, $n = 3$. * $P < 0.05$.

Fig. 3k,i. **k**, CD8⁺ T cell cytotoxicity when cocultured with macrophage-derived conditioned medium (CM). **i**, The total apoptosis rate of ID8-OVA cells induced by corresponding CD8⁺ T cells. Data represent the mean \pm SD, two-sided unpaired Student's t test, $n = 3$, **** $P < 0.0001$.

Extended Data Fig. 3f-t. The proportions of CD3⁺CD8⁺GZMB⁺, CD3⁺CD8⁺PRF1⁺ T cells of OT-1 T cells or human CD8⁺ T cells co-cultured with CM from BMDMs-puroNC and BMDMs-OE-Selenop (**f, g**); BMDMs-BSDNC, BMDMs-KD1-Selenop, BMDMs-KD2-Selenop, BMDMs-KD1-Selenop and BMDMs-KD2-Selenop supplemented with rmSELENOP (**h, i**); THP-1-puroNC and THP-1-OE-SELENOP (**k, l**), respectively, and the total apoptosis rate of ID8-OVA cells (**j**) or OVCAR3 (**m**) induced by corresponding CD8⁺ T cells. **n**, Antigen presentation assay. The proportions of CD3⁺CD8⁺GZMB⁺, CD3⁺CD8⁺PRF1⁺ T cells of OT-1 T cells cocultured directly with BMDMs-puroNC and BMDMs-OE-Selenop (**o, p**); BMDMs-BSDNC, BMDMs-KD1-Selenop, BMDMs-KD2-Selenop, BMDMs-KD1-Selenop and BMDMs-KD2-Selenop supplemented with rmSELENOP (**r, s**), respectively, and the total apoptosis rate of ID8-OVA cells (**q, t**) induced by corresponding CD8⁺ T cells. BMDMs, bone marrow derived macrophages; BMDMs-puroNC, BMDMs transfected with negative control lentivirus; BMDMs-OE-Selenop, BMDMs overexpressing *Selenop* after lentiviral transfection. BMDMs-BSDNC, BMDMs

transfected with negative control shRNA; BMDMs-KD1-Selenop, BMDMs with *Selenop* knockdown after transfected with sh*Selenop*-1. BMDMs-KD2-Selenop, BMDMs with *Selenop* knockdown after transfected with sh*Selenop*-2. THP-1-puroNC, THP-1 transfected with control negative lentivirus; THP-1-OE-SELENOP, THP-1 overexpressing *SELENOP* after lentiviral transfection. For **b-d**, **f-n**, **p-u**, data represent the mean \pm SD. For **f-n**, **p-u**, One-way ANOVA with Bonferroni post hoc test for multiple groups, two-sided unpaired Student's t test for two groups, $n = 3$. ns, not significant, $*P < 0.05$, $**P < 0.01$, $***P < 0.001$, $****P < 0.0001$.

“9. The bioluminescence images show only one representative mouse per group. Can the authors confirm whether the same mouse was imaged on days 7, 14, and 42? To enhance data reliability, it is recommended that images of at least three mice per group be included.”

Response and revision in the manuscript: We agree and thank the reviewer for the comment. We have included the same three representative mice images per group on days 7, 14, and 49 in the revised figures (Fig. 7a-c). This modification improves the reliability and robustness of the data (Results, lines 453-459).

Fig. 7a-c. **a**, Schematic diagram of the protocol for *in vivo* experiments. C57BL/6 mice inoculated with ID8-luciferase cells are treated with 2.5 mg/kg IgG (control mouse; $n = 9$) or 2.5 mg/kg anti-VEGFA antibody (anti-VEGFA treatment mouse; $n = 9$). Representative images (**b**) and quantifications (**c**) of C57BL/6 mice detected at 7d (left), 14d (middle) and 49d (right) by bioluminescence imaging after inoculation with ID8-luciferase cells. 49d after inoculation with ID8-luciferase cells, mice in both groups were euthanized ($n = 9$ mice per group). For **c**, data represent the mean \pm SD. P -values: two-sided unpaired Student's t test. ns, not significant, $*P < 0.05$.

“Minor points:

1. The authors should provide the full name of ISG and PARPi and others if they are first mentioned.”

Response and revision in the manuscript: We thank the reviewer for pointing this out. We have now provided the full names for all abbreviations (e.g., ISG and PARPi) at their first appearance in the manuscript (**Introduction, lines 56-57**) (**Supplementary Note 2, lines 44**).

“2. The authors should indicate the cell fate on top of the corresponding barplot in Fig4e for better readability.”

Response and revision in the manuscript: We thank the reviewer for pointing this out. We have added the corresponding cell fate annotations above each bar in **Fig. 4e** to improve readability.

Fig. 4e. Pathway enrichment analysis of the differential genes of cells in different state. n = 17 scRNA-seq cohort solid site samples.

“3. In the first paragraph of the first results section, line 12, there is an extra "and" at the end of the sentence.”

Response and revision in the manuscript: We thank the reviewer for the careful reading and have removed the redundant "and" in the first paragraph of the results section (**lines 106**).

REFERENCES OF RESPONSE

1. Domcke, S., Sinha, R., Levine, D.A., Sander, C. & Schultz, N. Evaluating cell lines as tumour models by comparison of genomic profiles. *Nature communications* **4**, 2126 (2013).

2. Launonen, I.M., *et al.* Chemotherapy induces myeloid-driven spatially confined T cell exhaustion in ovarian cancer. *Cancer cell* **42**, 2045-2063.e2010 (2024).
3. Zhang, L., *et al.* Lineage tracking reveals dynamic relationships of T cells in colorectal cancer. *Nature* **564**, 268-272 (2018).
4. Cheng, S., *et al.* A pan-cancer single-cell transcriptional atlas of tumor infiltrating myeloid cells. *Cell* **184**, 792-809.e723 (2021).
5. Burk, R.F. & Hill, K.E. Selenoprotein P-expression, functions, and roles in mammals. *Biochimica et biophysica acta* **1790**, 1441-1447 (2009).
6. Mizuno, A., *et al.* An efficient selenium transport pathway of selenoprotein P utilizing a high-affinity ApoER2 receptor variant and being independent of selenocysteine lyase. *The Journal of biological chemistry* **299**, 105009 (2023).
7. Burk, R.F. & Hill, K.E. Regulation of Selenium Metabolism and Transport. *Annual review of nutrition* **35**, 109-134 (2015).
8. Mu, C., *et al.* Spatial Transcriptome and Single Nucleus Transcriptome Sequencing Reveals Tetrahydroxy Stilbene Glucoside Promotes Ovarian Organoids Development Through the Vegfa-Ephb2 Pair. *Advanced science (Weinheim, Baden-Wuerttemberg, Germany)* **12**, e2410098 (2025).
9. Berek, J.S., Kehoe, S.T., Kumar, L. & Friedlander, M. Cancer of the ovary, fallopian tube, and peritoneum. *International journal of gynaecology and obstetrics: the official organ of the International Federation of Gynaecology and Obstetrics* **143 Suppl 2**, 59-78 (2018).
10. Gao, Q., *et al.* Heterotypic CAF-tumor spheroids promote early peritoneal metastasis of ovarian cancer. *The Journal of experimental medicine* **216**, 688-703 (2019).
11. Hagiwara, A., *et al.* Milky spots as the implantation site for malignant cells in peritoneal dissemination in mice. *Cancer research* **53**, 687-692 (1993).
12. McPherson, A., *et al.* Divergent modes of clonal spread and intraperitoneal mixing in high-grade serous ovarian cancer. *Nature genetics* **48**, 758-767 (2016).
13. Tan, D.S., Agarwal, R. & Kaye, S.B. Mechanisms of transcoelomic metastasis in ovarian cancer. *The Lancet. Oncology* **7**, 925-934 (2006).
14. Ford, C.E., Werner, B., Hacker, N.F. & Warton, K. The untapped potential of ascites in ovarian cancer research and treatment. *British journal of cancer* **123**, 9-16 (2020).
15. Bellinger, F.P., *et al.* Changes in selenoprotein P in substantia nigra and putamen in Parkinson's disease. *Journal of Parkinson's disease* **2**, 115-126 (2012).
16. Ma, S., *et al.* YTHDF2 orchestrates tumor-associated macrophage reprogramming and controls antitumor immunity through CD8(+) T cells. *Nature immunology* **24**, 255-266 (2023).

17. Jeong, J.M., *et al.* CX3CR1+ macrophages interact with HSCs to promote HCC through CD8+ T-cell suppression. *Hepatology (Baltimore, Md.)* **82**, 655-668 (2025).
18. Wu, N., *et al.* MerTK(+) macrophages promote melanoma progression and immunotherapy resistance through AhR-ALKAL1 activation. *Science advances* **10**, eado8366 (2024).
19. Lin, S.C., *et al.* Periostin promotes ovarian cancer metastasis by enhancing M2 macrophages and cancer-associated fibroblasts via integrin-mediated NF- κ B and TGF- β 2 signaling. *Journal of biomedical science* **29**, 109 (2022).
20. Wang, C., *et al.* The heterogeneous immune landscape between lung adenocarcinoma and squamous carcinoma revealed by single-cell RNA sequencing. *Signal transduction and targeted therapy* **7**, 289 (2022).
21. Cui, X., Liu, S., Song, H., Xu, J. & Sun, Y. Single-cell and spatial transcriptomic analyses revealing tumor microenvironment remodeling after neoadjuvant chemoimmunotherapy in non-small cell lung cancer. *Molecular cancer* **24**, 111 (2025).
22. Solinas, G., *et al.* Tumor-conditioned macrophages secrete migration-stimulating factor: a new marker for M2-polarization, influencing tumor cell motility. *Journal of immunology (Baltimore, Md. : 1950)* **185**, 642-652 (2010).
23. Bosschaerts, T., *et al.* Alternatively activated myeloid cells limit pathogenicity associated with African trypanosomiasis through the IL-10 inducible gene selenoprotein P. *Journal of immunology (Baltimore, Md. : 1950)* **180**, 6168-6175 (2008).
24. Corvino, D., *et al.* Type I Interferon Drives a Cellular State Inert to TCR-Stimulation and Could Impede Effective T-Cell Differentiation in Cancer. *European journal of immunology* **55**, e202451371 (2025).
25. Zheng, L., *et al.* Pan-cancer single-cell landscape of tumor-infiltrating T cells. *Science (New York, N.Y.)* **374**, abe6474 (2021).
26. Xu, J., *et al.* Single-Cell RNA Sequencing Reveals the Tissue Architecture in Human High-Grade Serous Ovarian Cancer. *Clinical cancer research : an official journal of the American Association for Cancer Research* **28**, 3590-3602 (2022).
27. Chai, C., *et al.* Single-cell transcriptome analysis of epithelial, immune, and stromal signatures and interactions in human ovarian cancer. *Communications biology* **7**, 131 (2024).
28. Ma, W., Hu, Z.B. & Drexler, H.G. Sensitivity of different methods for the detection of myeloperoxidase in leukemia cells. *Leukemia* **8**, 336-342 (1994).
29. Hopert, A., Uphoff, C.C., Wirth, M., Hauser, H. & Drexler, H.G. Specificity and sensitivity of polymerase chain reaction (PCR) in comparison with other methods for the detection of mycoplasma contamination in cell lines. *Journal of immunological methods* **164**, 91100 (1993).

30. Kokosková, B., Mráz, I. & Hyblová, J. Comparison of specificity and sensitivity of immunochemical and molecular techniques for reliable detection of *Erwinia amylovora*. *Folia microbiologica* **52**, 175-182 (2007).

We have addressed each comment and point-by-point responses are provided below:

EDITORS' Comment:	1
REVIEWER 1:	3
REVIEWER 2:	10
REVIEWER 3:	16
REVIEWER 4:	35
REVIEWER 5:	48
REVIEWER 6:	55

In our letter, the reviewer's comments are shown in *blue italics*, all the responses are shown in black Arial. We have inserted some of the revised and newly added figures with their legends, and the corresponding revised text excerpts are provided (**section name, line numbers**) from the clean version of the manuscript for ease of accessibility.

Also, in the revised version of the manuscript, the removed parts from the original paper are shown in double strikethrough and all the changes and additions are highlighted in blue Times New Roman.

Detailed response to the editors' comment as following:

EDITORS' Comment:

"In particular, we would expect your revision to address Reviewer #1's concern on the lack of sample size and treatment response data, Reviewer #3's concern on a lack of mechanistic details and controls to support the proposed crosstalk, Reviewer #4's concern on a lack of direct evidence to support the proposed direct involvement of SELENOP+ macrophages in activating CD8+ T cells, and Reviewer #6's concerns on the relevance of Tregs on HGSOc progression, discrepancy on the SELENOP+ macrophage abundance inferred from the two datasets, and the potential sample composition effect on the BRCA analysis. That is not to say that we consider any of the remaining reviewer concerns to be any less important, and we would expect your revision to address them in full."

Response and revision in the manuscript:

First, we again thank the editors for giving us the option of responding to reviewer comments and revising the manuscript for *Nature Communications*. It has taken us nearly 6 months to complete the studies owing to the technically demanding and time-consuming nature of the *in vivo* experiments. We thank the editorial team for their patience.

(1) To address concerns on the lack of sample size and treatment response data raised by Reviewers #1, we expand sample size by integrating our single-cell RNA sequencing (scRNA-seq) data with a publicly available dataset GSE184880, and the results regarding macrophages remained consistent with our original observations. We also analyzed the GSE266577, which included paired samples from chemo-naive and post-neoadjuvant chemotherapy (interval debulking surgery, IDS) patients. The results revealed a reduced interaction potential between *SELENOP*⁺ macrophages and *GZMH*⁺ precursor exhausted CD8⁺ T cells following neoadjuvant chemotherapy (NACT).

(2) In response to Reviewer #3, we conducted a comprehensive set of additional experiments, including SELENOP antibody validation assay, Western blotting of GPX1 in CD8⁺ T cells, ELISA for SELENOP, selenite pretreatment assay, macrophage-derived CM and CD8⁺ T cell coculture system, and CRISPR-Cas9-mediated EPHB2 deficiency, serum SELENOP concentrations in orthotopic ovarian cancer mice, adoptive cellular transfer assay *in vivo*, along with several other assays suggested by Reviewer #3. These experiments have refined mechanistic details and controls to support the proposed crosstalk.

(3) To address Reviewer #4's comments, we performed targeted experiments that directly support the role of *SELENOP*⁺ macrophages in CD8⁺ T cells activation, including antigen presentation assay *in vitro* and adoptive cellular transfer assay *in vivo*.

(4) In response to Reviewer #6, we analyzed CD4⁺ T cell heterogeneity using our scRNA-seq data and identified a potential involvement of regulatory T cells (Tregs) in the metastatic progression of HGSOc. We explained the discrepancy in *SELENOP*⁺ macrophage abundance between different detection methods. This methodological

difference would not affect data explanation. In addition, we incorporated two previously unassigned LAT samples, classified as *BRCA* wild-type based on newly generated whole-exome sequencing (WES) results, into the subsequent analysis. The results remained consistent with our original observations.

(5) Furthermore, as recommended by Reviewers #2 and #5, we clarified methodological details and improved figure quality and presentation throughout the manuscript. To strengthen the reported data and conclusions, we now include the relevant statistical information in figures and their legends, and have generated a new Supplementary Table (**Source Data**) detailing every result for statistical analysis.

We believe that the revisions made in response have adequately addressed the reviewers' concerns, substantially improving the quality of the manuscript. We are confident that the revised manuscript offers valuable insights into HGSOC, with implications for future research, the development of effective therapeutic strategies, and potential translational applications.

POINT-BY-POINT RESPONSE:

REVIEWER 1:

"The study of Liu et al., provides a comprehensive analysis of the tumor microenvironment (TME) in high-grade serous ovarian cancer (HGSOC) metastasis using advanced techniques like single-cell RNA sequencing and spatial transcriptomics. The authors identified critical cellular and molecular mechanisms influencing tumor progression and immune suppression. The SELENOP⁺ macrophages were found to enhance antitumor immunity by activating CD8⁺ T cells, whereas SPP1⁺ macrophages promoted tumor metastasis. Hypoxia-driven malignant epithelial cells were shown to reprogram SELENOP⁺ macrophages through VEGFA -EPHB2 signaling, diminishing the immune response. This interaction was associated with HGSOC metastasis. Then, anti-VEGFA therapy increased the proportion of SELENOP⁺ macrophages and CD8⁺ T cells, highlighting its potential for improving therapeutic outcomes. The study also revealed that BRCA-mutated HGSOC tumors exhibit a more active immune response compared to BRCA wild-type, suggesting the need for targeted immunotherapeutic strategies.

Key findings of the study include:

- Identification of two distinct macrophage subsets: SELENOP⁺ macrophages (M06) with antigen-processing and presentation capabilities, and SPP1⁺ macrophages (M09) involved in extracellular matrix remodeling. Also, M06 macrophages are associated with better patient outcomes, whereas M09 macrophages are linked to poor outcomes. - Demonstration that SELENOP⁺ macrophages possess immunostimulatory properties, potentially enhancing the cytotoxicity of CD8⁺ T cells via SeP. This is in contrast with pro-tumorigenic effects of SPP1⁺ macrophages.*
- SELENOP⁺ macrophages are enriched in adnexal tumors, but their presence decreases as the tumor metastasizes.*
- Identification of seven consensus metaprograms (MPs) within malignant epithelial cells, each with distinct gene signatures*
- Hypoxia-driven malignant epithelial cells reprogram SELENOP⁺ macrophages via VEGFA-EPHB2, as well as Hypoxia and FTL promote metastasis and therapy resistance. The study shows that VEGFA inhibition increases SELENOP⁺ macrophage proportions in primary tumors and peritoneal metastasis.*
- The study also characterizes the heterogeneity of CD8⁺ T cells and their functional states in different anatomical sites.*

Although the study is nicely presented as such, several major concerns should be addressed before the publication:"

General response: We thank the reviewer for the constructive summary of our study. We appreciate that the key findings and overall significance of the work were acknowledged, which was noted as "*nicely presented*". Below, we provide a point-by-point response to the reviewer's concerns.

“1) *Limited Sample Size: The authors acknowledge that the study may be limited by a relatively small sample size, especially in some groups such as EAT. This may reduce the significance of the findings. We suggest to increase the sample size, at least for profiling the macrophage subsets.*”

Response and revision in the manuscript: We thank the reviewer for this insightful comment. We integrated our scRNA-seq dataset with GSE184880, resulting in a combined dataset including seven EAT, eight LAT and nine Met samples. By analyzing the inferred macrophage subpopulations in the integrated dataset, we confirmed and further substantiated our original observations: a progressive decrease in *SELENOP*⁺ macrophages and a concomitant increase in *SPP1*⁺ macrophages from EAT to LAT and ultimately to Met (**Supplementary Fig. 6d**) (**Results, lines 286-290**).

Supplementary Fig. 6d. Tissue preference of each macrophage clusters estimated by $R_{0/e}$ (integrated GSE184880 with our scRNA-seq data) ($P = 2.2 \times 10^{-16}$). P -values: chi-squared test. $n = 24$ samples, $n = 7$ EAT, $n = 8$ LAT, $n = 9$ Met, biological replicates.

“2) *Lack of Treatment Response Data: While the study provides an in-depth analysis of the TME, it does not incorporate the effects of chemotherapy or other treatments on the TME or link molecular findings to clinical outcomes like platinum-free interval (PFI). All the patient samples investigated by the authors are from treatment-naive cohorts. I suggest that the authors should investigate:*

- *what is the effect of chemotherapy on macrophages distribution and characterization in HGSOV?*

- *how does the chemotherapy affect the functionality of M06 and M09 populations, in terms of immunostimulatory properties?*”

Response and revision in the manuscript: We sincerely thank the reviewer for these thoughtful and important suggestions. We have expanded our analysis using GSE266577 including chemo-naive and respective IDS patients following NACT. Our new findings are summarized as below:

(1) The abundance of macrophages within the myeloid cells decreased following NACT, accompanied a shift from anti-inflammatory to pro-inflammatory states (**Supplementary**

Fig. 15a-c). Inferred *SELENOP*⁺ macrophages showed reduced *SELENOP* expression and lower antigen processing and presentation signature scores, while *SPP1*⁺ macrophages exhibited decreased matrix remodeling signature scores following NACT, with no significant enrichment between the two groups (**Supplementary Fig. 15d-g**) (**Results, lines 483-490**).

Supplementary Fig. 15a-g. **a**, Tissue preference of myeloid clusters estimated by $R_{o/e}$ between chemo-naive and post-neoadjuvant chemotherapy (interval debulking surgery, IDS) groups in GSE266577 ($P = 2.2 \times 10^{-16}$). Violin plots comparing the proinflammatory (**b**) and anti-inflammatory function scores (**c**) between groups. **d**, Tissue preference of inferred *SELENOP*⁺ and *SPP1*⁺ macrophages estimated by $R_{o/e}$ between chemo-naive and IDS groups in GSE266577. Violin plots comparing the *SELENOP* expression (**e**), antigen processing and presentation of *SELENOP*⁺ macrophages (**f**) and matrix remodeling signature scores of *SPP1*⁺ macrophages (**g**) between groups. For **a-g**, a total of $n = 18$ HGSOc paired samples, including $n = 9$ chemo-naive, $n = 9$ IDS, biological replicates. For **a, d**, P -values: two-sided chi-squared test. For **b, c, e-g**, box, median \pm interquartile range, and the whiskers extend up to the minimum and maximum values. P -values: two-sided Wilcoxon test.

(2) Clinically, in our scRNA-seq cohort, *SELENOP*⁺ macrophages were enriched in non-relapsed patients, while *SPP1*⁺ macrophages predominated in relapsed ones. In GSE266577, patients with long PFIs exhibited enrichment of *SELENOP*⁺ macrophages along with higher *SELENOP* expression, whereas the proportion of *SPP1*⁺ macrophages was more abundant in short PFI group, with higher matrix remodeling signature scores (**Supplementary Fig. 15j-m**). The finding related to *SELENOP*⁺ macrophages was validated by mIHC in an independent cohort (**Supplementary Fig. 15n and Supplementary Table 7**). These results expanded our knowledge on the impact of chemotherapy on macrophages in HGSOc (**Results, lines 501-508**).

Supplementary Fig. 15j-n. **j**, Tissue preference of *SELENOP*⁺ and *SPP1*⁺ macrophages estimated by $R_{o/e}$ between groups in our scRNA-seq dataset ($P = 2.2 \times 10^{-16}$). **k**, Tissue preference of inferred *SELENOP*⁺ and *SPP1*⁺ macrophages estimated by $R_{o/e}$ between short and long platinum free interval (PFI) groups in GSE266577 ($P = 2.2 \times 10^{-16}$). Violin plots comparing the *SELENOP* expression of *SELENOP*⁺ macrophages (**l**), and matrix remodeling signature scores of *SPP1*⁺ macrophages (**m**) between groups. **n**, Representative image of ovarian tumor stained by multiplex immunohistochemistry (left), scale bar, 100 μ m and the quantification plots (right). Data represent the mean \pm SD. Two-sided unpaired Student's t test, $n = 10$, biological replicates. For **j**, $n = 7$ solid sites samples in non-replased group, $n = 8$ solid sites samples in replased group, biological replicates. For **km**, a total of $n = 22$ HGSOc samples, including $n = 6$ long PFI, $n = 16$ short PFI, biological replicates. For **k**, P -values: two-sided chi-squared test. For **l**, **m**, box, median \pm interquartile range, and the whiskers extend up to the minimum and maximum values. P -values: two-sided Wilcoxon test.

“3) The authors demonstrate that hypoxia-driven malignant epithelial cells reprogram M06 macrophages, and for this they did experiment using only OVCAR3 cell-derived conditioned media. OVCAR3 is not the best representative of HGSOc cell lines. I suggest the authors should use at least 2 more cell lines, and choose some BRCA1 positive and some BRCA1 negative (See Domcke et al, 2013, <https://www.nature.com/articles/ncomms3126>).”

Response and revision in the manuscript: We thank the reviewer for this important and constructive suggestion. We extended our *in vitro* experiments to include two additional HGSOc cell lines besides OVCAR3 (*BRCA1* wild-type) with distinct *BRCA1* statuses: CAOV3 (*BRCA1* wild-type) and COV362 (*BRCA1* mutant)¹. Different hypoxic culture durations were tested in these cell lines (24 h, 48 h and 72 h) (**Fig. 6e**). Conditioned media (CM) derived from both CAOV3 and COV362 recapitulated the reprogramming effects on THP-1 cells as OVCAR3, characterized by reduced *SELENOP* expression cocultured with the hypoxic CM, which were reversed by VEGFA neutralization. Parallel experiments using hypoxia-CM from ID8 and BMDMs yielded consistent results (**Fig. 6f**). These findings support the robustness and consistency of our observations across multiple genetically diverse ovarian cancer models (**Results, lines 430-434**).

Fig. 6e,f. **e**, Western blot images and quantifications of VEGFA protein expression in cell lines under normoxic or hypoxic conditions for 24 h, 48 h, and 72 h. Two-sided unpaired Student's t test. **f**, SELENOP expression in THP-1 cells or BMDMs after cultured with conditioned medium (CM) from cell lines under normoxic or hypoxic conditions for 48 h or 72 h, with or without VEGFA antibody (VEGFA Ab). One-way ANOVA with Bonferroni post hoc test. Data represent the mean \pm SD. n = 3, biological replicates.

“4) The study does not focus on the detailed interaction of CD8⁺ T cells with myeloid cells, particularly after chemotherapy, as is found in recent study (<https://doi.org/10.1016/j.ccell.2024.11.005>). The study also lacks focus on Myelons. The authors should address these in treatment-naive and post-chemotherapy.”

Response and revision in the manuscript: We thank the reviewer for this highly insightful comment.

(1) We performed the intercellular communication analysis on the dataset GSE266577. Our findings reveal a reduction in the interaction potential between *SELENOP*⁺ macrophages and *GZMH*⁺ CD8⁺ T cells following NACT (**Supplementary Fig. 15h**). Further analysis using t-CyCIF-guided GeoMX data indicate no changes in spatial proximity between these two populations (**Supplementary Fig. 15i**), suggesting a disruption of the immunoregulatory axis between *SELENOP*⁺ macrophages and *GZMH*⁺ CD8⁺ T cells following NACT, which may be attributable, at least in part, to a reduction in their functional interaction potential rather than the changes in spatial proximity (**Results, lines 490-497**).

Supplementary Fig. 15h, i. **h**, Heatmaps illustrating the cell-cell interaction patterns in chemo-naive (upper) and IDS samples (lower). **i**, Boxplots showing the coexistence score in stromal (upper) or tumor (lower) areas of interest of *SELENOP*⁺ macrophages and *GZMH*⁺ precursor exhausted CD8⁺ T cells for between chemo-naive and IDS groups. AOs, areas of interest. For **h**, total $n = 46$, including $n = 26$ chemo-naive, $n = 20$ IDS, biological replicates. For **i** (upper), total $n = 52$, including $n = 27$ Chemo-naive, $n = 25$ IDS, biological replicates. For **i** (lower), total $n = 52$, including $n = 26$ Chemo-naive, $n = 26$ IDS, biological replicates. For **i**, box, median \pm interquartile range, and the whiskers extend up to the minimum and maximum values. P -values: two-sided Wilcoxon test.

(2) Myelonets are spatial structures that form interconnected myeloid networks². As suggested, we collected ten paired omental metastatic lesions samples from HGSOC patients of chemo-naïve and IDS to analyze the Myelonets involved in *SELENOP*⁺ macrophage. The results from mIHC showed no change of the proportion of *SELENOP*⁺ macrophages clustered together in 100 μ m circle in total macrophages, suggesting no detectable effect of chemotherapy on the Myelonets consist of *SELENOP*⁺ macrophage (**Supporting Fig. 1a-c**).

Supporting Fig. 1. Representative examples of ovarian tumor stained by multiplex immunohistochemistry in chemo-naive group (a) and in IDS group (b), scale bar, 100 μ m and the quantification plots (c). Two-sided paired Student test, n =10. ns: not significant.

“5) Address the relevance of treatment impact on macrophage subsets in the discussion as well. It will be interesting to see whether chemotherapy positively or negatively influences M06 or M09 enrichment and their functional roles.”

Response and revision in the manuscript: We thank the reviewer for this valuable suggestion. In the revised manuscript, we have added relevant content in the **Discussion** section (**lines 601-615**).

REVIEWER 2:

“Previous work from this group showed that CD8⁺ T cells were tumour-specific in adnexal tumours but functioned as bystanders in HGSO17 metastasis, which involved paired scRNA seq and scTCR-seq analyses. The intention of this manuscript is to explore the dynamic changes in CD8⁺ T cells during HGSO17 metastasis, i.e., alteration to functional and differentiation states. The manuscript is a complicated paper full of bioinformatic data analysis (densely populated figures that are enriched with complex data). The information shown supports the narrative of the paper, showing intratumoral heterogeneity and complex epigenetic changes that causes remodelling of the tumour microenvironment in HGSO17. The finding that hypoxia-driven malignant epithelial cells could reprogram SELENOP⁺ macrophages is backed up with additional data. The addition of the mouse work adds weight to this study and a mechanism of tumour progression/survival/reduced immune surveillance, showing that VEGFA signalling is a key component and highlights that anti-angiogenic therapies could be used as a treatment these tumours. This work is an important contribution to our understanding of HGSO17 and is a resource for future work.”

General response: We thank the reviewer for the detailed and thoughtful evaluation of our manuscript as *“an important contribution to our understanding of HGSO17 and is a resource for future work”*. We appreciate the recognition of the study's contribution to understanding the tumor microenvironment and immune dynamics in HGSO17, as well as the potential translational relevance of our findings.

“1) Last paragraph of the introduction is very long and acts as an abstract of the full manuscript. This is inappropriate, as the introduction should rather summarise the intention/rationale of the study and study approach.”

Response and revision in manuscript: We thank the reviewer for the helpful suggestion. In response, we have revised the final paragraph of the **Introduction (lines 69-84)**.

“2) Should consider moving some methods into supplementary to reduce the length of the manuscript.”

Response and revision in the manuscript: Thank you for this valuable suggestion. We have moved the methods related to Spatial transcriptomic experiment, WES, Flow cytometry, Western blot and RT-qPCR to the **Supplementary Methods** section in the **Supplementary Information**.

“3) Figure 2B, the dark colour of the extreme -1 and +1 are heavily blackened, which results in not knowing what end of the colour the T06_CD8T-CXCL13 cell population is for the exhausted and TEX boxes, either very dark/blackened red (+1) or very dark/blackened blue (-1). I suggest taking out the black shading from these boxes. While not as bad, same could be said for Figure 2A with several dark blue spots that could be

misinterpreted in the figure as being over-expressed (especially as the spots are very small). I suggest to make all the dots larger, as there is room in the figure for this. Same can be said for Figure 3A and 3D.”

Response and revision in the manuscript: We appreciate the reviewer’s helpful feedback. In response, we have revised the figures accordingly.

Fig. 2a,b. **a**, UMAP plots depicting the clusters of CD8⁺ T cells, colored by cluster. Conventional and unconventional CD8⁺ T cells are highlighted with black dotted lines, respectively (left). Dot plot showing marker genes of conventional CD8⁺ T clusters (right). Dot size representing percent of expressing cells in each cluster and color represents z-score of normalized mean expression level of selected genes. **b**, Heatmap indicating the expression levels of gene signatures of previously reported Texp and Tex cell states, and exhaustion signature in conventional CD8⁺ T subtypes. Tpex, precursor exhausted T cells. Tex, exhausted T cells. For **a, b**, n = all 34 scRNA-seq cohort samples, biological replicates.

Fig. 3a. **a**, UMAP projection of 13 myeloid clusters colored by clusters (left). Dot plot showing expression patterns of selected genes across indicated clusters. Dot size represents percent of expressing cells in each cluster and color represents z-score of normalized mean expression level of selected genes (right). For **a**, n = all 34 scRNA-seq cohort samples, biological replicates.

Fig. 3d. Enriched pathways of each macrophage subset in KEGG databases. n = all 34 scRNA-seq cohort samples, biological replicates.

“4) Line 178: ‘Subcluster T06_CD8T-CXCL13, defined by the expression of CXCL13, CTLA4, and TIGIT, exhibited a transcriptional signature indicative of exhaustion, thus classified as exhausted T cells (Tex) (Fig. 2a,b).’ Please add a sentence to explain why this expression signature is indicative of exhaustion and reference a paper if possible to help readers that might not be as knowledgeable.”

Response and revision in the manuscript: We sincerely thank the reviewer for the careful and insightful comments on our manuscript. We have added a brief explanation in the revised manuscript (**Results, lines 126-128**).

“5) Figure 2D, in annotation ‘expasion index’ is spelt wrong. In this figure there should also be axis annotations that have been omitted.”

Response and revision in the manuscript: We thank the reviewer for pointing this out. We updated Figure 2D (now **Supplementary Fig. 2d**) in the revised manuscript accordingly.

Supplementary Fig. 2d. Clonal expansion of the clusters of conventional CD8⁺ T cells quantified by STARTRAC indices (HRA002184). Each dot represents a patient. Center line indicates the median value, lower and upper hinges represent the 25th and 75th percentiles, respectively and whiskers denote 1.5 × interquartile range. n = 14 HGSOc samples, biological replicates. P-values are calculated by the two-sided Kruskal–Wallis test with Bonferroni post hoc test.

“6) Figure 2F: in result text, lack of annotation in the figure and a different description in the figure legend, i.e., what is ‘ $R_{o/e}$ ’? Currently, it is ambiguous to what the data represents. Can the authors improve the labelling of the figure and description in the text and figure legend to make this result clearer.”

Response and revision in the manuscript: We sincerely appreciate the reviewer’s careful evaluation of Figure 2f.

(1) In our analysis, the ratio of observed to expected cell numbers ($R_{o/e}$) was calculated using the formula: $R_{o/e} = \text{Observed} / \text{Expected}$, as described by Zhang *et al*^β and Chen *et al*^α, to assess tissue preference of specific cell clusters. A cluster was deemed enriched in a tissue if its $R_{o/e}$ ratio exceeded 1. We have included a detailed description in the **Methods** section to ensure clarity for readers unfamiliar with this metric (**Methods, lines 737-744**).

(2) We have provided a clear explanation of $R_{o/e}$ at its first mention (**Supplementary Note 1**) and in the figure legend of **Extended Data Figure. 2c** (Figure. 2f in the original manuscript).

Supplementary Fig. 2c. Tissue preference of each CD8⁺ T cluster, estimated by $R_{o/e}$ (the ratio of observed to expected cell numbers) ($P = 2.2 \times 10^{-16}$, left) ($P = 3.512 \times 10^{-15}$, right). P -values are calculated by the two-sided chi-squared test.

“7) Line 595: Figure 6h, siEPHB is not effective at knocking down siEPHB. I would suggest using other siRNA that are more effective and employ EPHB drug inhibitors to add validity to this work.”

Response and revision in the manuscript: We thank the reviewer for this valuable and constructive comment. We employed the CRISPR-Cas9 system and successfully generated *EPHB2*-deficient THP-1 cells (**Supplementary Fig. 5c**) (**Methods, lines 940-952**). Treatment of THP-1 cells with exogenous recombinant VEGFA protein for 48 h resulted in a decrease in SELENOP expression and secretion, which could be reversed by knockdown of *EPHB2* expression, further supporting that VEGFA-mediated

suppression of SELENOP is largely dependent on EPHB2 signaling (**Fig. 6g,h**) (**Results, lines 434-436**).

Supplementary Fig. 5c. Western blot analysis confirmed CRISPR/Cas9-mediated EPHB2 deletion in THP-1 cells. sgNC: empty vector for negative control of sgRNA. Data represent the mean \pm SD. Two-sided unpaired Student's t test for two groups, n = 3, biological replicates.

Fig. 6g, h. Western blot images and quantifications of SELENOP protein expression after knocking down *EPHB2* by CRISPR-Cas9 in THP-1 cells (**g**) with 50 ng/mL rhVEGAF treated for 48h. sgNC, empty vector for negative control of sgRNA; sgEPHB2, EPHB2 deletion mediated by CRISPR/Cas9; rhVEGAF, recombinant human VEGFA protein. **h**, ELISA showing SELENOP level in the culture supernatants in **g**. For **g, h**, one-way ANOVA with Bonferroni post hoc test, n = 3, biological replicates.

“8) Line 1191: Amplicon lengths should be provided here, and whether dissociation curves were also run routinely to ensure specificity of reactions.”

Response and revision in the manuscript: We thank the reviewer for this helpful suggestion. In the revised manuscript, we have added the amplicon lengths and clarified that dissociation curves were routinely performed to ensure the specificity of the reactions in the **Supplementary Information (Supplementary Methods 10)**. We also determined the amplified FTL (left) and ACTB (right) on 1.5% TBE agarose gels (**Supplementary Fig. 16g**). Comparative dissociation curves are presented in **Supporting Figure 2a below**. In addition, the RT-qPCR products were sequenced and verified by BLAST analysis to further confirm the specificity of amplification (**Supporting Figure 2b**).

Supplementary Fig. 15g. Size and purity determination of amplified *FTL* (upper) and *ACTB* (lower) on 1.5% TBE agarose gels. Amplified products of RT-qPCR from COV362 cell line cultured under normoxic (Line 1, 3, 5) and hypoxic (Line 2, 4, 6) conditions.

Supporting Fig. 2. Evaluation of the specificity of real-time RT-qPCR in quantifying FTL expression. (a) Comparative dissociation curves of amplified *FTL* (left) and *ACTB* (right). (b) BLAST analysis for *FTL* (left) and *ACTB* (right) sequence.

REVIEWER 3:

“This study reveals a novel cross-talk between macrophages, T cells and epithelial cells during the progression of high-grade serous ovarian cancer which centres around SELENOP⁺ macrophages. This cell population was identified by an extensive single cell RNAseq approach considering both the temporal aspect of tumor development as well as different anatomical sides. While this first screening approach is very convincing and well justified to identify the relevant cellular players the mechanistic part analysed by cell culture and mouse experiments remains rather unclear and needs to be substantially improved and extended to support the conclusions drawn by the authors.

The identification of the proposed cross-talk within the tumor microenvironment might not only be relevant for ovarian cancer but could extend to additional cancer types given that this SELENOP⁺ macrophage population has also been identified in lung cancer before. Therefore, the manuscript would be of great interest to the broad readership of Nature Communications. However, in my view, the manuscript still lacks mechanistic details and controls and should be thoroughly revised to address the following comments.”

General response: We sincerely thank the reviewer for the thoughtful and encouraging comments that underscore the significance and broader relevance of our findings as *“of great interest to the broad readership of Nature Communications”*. We fully agree that the mechanistic aspects of the study required further investigation. To address this, the mechanistic analyses have been substantially improved and extended through additional *in vitro* and *in vivo* experiments to support the proposed cross-talk among SELENOP⁺ macrophages, T cells, and epithelial cells during HGSOC progression.

“The whole manuscript should be more clearly structured according to the information provided in the abstract. So far, it is very complicated for the reader to have data provided as figures, extended data figures, and supplementary figure. In my view, a fundamental restructuring of the data is really needed to support the story described in the abstract. Thus, later sections e.g. about immune checkpoints or the section starting at line 436 about dendritic cells are not really related to this story and should be excluded. Also the comparison between different anatomical sides within one patient could be left out. At least it is not getting clear how this relates to the main story line.”

Response and revision in the manuscript: We sincerely thank the reviewer for the constructive comments regarding the manuscript structure. As suggested, they have been removed.

“Based on the abstract, the title could also more specifically describe the novel interactions identified.”

Response and revision in the manuscript: We appreciate the reviewer’s thoughtful suggestion regarding the title. In response, we have revised the title. The new title is “Hypoxia-Driven Remodeling of SELENOP⁺ Macrophages Shapes T Cell Dynamics and

Promotes Ovarian Cancer Metastasis". We hope this revised title more accurately captures the scope and significance of our study.

"Throughout the manuscript: Please introduce the genes/proteins that were discussed in more detail. This is mandatory especially when addressing a broader readership."

Response and revision in the manuscript: We thank the reviewer for this important comment. Accordingly, we have expanded the introduction of SELENOP⁵⁻⁷ (**Results, lines 236-237 and Discussion, lines 547-558**) and VEGFA-EPHB2⁸ (**Results, lines 422-426**).

"Introduction, line 63: Please elaborate on the results gained in the previous publication (Ref 17) a bit more. Which questions remained unanswered after this publication? Why did you now follow up an alternative approach?"

Response and revision in the manuscript: We thank the reviewer for this helpful comment. We have added relevant content in the **Introduction** section (**lines 57-61**).

"Introduction, lines 72-94: This section intensively summarizes the generated results but I miss a clear aim of the study."

Response and revision in the manuscript: We thank the reviewer for pointing this out. We have clarified the aim of our study at this section to better guide the reader through the subsequent summary of our findings (**Introduction, lines 69-84**).

"Results, line 114: The authors conclude that the anatomical site was the primary driver of heterogeneity. How does this relate to the following results which mainly take the tumor progression / stage into account? Here it would have been also helpful for my understanding to comment on the dependency of samples because different anatomical samples are taken from one patient (dependent variables) while different stages are analysed in independent samples from different patients."

Response and revision in manuscript: We appreciate the reviewer's insightful comment. (1) Ovarian cancer (OC) patients are staged based on the locations of different peritoneal metastasis sites according to the FIGO staging system⁹. Studies have shown that different peritoneal metastasis sites are closely related to ovarian cancer staging, treatment resistance, and even prognosis¹⁰⁻¹². Thus, metastasis sites are indicators of progression/stage. We have added this biological background in the revised manuscript (**Introduction, lines 55-56**).

(2) We would like to clarify that the original statement "*anatomical site was the primary driver of heterogeneity*" referred specifically to the distinction between solid and fluid sites-derived samples, rather than to differences among anatomical locations such as adnexal

tumors, peritoneal metastases, or omental lesions. We agree that the phrasing could be misleading, and in the revised manuscript, we have updated the statement for greater clarity as follows:

“To assess global transcriptomic variation across all samples from scRNA -seq cohort, we performed Principal Component Analysis (PCA). The results revealed that the primary source of heterogeneity in the HGSOC tumor microenvironment was the distinction between solid and fluid sites, as samples from these two compartments were clearly separated.” We believe this revised wording more accurately reflects our findings and avoids potential misinterpretation (**Results, lines 100-104**).

(3) In addition, we would like to clarify that the samples included in the PCA were obtained from different individuals rather than from multiple sites in a single patient. We have updated the PCA plot by labeling each data point with the corresponding patient ID (**Fig. 1b**), demonstrating that the observed clustering is not primarily driven by inter-patient variation, but instead reflects site-associated transcriptional differences.

Fig. 1b. The Principal Component Analysis (PCA) plot showing the PC1 vs PC2 projection of all samples at the pseudo-bulk RNA-seq level. For **b**, n = all 34 scRNA-seq cohort samples, biological replicates.

“When looking at progression, the comparison between tumor and fluids is not conclusive. Why was this incorporated into the analyses? Effects of anatomical localization and tumor progression should be more clearly separated from each other (or even leave out anatomical localization results).”

Response and revision in manuscript: We sincerely thank the reviewer for this valuable comment.

(1) Unlike other solid tumors, OC is marked by extensive transcoelomic dissemination and the accumulation of ascites^{13,14}. Peritoneal fluids in patients with OC reflect the FIGO stage IC3, serving as an essential part of the tumor ecosystem⁹. We have added this biological background in the revised manuscript to facilitate readers’ interpretation (**Introduction, lines 55-56**).

(2) We have removed the anatomical localization results (i.e., the comparison between the Met.Per and Met.Ome groups) from the revised manuscript to maintain a clearer and more focused narrative.

“Figure 1 and 2 extensively characterize the samples analysed. Those results could be narrowed down a bit to provide a clearer focus.”

Response and revision in the manuscript: We appreciate the reviewer’s constructive suggestion. In response, we have removed **Fig. 1f** (right), **Supplementary Fig. 1c**, **Supplementary Fig. 2c** in the original manuscript. In addition, **Figure. 2f** has been relocated to **Supplementary Fig. 2c** in the revised manuscript.

“Figure 3: the SELENOP⁺ macrophage cluster was identified. It is very important to keep in mind that this is only gene expression data which does not necessarily relate to protein expression because selenoprotein synthesis is strictly dependent on selenium availability which, under conditions with limited supply, overwrites a transcriptional regulation.”

Response and revision in the manuscript: We thank the reviewer for this important comment. We confirmed SELENOP expression at protein level in macrophages using multiple approaches, such as Western blot, ELISA and flow cytometry analyses, which are specified in the response to *“For all cell culture experiments,...pretreatment for 72 h.”*

“Please provide data showing the specificity of the SELENOP immunohistochemistry staining and Western Blot results (e.g. by staining cells cultured under conditions with and without selenium or by using specific blocking peptides). Unfortunately, there are many unspecific SELENOP antibodies available.”

Response and revision in the manuscript: We thank the reviewer for this important comment. SELENOP is a glycoprotein that typically appears as bands around ~55–60 k Da or at approximately 46 and 52 k Da in Western blot analyses, depending on the tissue type¹⁵. We tested three commercially available anti-SELENOP antibodies (**Supporting Table 1**) by Western blot using lysates from THP-1 cells, THP-1 cells cultured with selenium supplementation, and HEK293 cells (used as a negative control, as previously described¹⁵), respectively. Among these, Antibody 3 demonstrated superior specificity (**Supporting Fig. 3a**). We also assessed antibody 1 and 3 for IHC, via staining human normal gastric tissue adjacent to gastric cancer and normal lymphoid tissue, which are known to express high and low levels of SELENOP, respectively, according to the Human Protein Atlas (HPA). Antibody 3 also showed superior specificity (**Supporting Fig. 3b**). Thus, Antibody 3 was selected for all subsequent experiments to ensure the reliable detection of SELENOP protein expression throughout this study.

Supporting Table 1. Three antibodies for specificity validation.

ID	Company	Catalog number	Applications
Antibody 1	Santa Cruz	sc-376858	WB, IP, IF, IHC, ELISA
Antibody 2	abcam	ab277526	ICC/IF, Flow Cyt, WB, IP
Antibody 3	Invitrogen	PA5-112707	WB, IHC, ELISA

Supporting Fig. 3. Validation of SELENOP Antibodies Specificity. (a) Western blot: the first and last lanes show the molecular weight marker (red line indicates 70 k Da). The second lane: HEK293 cells. The third and fourth lanes contain protein lysates from THP-1 cells and THP-1 cells cultured with selenium supplementation, respectively. (b) Immunohistochemistry staining: SELENOP expression in representative normal human gastric tissue (left panel) and normal human lymphoid tissue (right panel), scale bar, 100μm. Two-way ANOVA test.

“As only very shortly stated in line 334-336, SELENOP is secreted by hepatocytes but also other cell types (including macrophages?) and can be taken up by other cells via the receptor LRP8. SELENOP is broken down to release selenium which is used for the synthesis of other selenoproteins such as GPX or TXNRD. Those parameters definitely need to be analysed to show that SELENOP is taken up by T cells after culturing them with recombinant SELENOP. I would recommend to analyse GPX1 protein expression but other selenoproteins will work as well. Alternatively, the intracellular selenium concentration could be measured.”

Response and revision in the manuscript: We thank the reviewer for this insightful and constructive suggestion. In response, we examined GPX1 expression in CD8⁺ T cells after culturing them with recombinant SELENOP. Our results showed that treatment with 200 ng/mL recombinant SELENOP significantly increased the protein levels of GPX1 in OT-1 T cells (**Supplementary Fig. 4e**). This serves as mechanistic detail supporting the proposed cross-talk described in our study (**Results, lines 245-246**).

Supplementary Fig. 4e. Western blot images and quantifications of GPX1 expression in OT-1 cells following treated with rm SELENOP or an equivalent volume of PBS as control for 48 h. Data represent the mean \pm SD. One-way ANOVA with Bonferroni post hoc test for multiple groups, $n = 3$, biological replicates.

“For all cell culture experiments, it is mandatory to know the basal selenium concentration in FCS and as a proof of concept, cells should be also treated +/- selenite. Using THP1 macrophages, the authors should show that macrophages indeed secrete SELENOP. To strengthen the results, the authors could perform this experiment also with monocytes isolated from patient buffy coats. The conditioned medium from THP1 or other monocytes/macrophages could be used in addition to recombinant SELENOP to treat T cells. Also treating THP1 cells with IFN γ should be performed in cells +/- selenite pretreatment for 72 h.”

Response and revision in manuscript: We appreciate the reviewer’s thoughtful and detailed suggestions.

(1) The basal selenium concentration in fetal calf serum (FCS), measured using an inductively coupled plasma mass spectrometry (ICP-MS) (Agilent 7700x), was 17.389 μ g/L (**Supporting Table 2**), which is consistent with a previous report¹⁶.

Supporting Table 2. Selenium concentration in fetal calf serum (FCS).

Type	Standard level	Samples name	Selenium	Germanium (internal standard)	Average Concentration [ug/L] (diluted 1:50)	Original Concentration [ug/L]
			Concentration [ug/L] (diluted 1:50)	CPS		
CalBlk	1	0	0	332838.7067		
CalStd	2	1	0.74733591	314409.7467		
CalStd	3	5	3.984403166	312339.5467		
CalStd	4	10	10.97822277	301700.8033		
CalStd	5	50	50.83227654	309349.42		
CalStd	6	100	99.53934594	300829.3467		
Sample		1112-	0.495611688	201331.2533		
Sample		1112-	0.143606124	287073.0067	0.347787525	17.38937626
Sample		1112-	0.404144764	268455.8667		

(2) While assessing SELENOP protein levels in THP-1 cells and macrophages differentiated from patient-derived human peripheral blood mononuclear cells (hPBMCs) (MDMs) cells using Western blot, we also performed ELISA to evaluate SELENOP secretion (**Methods, lines 985-993**). The results showed that macrophages do secrete SELENOP, with increased levels upon IFN γ stimulation (**Fig. 4k,I** and **Supplementary Fig. 6f,g**). These results demonstrate that macrophages are indeed capable of synthesizing and secreting SELENOP (**Results, lines 318-322**).

Fig. 4k, I. k, Western blot images (left) and quantification (right) of the levels of *SELENOP* proteins in THP-1 cells under control conditions or after treatment with 70 nM Se, 40 ng/mL IFN γ , or their combination. Se: Na₂Se₃. **I**, ELISA showing *SELENOP* level in the culture supernatants in **k**. For **k** (right), **I**, data represent the mean \pm SD. One-way ANOVA with Bonferroni post hoc test, n = 3, biological replicates.

Supplementary Fig. 6f,g. Western blot analysis and quantification of SELENOP expression in MDMs (f) after treatment with PBS as control or IFN γ . g, ELISA showing SELENOP level in the corresponding culture supernatants in f. For f (right), g, data represent the mean \pm SD. For f, g, two-sided unpaired Student's t test, n = 3, biological replicates.

(3) Macrophage-derived conditioned medium (CM) and CD8 $^+$ T cell coculture system were established to assess the role of macrophage-derived SELENOP (Fig. 3k) (Methods, lines 954-958). CM from *SELENOP*-overexpressing BMDMs modulated via lentiviral transduction (Methods, lines 940-952), significantly enhanced the tumor cell killing capacity of OT-1 T cells (Fig. 3l), which was accompanied by increased GZMB and PRF1 expression (Supplementary Fig. 4f, g). Conversely, *Selenop*-knockdown BMDMs exhibited the opposite effects, which could be reversed by exogenous supplementation of recombinant SELENOP (Supplementary Fig. 4h-j). Parallel experiments using CM from THP-1 cells overexpressed SELENOP and human CD8 $^+$ T cells of HGSOC patients yielded consistent results (Supplementary Fig. 4k-m). Indeed, we further performed antigen presentation assay by directly coculturing OVA-loaded BMDMs with OT-1 T cells (Supplementary Fig. 4n) (Methods, lines 960-965). The results were also consistent with those observed in the macrophage-CM and CD8 $^+$ T cell co-culture system, suggesting the direct priming capacity of *SELENOP* $^+$ macrophages on CD8 $^+$ T cells (Supplementary Fig. 4o-t) (Results, lines 247-267).

Fig. 3k,l. k, (3) Macrophage-derived conditioned medium (CM) and CD8 $^+$ T cell coculture system. l, The total apoptosis rate of ID8-OVA cells induced by corresponding CD8 $^+$ T cells. Data represent the mean \pm SD, two-sided unpaired Student's t test, n = 3, biological replicates.

Supplementary Fig. 4f-t. The proportions of CD3⁺CD8⁺GZMB⁺, CD3⁺CD8⁺PRF1⁺ T cells of OT-1 T cells or human CD8⁺ T cells co-cultured with CM from BMDMs-puroNC and BMDMs-OE-Selenop (**f, g**); BMDMs-BSDNC, BMDMs-KD1-Selenop, BMDMs-KD2-Selenop, BMDMs-KD1-Selenop and BMDMs-KD2-Selenop supplemented with rmSELENOP (**h, i**); THP-1-puroNC and THP-1-OE-SELENOP (**k, l**), respectively, and the total apoptosis rate of ID8-OVA cells (**j**) or OVCAR3 (**m**) induced by corresponding CD8⁺ T cells. **n**, Antigen presentation assay. The proportions of CD3⁺CD8⁺GZMB⁺, CD3⁺CD8⁺PRF1⁺ T cells of OT-1 T cells cocultured directly with BMDMs-puroNC and BMDMs-OE-Selenop (**o, p**); BMDMs-BSDNC, BMDMs-KD1-Selenop, BMDMs-KD2-Selenop, BMDMs-KD1-Selenop and BMDMs-KD2-Selenop supplemented with rmSELENOP (**r, s**), respectively, and the total apoptosis rate of ID8-OVA cells (**q, t**) induced by corresponding CD8⁺ T cells. BMDMs, bone marrow derived macrophages; BMDMs-puroNC, BMDMs transfected with negative control lentivirus; BMDMs-

OE-Selenop, BMDMs overexpressing *Selenop* after lentiviral transfection. BMDMs-BSDNC, BMDMs transfected with negative control shRNA; BMDMs-KD1-Selenop, BMDMs with *Selenop* knockdown after transfected with sh*Selenop*-1. BMDMs-KD2-Selenop, BMDMs with *Selenop* knockdown after transfected with sh*Selenop*-2. THP-1-puroNC, THP-1 transfected with control negative lentivirus; THP-1-OE-SELENOP, THP-1 overexpressing *SELENOP* after lentiviral transfection. For **f-m**, **o-t**, data represent the mean \pm SD. One-way ANOVA with Bonferroni post hoc test for multiple groups, two-sided unpaired Student's t test for two groups, n = 3, biological replicates.

(4) Compared to the untreated controls, pretreatment of THP-1 cells with sodium selenite for 72 h significantly upregulated SELENOP expression and secretion under both IFN γ -stimulated and unstimulated conditions, highlighting the dependence of SELENOP synthesis on selenium availability in macrophages (**Fig. 4k**, I presented in (2)) (**Methods, lines 969-972**) (**Results, lines 318-322**). This result also serves as a proof of concept, aligning with the reviewer's suggestion that "...cells should also be treated \pm selenite."

"Section on epithelial cells (starting at line 461): The rationale for the transition from immune cells to epithelial cells is not getting clear. This section should be shortened. For example, the link to ferritin is not supporting the story line. This is also not mentioned later in the discussion again."

Response and revision in manuscript: We appreciate the reviewer's insightful comment.

(1) We have added a paragraph emphasizing the importance of malignant cells in metastasis, based on the "seed and soil" theory (**Results, lines 337-341**).

(2) As suggested by the reviewer, we have shortened the content related to ferritin in the section on epithelial cells (**Results, lines 381-384**).

"Line 587: Briefly introduce the VEGFA-EPHB2 pathway."

Response and revision in the manuscript: We thank the reviewer for the suggestion. We have added a brief introduction to the VEGFA-EPHB2 pathway, highlighting its relevance in cell-cell communication (**Results, lines 422-426**).

"The knockdown efficiency of EPHB2 is quite poor. Is there an alternative way of inhibiting the pathway? A knockout would be more conclusive to show a dependency."

Response and revision in the manuscript: We appreciate the reviewer's thoughtful and constructive comments. We employed the CRISPR-Cas9 system and successfully generated *EPHB2*-deficient THP-1 cells (**Supplementary Fig. 5c**) (**Methods, lines 940-952**). Treatment of THP-1 cells with exogenous VEGFA protein for 48 h resulted in a decrease in SELENOP expression and secretion, which could be reversed by knockdown of *EPHB2* expression, further supporting that VEGFA-mediated suppression of SELENOP is largely dependent on *EPHB2* signaling (**Fig. 6g,h**) (**Results, lines 434-436**).

Supplementary Fig. 5c. Western blot analysis confirmed CRISPR/Cas9-mediated EPHB2 deletion in THP-1 cells. sgNC: empty vector for negative control of sgRNA. Data represent the mean \pm SD. Two-sided unpaired Student's t test for two groups, n = 3, biological replicates.

Fig. 6g, h. Western blot images and quantifications of SELENOP protein expression after knocking down *EPHB2* by CRISPR-Cas9 in THP-1 cells (**g**) with 50 ng/mL rhVEGAF treated for 48h. sgNC, empty vector for negative control of sgRNA, rh VEGAF: recombinant human VEGFA protein. **h**, ELISA showing SELENOP level in the culture supernatants in **g**. One-way ANOVA with Bonferroni post hoc test, n = 3 Supplementary Fig. 5c.

“Line 594: If the effect of VEGFA treatment should be reversed by the EPHB2 knockdown, please combine the VEGFA treatment with the knockdown to draw this conclusion.”

Response and revision in the manuscript: We thank the reviewer for the insightful suggestion. We performed experiments combining VEGFA treatment with *EPHB2* knockdown in **Fig. 6g, h** presented in the previous response.

“Using patient-derived organoids is an interesting alternative approach but as stated above this should be combined with primary human monocytes/macrophages and not again with the THP1 cell line.”

Response and revision in the manuscript: We thank the reviewer for this valuable suggestion. In response, we combined CM from patient-derived cancer organoid (PDOs) under normoxic or hypoxic conditions with MDMs. CM from hypoxia-treated PDOs for 96 h significantly suppressed SELENOP expression in MDMs (**Fig. 6k**), supporting the role of hypoxia-driven epithelial signals in modulating macrophage states at the individual patient level (**Results, lines 437-439**).

Fig. 6k. SELENOP protein expression in MDMs after cultured with CM from organoids (HGSOc-12) under normoxic or hypoxic conditions (96 h) for 48 h (**k**). Data represent the mean \pm SD. Two-sided unpaired Student's t test, n = 3 technical replicates.

“Line 613, the in vivo model: The SELENOP⁺ macrophages at this stage really express more SELENOP protein as this has been characterized by FACS, in contrast to the RNAseq data which only rely on mRNA data (which might be quite irrelevant for selenoprotein expression). Please use another term for these macrophages to make this difference clearer to the reader.”

Response and revision in the manuscript: We thank the reviewer for this important clarification. We have used the term “F4/80⁺CD11b⁺/SELENOP⁺ macrophages” to refer specifically to the macrophage population phenotypically defined by FACS (**Results, lines 452, and 468**).

“In addition, the authors should provide information about the selenium concentration in the mouse chow. In addition, it would be very interesting to also measure SELENOP concentrations in the serum of mice to exclude a systemic effect of VEGF on SELENOP secreted from the liver.”

Response and revision in manuscript: We appreciate the reviewer's insightful comments. (1) We have added information regarding the selenium concentration (0.13 mg/kg) in the mouse chow in the revised manuscript (**Supporting Figure 4**) (**Methods, lines 1035-1038**).

Supporting Fig. 4. a, Cover page of the test report for the mouse chow. **b**, The selenium concentration in the mouse chow determined by this test (upper) and English translation (lower).

(2) We also measured SELENOP concentrations in the serum of mice (**Methods, lines 1058-1060**). The results showed that VEGFA inhibition did not significantly alter circulating SELENOP levels, thereby excluding a substantial systemic effect of VEGF on SELENOP secreted from the liver in this murine model (**Fig. 7I**) (**Results, lines 460-462**).

Fig. 7I. Blood serum SELENOP concentrations of mice treated with IgG or anti-VEGFA antibody. Data represent the mean \pm SD. *P*-values: two-sided unpaired Student's *t* test.

“Please quantify primary tumor, omental metastasis and peritoneal metastasis formation and not only provide fluorescence images and FACS results.”

Response and revision in manuscript: We thank the reviewer for the helpful suggestion. In the revised manuscript, we have quantified the formation of primary tumors and peritoneal metastases (**Fig. 7d-g**) (**Results, lines 448-451**). Although we initially intended to evaluate omental metastases, no detectable lesions were observed in the omenta of the mice in this murine model.

Fig. 7d-g. Representative images showing characteristic (d) and quantifications (e) of peritoneal metastases (black arrows) in mice treated with IgG (upper), and the absence of peritoneal metastases in mice treated with anti-VEGFA antibody (lower). Representative images (f) and quantification of tumor volumes (g) of primary tumors of mice treated with IgG or anti-VEGFA antibody. Bars, 10 mm. For e, g, data represent the mean \pm SD. P-values: two-sided unpaired Student's t test.

“To really show that macrophage-derived SELENOP is needed to modulate the VEGF effect on tumor metastasis in vivo, the authors should use SELENOP knockout mice with a macrophage specific SELENOP knockout (SELENOPdeltaMye mice).”

Response and revision in the manuscript: We appreciate the reviewer's valuable suggestion.

(1) While we fully agree that a macrophage-specific *SELENOP* knockout mouse model would be an ideal tool for mechanistic studies, it poses inherent limitations, including long time required for model construction, a relatively low success rate and the potential for developmental compensations associated with germline knockouts. Upon thorough evaluation, we employed an established adoptive cellular transfer system, which has been widely used to investigate macrophage-specific functions in tumor immunity¹⁷⁻¹⁹, to address this concern, as illustrated in **Supplementary Fig. 14a** (**Methods, lines 1066-1076**).

(2) Then we found that, compared with the BMDMs group, the proportion of CD45.2+CD3+CD8+GZMB+ T cells decreased, accompanied by reduced frequencies of CD45.1+F4/80+CD11b+SELENOP+ macrophages in the primary tumors of the BMDMs-KD1-Selenop group. Meanwhile, VEGFA inhibition led to a concomitant increase in CD45.2+CD3+CD8+GZMB+ T cells and CD45.1+F4/80+CD11b+SELENOP+ macrophages in the primary tumors of mice injected with BMDMs-KD1-Selenop cells, but had no significant effect in mice injected with BMDMs (**Supplementary Fig. 14b,c**). These findings support the role of SELENOP+ macrophages in priming CD8+ T cell cytotoxicity within the HGSOc tumor microenvironment *in vivo*, and further emphasize a VEGFA-mediated regulatory mechanism that appears to be more pronounced in macrophages with low SELENOP expression (**Results, lines 463-477**).

(3) In addition, together with the observed reduction in tumor metastasis in our another orthotopic models after VEGFA blockade for 35 days (**Fig 7d-g** as presented in the previous response), these findings suggest a potential requirement for macrophage-derived SELENOP in modulating the effects of VEGFA on tumor metastasis (**Results, lines 448-462**).

Supplementary Fig. 14a-c. **a**, Schematic diagram of the protocol for adoptive cellular transfer experiments. Fresh primary tumor tissues were digested and stained for flow cytometry analysis. Shown is the proportion of CD45.1⁺F4/80⁺CD11b⁺SELENOP⁺ cells (**b**) and CD45.2⁺CD3⁺CD8⁺GZMB⁺ cells (**c**) across different sites of each group. Pri. tumor: primary tumor of orthotopic ovarian cancer mouse model; Met.Per: peritoneal metastasis of orthotopic ovarian cancer mouse model. *P*-values are calculated by two-way ANOVA with Tukey post hoc test. *n* = 5 in BMDMs group, *n* = 5 in BMDMs-KD1-Selenop group, *n* = 4 in BMDMs combined with anti-VEGFA group, *n* = 5 in BMDMs-KD1-Selenop combined with anti-VEGFA group.

“Line 621: The last section of the results is again out of the focus of the manuscript and should be clearly shortened.”

Response and revision in the manuscript: We appreciate the reviewer’s insightful comment. In response, we have substantially shortened the subsection “Distinct Immune Cell Subpopulations Enriched in Solid Sites Across Different *BRCA* Statuses” (**Results, lines 512-522**).

“The model provided in Fig. 8 is very interesting but it is not really supported by the data provided in this manuscript (see comment above).”

Response and revision in the manuscript: We thank the reviewer for this important comment. In the revised manuscript, we have incorporated additional experiments as suggested above by reviewer. These new data provide stronger support for the conceptual framework illustrated in **Figure 8** in the revised manuscript.

Fig. 8. Graphical summary of spatiotemporal heterogeneity in TME and model for the role of the malignant cells-macrophages-CD8⁺ T cells axis in immunosuppressive TME formation associated with HGSOC metastasis.

“Discussion.

The discussion of the novel crosstalk should be extended. There is much more literature on SELENOP and its mode of action which should be discussed here. In addition, a putative role of the trace element selenium could be discussed as well.”

Response and revision in the manuscript: We appreciate the reviewer’s valuable suggestion. We have added relevant content in the **Discussion** section (**lines 536-558**).

“For the discussion of hypoxia-induced effects on SELENOP refer to Becker et al., 2014, Hypoxia reduces and redirects selenoprotein biosynthesis.”

Response and revision in the manuscript: In response, we have added the relevant content in the **Discussion** section (**lines 579-581**).

“The second part of the discussion starting in line 688 is again out of topic and has not much to do with the results described in the abstract.”

Response and revision in manuscript: We appreciate the reviewer’s comment. In the revised manuscript, we have removed the content related to immune checkpoints from the **Discussion** section.

*“Minor comments:
Abstract, line 37: malignant epithelial cells.”*

Response and revision in manuscript: Thank you for your suggestion. We have updated the manuscript to change "hypoxia malignant cells" to "malignant epithelial cells", now in **line 37** of the **Abstract** section.

“Please use SELENOP as abbreviation for selenoprotein P as this is the official nomenclature.”

Response and revision in the manuscript: Thank you for your suggestion. We have updated the manuscript to use "SELENOP" as the abbreviation for selenoprotein P, in accordance with the official nomenclature throughout the revised manuscript.

“Line 297: I would mention here again that “elevated phagocytosis function” is another result of the KEGG pathway analysis. Otherwise, one could expect a functional readout.”

Response and revision in manuscript: Thank you for pointing this out. We have removed this sentence to avoid any potential misunderstanding.

“Line 334: Please delete “theoretically”.”

Response and revision in the manuscript: Thank you for the suggestion. We have removed the word “theoretically” in the revised version (**Results, line 237**).

“Fig. 6i is not very informative for the reader.”

Response and revision in manuscript: Thank you for the suggestion. We have added the representative images of PDOs under hypoxic and normoxic conditions, respectively (**Fig. 6I**). We observed a decrease trend in the PDOs volume under hypoxic conditions (**Results, lines 440**).

Fig. 6I. Representative images of PDOs under normoxic (upper) or hypoxic conditions (lower).

“Line 661, Ref 66-68: Only Ref 66 is supporting the respective information about the macrophage subset in lung cancer.”

Response and revision in the manuscript: Thank you for pointing out this point. We have removed Ref 67 and 68 and kept Ref 66, now as Ref 72, which specifically supports the information about the macrophage subset in lung cancer.

“Lines 805-806: specify the title of the section.”

Response and revision in manuscript: We appreciate the reviewer’s suggestion. In response, we have revised the section title to "Cell Type Identification and Characterization" in order to more accurately and concisely reflect the content of this section. The change has been made accordingly in the revised manuscript (**Methods, line 702**).

“Line 1225: Is the amount injected (15 μl) correct?”

Response and revision in the manuscript: Thank you for your comment. We confirm that the injection volume of 15 μl is correct. This protocol was adopted following the methodology described in the published study and has been consistently applied in our

orthotopic implantation experiments²⁰. We have cited this reference in the revised manuscript as Ref 95.

“Line 1238: statistical analysis One-Way ANOVA is missing (is written in figure legends).”

Response and revision in the manuscript: Thank you for pointing out this question. We have included the use of one-way ANOVA in the revised manuscript (**Methods, lines 1082**).

REVIEWER 4:

“This work by Liu et al. uses multiple state of the art single-cell multi-omics and spatial transcriptomics methods to map the spatiotemporal heterogeneity of HGSOC metastasis and tumor microenvironment (TME) and identifies two major populations of tumor-associated macrophages (TAMs): SELENOP⁺ and SPP1⁺, respectively, which undergo dynamic transition in the setting of hypoxic malignant ovarian cells to subvert immune-mediated tumor control, thus facilitating HGSOC metastasis. Some of the findings are consistent with what others have found in various types of cancers but not all of them have been functionally and definitively validated in patients or in vivo model systems. Given the limitation, it is important that, at a minimum, the pro-metastatic role of SPP1⁺ macrophages, as postulated in this study, should be unequivocally demonstrated by taking at least one of the suggested approaches as follows:

- 1. SPP1-targeting antibodies: use neutralizing antibodies against SPP1 to block its activity and downstream signaling.*
- 2. SPP1 knockout: Global and conditional SPP1 KO mice are commercially available.*
- 3. Receptor Blockade: SPP1 interacts with integrins (e.g., α v β 3, α v β 5) and CD44. One could use receptor-blocking antibodies or peptides (e.g., RGD peptides for integrins).”*

General reponse: We thank the reviewer for the constructive and thoughtful assessment of our study. We acknowledge that these approaches could reveal the pro-metastatic role of SPP1⁺ macrophages, however, we consider that this is beyond the scope of this study. Nonetheless, we would like to provide supporting **Fig. 5**, which is under review for another manuscript by our group, to appreciate the reviewer’s taste. As suggested by the reviewer, we conducted *in vivo* experiments, which showed that CD44 blockade (Angstrom6, **Supporting Fig. 5A-C**) and SPP1 antibody (**Supporting Fig. 5D-F**) markedly suppressed ovarian metastasis in mice.

[Redacted]

[Redacted]

“There are also concerns regarding the claim of direct involvement of SELENOP⁺ macrophages in priming T cells, as there is no direct evidence showing that SELENOP⁺ macrophages can activate CD8⁺ T cells. In contrast, previous studies (PMID: 18424738, 20530259) show that SELENOP is highly upregulated in M2 macrophages, which are generally considered more pro -tumorigenic than M1 macrophages.”

Response: We appreciate the reviewer's insightful comments.

(1) Indeed, SELENOP⁺ macrophages have been identified in lung cancer, exhibiting antigen-presenting capabilities with proposed antitumor function associated with CD8⁺ T cells^{21,22}, but without direct evidence. We conducted both *in vitro* and *in vivo* target experiments, which provided direct evidence supporting a role for SELENOP⁺ macrophages in promoting CD8⁺ T cell activation.

(2) Although SELENOP has previously been identified as part of M2-associated gene signatures^{23,24}, numerous studies have demonstrated that tumor-associated macrophages (TAMs) within the TME exhibit remarkable plasticity and heterogeneity, thereby challenging the traditional M1/M2 polarization paradigm. We characterized the phenotype of SELENOP⁺ macrophages in the context of HGSOC.

These results above are detailed in our response to Comment #3 and #7.

“Some additional specific issues of concern:

1.Fig. 2- Out of on the scRNA-seq data (Extended Dara Fig. 2a), only CD8⁺ T cells are analyzed in detail (Fig. 2). However, CD4⁺ T cells are also abundant in the tumor. Why no further analysis on the CD4⁺ T cell population? Are regulatory T cells enriched in late state tumors?”

Response and revision in the manuscript: We thank the reviewer for raising this important point. We have extended our analysis to include CD4⁺ T cell heterogeneity, with a specific focus on regulatory T cells (Tregs). We identified a FOXP3⁺ CD4⁺ Treg subpopulation that is enriched in solid tumor regions and shows a progressive increase from early to late stages (**Extended Fig. 2g-j**). These findings indicate a potential involvement of FOXP3⁺ CD4⁺ Tregs in the metastatic progression of HGSOC (**Results, lines 180-189**).

Supplementary Fig. 2g-j. h, g, UMAP plots depicting the clusters of all CD4⁺T cells, colored by cluster. **h**, Dot plot showing marker genes of CD4⁺T clusters. **i**, Dot plot showing expression patterns of functional signatures across indicated clusters. **j**, Tissue preference of each CD4⁺T clusters, estimated by R_{oe} ($P < 2.2e-16$). P -values are calculated by chi-squared test. For **g, h, i**, $n = 34$ scRNA-seq cohort samples. For **j** (left), $n = 25$ scRNA-seq cohort samples except for Nor.Ovr and PLF.UF. For **j** (right), total $n = 17$ scRNA-seq cohort solid site samples.

“2.Fig. 3h- The resolution of spatial transcriptomics data is too low. T04_CD8T-GZMH gene expression doesn't show clear cluster on the spatial transcriptomics data. It appears that the gene expression is mapped to the whole slide. CD8⁺ T cell clusters can be observed on the immunofluorescence staining slides (Fig. 3j). However, based on the Visium data (Supplementary Fig. 4e), the correlation between SELENOP⁺ macrophage and T04_CD8T-GZMH abundance is very weak ($R = 0.11$, Supplementary Fig. 4e).”

Response and revision in the manuscript: We thank the reviewer for this constructive comment.

(1) We have improved the resolution of the spatial transcriptomics data to optimized the visualization of spatial transcriptomics data. T04_CD8T-GZMH gene expression show clear cluster in **Fig. 3h** in the revised manuscript.

(2) We agree with the reviewer that this correlation appears modest, which may be partially attributed to the limited spatial resolution and mixed-cell capture of the Visium platform. Nevertheless, our integrated analyses, including spatial transcriptomics (**Fig. 3h**, presented in (1)) and immunofluorescence staining (**Fig. 3j**), consistently support the spatial proximity between these two cell populations, providing spatial support for the proposed role of SELENOP⁺ macrophages in promoting CD8⁺ T cell activation.

Fig. 3h. Representative spatial co-localizations of *SELENOP*⁺ macrophage with T04_CD8T-GZMH in tumor spots of slide from adnexal sites (left) and zoom-in image (right) in our spatial transcriptomics cohort.

Fig. 3j. Representative immunofluorescence staining showing co-localization of CD68 (orange), *SELENOP* (green), CD8A (red), GZMH (magenta) and DAPI (blue) in HGSOE samples. Scale bars of each group, 100 µm (left) and 50 µm (right). The white arrow points to the CD8⁺GZMH⁺ cell, and the yellow arrow points to the CD68⁺*SELENOP*⁺ TAM.

“3.Fig. 3k and Supplementary Fig. 4i- To study the effect of SELENOP⁺ macrophages on CD8⁺ T cell function, the authors use SeP protein to stimulate the OT1 cells. There is no dose-dependent change on the GZMB and PRF1. Can SELENOP⁺ macrophages directly prime CD8⁺ T cells? To verify the role of SELENOP⁺ macrophages, co-culturing of OT1-specific SIINFEKL peptide-loaded-SELENOP⁺ macrophages with OT1 cells should be performed. If it is not feasible to directly isolate SELENOP⁺ macrophages, the SELENOP protein can be overexpressed in macrophages, which can then be used to prime CD8⁺ T cells.”

Response and revision in the manuscript:

We sincerely thank the reviewer for the valuable suggestions.

(1) We treated OT-1 T cells with a refined concentration gradient of recombinant SELENOP (**Supplementary Fig. 4b**). The results demonstrated that GZMB and PRF1 expression in OT-1 T cells increased in a dose-dependent manner within the range of 0 to 100 ng/mL of SELENOP (**Supplementary Fig. 4c**), thereby supporting the biological relevance of SELENOP-mediated T cell activation within a defined concentration range (**Results, lines 241-242**).

Supplementary Fig. 4b,c. **b**, CD8⁺ T cell cytotoxicity followed by recombinant SELENOP treatment. **c**, OVA-specific OT-1 T cells were treated with recombinant murine SELENOP or an equivalent volume of PBS as control for 48 h. The proportions of CD3⁺CD8⁺GZMB⁺ or CD3⁺CD8⁺PRF1⁺ T cells of OT-1 T cells were analyzed using flow cytometry. Data represent the mean \pm SD. Brown-Forsythe and Welch ANOVA test. * $1D < 0.05$, ** $1D < 0.01$, *** $1D < 0.001$.

(2) As suggested, we performed the antigen presentation assay (**Supplementary Fig. 4n**) (**Methods, lines 960-965**). SELENOP was first overexpressed (BMDMs-OE-Selenop) or knocked down (BMDMs-KD-Selenop) in BMDMs via lentiviral vectors (**Methods, lines 940-952**). The results showed that OVA-loaded BMDMs-OE-Selenop significantly enhanced GZMB expression, as well as tumor-specific cytotoxic activity in CD8⁺ T cells, compared to the control BMDMs. Meanwhile, OVA-loaded *Selenop*-knockdown BMDMs elicited a significantly weaker cytotoxic response effect, with reduced GZMB and PRF1 expression, as well as the cytotoxic activity of CD8⁺ T cells (**Supplementary Fig. 4o-t**). These findings support the role of SELENOP⁺ macrophages in directly priming CD8⁺ T cells *in vitro* (**Results, lines 262-267**).

Supplementary Fig. 4n-t. n, Antigen presentation assay. The proportions of CD3⁺CD8⁺GZMB⁺, CD3⁺CD8⁺PRF1⁺ T cells of OT-1 T cells cocultured directly with BMDMs-puroNC and BMDMs-OE-Selenop (**o**, **p**); BMDMs-BSDNC, BMDMs-KD1-Selenop, BMDMs-KD2-Selenop, BMDMs-KD1-Selenop and BMDMs-KD2-Selenop supplemented with rmSELENOP (**r**, **s**), respectively, and the total apoptosis rate of ID8-OVA cells (**q**, **t**) induced by corresponding CD8⁺ T cells. BMDMs, bone marrow derived macrophages; BMDMs-puroNC, BMDMs transfected with negative control lentivirus; BMDMs-OE-Selenop, BMDMs overexpressing *Selenop* after lentiviral transfection. BMDMs-BSDNC, BMDMs transfected with negative control shRNA; BMDMs-KD1-Selenop, BMDMs with *Selenop* knockdown after transfected with sh*Selenop*-1. BMDMs-KD2-Selenop, BMDMs with *Selenop* knockdown after transfected with sh*Selenop*-2. Data represent the mean \pm SD. One-way ANOVA with Bonferroni post hoc test for multiple groups, two-sided unpaired Student's t test for two groups, n = 3 biological replicates.

“4.Fig.6f- The experiment should be performed with purified macrophages from the spleen of mice or bone marrow derived macrophages for more physiological relevance.”

Response and revision in the manuscript: We thank the reviewer for this important suggestion. In response, we repeated the experiment using BMDMs. The results also showed that CM from hypoxia-treated ID8 cells markedly suppressed the expression of SELENOP in BMDMs compared to the normoxic controls, an effect reversed by VEGFA neutralization, as in THP-1 cells co-cultured with CM from ovarian cancer cell lines, supporting the physiological relevance of our conclusions (**Fig. 6f**) (**Results, lines 431-434**).

Fig. 6f. SELENOP expression in THP-1 cells or BMDMs after cultured with conditioned medium (CM) from cell lines under normoxic or hypoxic conditions for 48 h, with or without VEGFA antibody (VEGFA Ab) (f). One-way ANOVA with Bonferroni post hoc test, n = 3 biological replicates.

“5.Fig.6g, h- The *EPHB2* knockdown efficiency is very low. However, the SELENOP expression increases dramatically in the siEPHB2 cells. Is it due to off-target effect of siEPHB2? Anti-VEGF neutralizing antibody can be used to verify the role of VEGF signaling on macrophages. The result should be validated using purified macrophages or bone marrow derived macrophages.”

Response and revision in the manuscript: We appreciate the reviewer’s thoughtful and constructive comments.

(1) We appreciate the reviewer’s thoughtful and constructive comments. We employed the CRISPR-Cas9 system and successfully generated *EPHB2*-deficient THP-1 cells (**Supplementary Fig. 5c**) (**Methods, lines 940-952**). Treatment of THP-1 cells with exogenous VEGFA protein resulted in a decrease SELENOP expression and secretion, which could be reversed by knockdown of *EPHB2* expression, further supporting that VEGFA-mediated suppression of SELENOP is largely dependent on *EPHB2* signaling (**Fig. 6g,h**). As suggested, parallel experiments using BMDMs with *Ephb2* knocked down via lentiviral transduction (**Supplementary Fig. 5d**) (**Methods, lines 940-952**) yielded consistent results (**Fig. 6i**) (**Results, lines 434-436**).

(2) Anti-VEGF neutralizing antibody has been used to verify the role of VEGF signaling on macrophages in **Fig. 6f** presented in Comment #4.

Supplementary Fig. 5c, d. **c**, Western blot analysis confirmed CRISPR/Cas9-mediated *EPHB2* deletion in THP-1 cells. sgNC: empty vector for negative control of sgRNA. **d**, Quantification of *EPHB2* by Western blot in BMDMs after *Ephb2* knockdown by sh*Ephb2*-1 (KD1) or sh*Ephb2*-2 (KD2). Data represent the mean \pm SD. Two-sided unpaired Student's t test for two groups, one-way ANOVA with Bonferroni post hoc test for multiple groups, $n = 3$ biological replicates.

Fig. 6g-i. **g**, Western blot images and quantifications of SELENOP protein expression after knocking down *EPHB2* by CRISPR-Cas9 in THP-1 cells (**g**) or by shRNA in BMDMs (**i**) with 50 ng/mL rVEGAF treated for 48h. sgNC, empty vector for negative control of sgRNA, rh (m) VEGAF: recombinant human (mouse) VEGFA protein. **h**, ELISA showing SELENOP level in the culture supernatants in **g**. One-way ANOVA with Bonferroni post hoc test, $n = 3$ biological replicates.

“6.Fig.7b- On day 14, the mouse in the anti-VEGF group seems to show stronger luminescence intensity than the control mouse, suggesting that anti-VEGF may not efficiently suppress primary tumor growth. Quantification of luminescence intensity from multiple mice per group should be provided with statistics.”

Response and revision in the manuscript:

We appreciate the reviewer's thoughtful comments. We have now included the corresponding luminescence data at days 7, 14, and 49 post-inoculation for both the control and anti-VEGF groups, with no statistically significant differences in baseline

luminescence intensity (days 7 and 14). This design allows for an accurate evaluation of anti-VEGF efficacy *in vivo*. We finally observed decreased luminescence intensity and tumor burden in the anti-VEGF group, demonstrating the antitumor effect of VEGFA inhibition. (Fig. 7a-g) (Results, lines 445-451).

Fig. 7a-g. a, Schematic diagram of the protocol for *in vivo* experiments. C57BL/6 mice inoculated with ID8-luciferase cells are treated with 2.5 mg/kg IgG (control mouse; n = 9) or 2.5 mg/kg anti-VEGFA antibody (anti-VEGFA treatment mouse; n = 9). Representative images (b) and quantifications (c) of C57BL/6 mice detected at 7d (left), 14d (middle) and 49d (right) by bioluminescence imaging after inoculation with ID8-luciferase cells. 49d after inoculation with ID8-luciferase cells, mice in both groups were euthanized (n = 9 mice per group). Representative images showing characteristic (d) and quantifications (e) of peritoneal metastases (black arrows) in mice treated with control IgG (upper), and the absence of peritoneal metastases in mice treated with anti-VEGFA antibody (lower). Representative images (f) and quantification of tumor volumes (g) of primary tumors of mice treated with control IgG or anti-VEGFA antibody. Bars, 10 mm. For c, e, g, data represent the mean ± SD. P-values: two-sided unpaired Student's t test for two groups.

“7.Fig.7c- How are SELENOP⁺ macrophages identified using flow cytometry? The gating strategy should be provided. In this mouse model, do SELENOP⁺ macrophages also co-

localize with CD8⁺ T_{pex}? What is the phenotype of this SELENOP⁺ macrophage population? Are they similar to M1 or M2 macrophages? Since the SELENOP⁺ macrophages can be found in this mouse model, those macrophages can be used to test their capacity to prime CD8⁺ T cells in vitro and in vivo.”

Response and revision in the manuscript: We sincerely thank the reviewer for the insightful and constructive comments.

(1) We have clarified that the gating strategy used to identify SELENOP⁺ macrophages by flow cytometry was based on CD45⁺/F4/80⁺CD11b⁺/SELENOP⁺ expression in **Supplementary Fig. 17c**, presented as below.

(2) We performed multiplex immunohistochemistry in the orthotopic ovarian cancer model. The results also revealed a close colocalization between SELENOP⁺ macrophages and CD8⁺GZMH⁺ T cells, similar to the spatial distribution observed in human HGSOc tumors (**Fig. 7k**) (**Results, lines 457-460**).

Fig. 7k. Representative immunofluorescence staining showing co-localization of F4/80 (red), SELENOP (green), CD8A (magenta), GZMH (cyan) and DAPI (blue) in C57BL/6 mice samples.

(3) We analyzed the correlation of the M1 score and M2 score of SELENOP⁺ macrophages in our scRNA-seq data, which revealed the coexistence of both functional phenotypes (**Supporting Fig. 6a**). Then we established a subcutaneous tumor model in female C57BL/6 mice and performed flow cytometric analysis on tumor-infiltrating macrophages. Following identification of these macrophages, we further stratified them into four subpopulations based on surface expression of the canonical M1 marker CD86 and the M2 marker CD206. Notably, we observed an enrichment of a CD86^{hi}_g⁺/CD206^{hi}_g⁺ subset in tumors, suggesting a noncanonical, intermediate polarization state (**Supporting Fig. 6b**).

Supporting Fig. 6. Profile of SELENOP⁺ macrophages. **a**, Scatterplot showing the Spearman correlation of the M1 score and M2 score of SELENOP⁺ macrophages in our scRNA-seq cohort (total $n = 34$ samples). **b**, FACS profiles illustrating the percentage of

F4/80⁺CD11b⁺SELENOP⁺CD86⁺CD206⁺ macrophages in subcutaneous tumors in female C57BL/6 mice (n=3).

(3) The proportion of SELENOP⁺ macrophages obtained by FACS (without permeabilization and stimulation) was low, below 10% (see **Supporting Fig. 7**), rendering their use in subsequent *in vivo* and *in vitro* experiments impractical. Therefore, we followed the reviewer's valuable suggestion in Comment #3 and evaluated their functional capacity both *in vitro* and *in vivo* using *Selenop*-overexpressing macrophages. The *in vitro* experiments to validate the capacity of SELENOP⁺ macrophages to prime CD8⁺ T cells were presented in the response to Comment #3.

Supporting Fig. 7. Proportion of SELENOP⁺ macrophages detected by FACS.

(4) The *in vitro* experiments were performed as below (**Supplementary Fig. 14a**). Then we found that, compared with the BMDMs group, the proportion of CD45.2⁺CD3⁺CD8⁺GZMB⁺ T cells decreased, accompanied by reduced frequencies of CD45.1⁺F4/80⁺CD11b⁺SELENOP⁺ macrophages in the primary tumors of the BMDMs-KD1-Selenop group. Meanwhile, VEGFA inhibition led to a concomitant increase in CD45.2⁺CD3⁺CD8⁺GZMB⁺ T cells and CD45.1⁺F4/80⁺CD11b⁺SELENOP⁺ macrophages in the primary tumors of mice injected with BMDMs-KD1-Selenop cells, but had no significant effect in mice injected with BMDMs (**Supplementary Fig. 14b,c**). These findings support the role of SELENOP⁺ macrophages in priming CD8⁺ T cell cytotoxicity within the HGSOc tumor microenvironment *in vivo*, and further emphasize a VEGFA-mediated regulatory mechanism that appears to be more pronounced in macrophages with low SELENOP expression (**Results, lines 463-477**).

Supplementary Fig. 14. a, Schematic diagram of the protocol for adoptive cellular transfer experiments. Fresh primary tumor tissues were digested and stained for flow cytometry analysis. Shown is the proportion of CD45.1⁺F4/80⁺CD11b⁺SELENOP⁺ cells (**b**) and CD45.2⁺CD3⁺CD8⁺GZMB⁺ cells (**c**) across different sites of each group. Pri. tumor: primary tumor of orthotopic ovarian cancer mouse model; Met.Per: peritoneal metastasis of orthotopic ovarian cancer mouse model. *P*-values are calculated by two-way ANOVA with Tukey post hoc test. *n* = 5 in BMDMs group, *n* = 5 in BMDMs-KD1-Selenop group, *n* = 4 in BMDMs combined with anti-VEGFA group, *n* = 5 in BMDMs-KD1-Selenop combined with anti-VEGFA group.

REVIEWER 5:

“The paper maps the full spatiotemporal progression of high-grade serous ovarian cancer (HGSOC). Thus it improves over the current literature, which predominantly focuses on the tumor microenvironment (TME) of advanced HGSOC. The authors employ multimodal profiling techniques including scRNA-seq, spatial transcriptomics, scTCR-seq, RNA-seq, organoids, and orthotopic syngeneic models. They report spatial proximity between SELENOP⁺ macrophages and precursor exhausted CD8⁺ T cells. Their findings suggest that hypoxia-driven remodeling of the TME (via the malignant epithelial cells) plays a critical role in facilitating metastasis, and indicate the potential for combining anti-VEGFA therapy with immunotherapy.

The question addressed by this work is important, and overall its execution is promising. I think this will be a useful contribution to scientific progress in this field. I particularly appreciate that the authors chose to validate some of their findings using publicly available TCGA data (Fig. 3f). I have a few comments that might help improve the impact of this work and its presentation.”

General response: We thank the reviewer for the positive and constructive feedback as *“The question addressed by this work is important, and overall its execution is promising. I think this will be a useful contribution to scientific progress in this field.”*. We are encouraged by the recognition of our study’s significance and its potential contribution to advancing the understanding of HGSOC progression. In the following, we provide a point-by-point response to the reviewer’s comments regarding the clarity and consistency of data presentation and methodology.

*“1. ****Ratio of observed to expected.**** The usage of the ratio of observed to expected seems inconsistent. For example, in Fig. 2f, each row sums up to 1, whereas in Fig. 3b, this is not the case. This should either be investigated, or the methodology should be explained more clearly.”*

Response and revision in the manuscript: We appreciate the reviewer’s careful observation. We would like to clarify that there is no inconsistency in the methodology regarding the ratio of observed to expected. In both **Fig. 2f** (now **Supplementary Fig. 2c**, left) and **Fig. 3b**, we applied the same calculation method for calculating the ratio of observed to expected cell numbers ($R_{o/e}$) as mentioned in Method section “ $R_{o/e}$ analysis” (**Methods, lines 737-744**). The fact that in **Fig. 2f** (now **Supplementary Fig. 2c**, left) the values in each row appear to sum to approximately 2 is purely coincidental due to rounding to two decimal places for visualization purposes. The raw values do not strictly sum to 2, as presented as **Supporting Table 3** below.

Supporting Table 3. Raw $R_{o/e}$ values of CD8⁺ T cell subclusters across solid and fluid compartments shown in Supplementary Fig. 2c, left.

CD8 ⁺ subclusters	Solid sites	Fluid sites
CD8 ⁺ TRDV2 ⁺ $\gamma\delta$ T (T09)	0.7584212	1.2423671
CD8 ⁺ SLC4A10 ⁺ MAIT (T08)	0.6207193	1.3805184
CD8 ⁺ CXCL13 ⁺ Tex (T06)	1.3975388	0.6011639
CD8 ⁺ IFIT3 ⁺ Tisg (T07)	1.5190566	0.4792496
CD8 ⁺ GNLY ⁺ NK-like (T05)	0.7069565	1.2939998
CD8 ⁺ GZMH ⁺ Tpex (T04)	1.0398623	0.9600077
CD8 ⁺ XCL1 ⁺ Trm (T03)	0.8697651	1.1306599
CD8 ⁺ GZMK ⁺ Tem (T02)	1.0334714	0.9664194
CD8 ⁺ CCR7 ⁺ Tn (T01)	0.8561904	1.1442788

*“2. **Methodology of NMF to define programs:** the way that the metaprograms are defined seems unnecessarily convoluted. I would rather start with the GO approach to get initial cluster annotations, and highlight the similarities with existing signatures afterwards.”*

Response and revision in the manuscript: As recommended, we first employed Gene Ontology (GO) analysis to obtain initial cluster annotations. The corresponding modifications have also been made in the revised manuscript (**Results, lines 348-351**).

*“3. **Grammatical errors and typos:** there are some minor grammatical errors within the text (e. g. lines 31, 57, 115, 318, 320, 370, 444, 585, 630). Fig. 2b refers to precursor exhausted T cells as Texp, whereas Tpex is used everywhere else. Fig. 2d should be “expansion index”. Line 130: “mere” seems like the wrong word here.”*

Response and revision in the manuscript: We thank the reviewer for pointing out these language issues.

(1) Corresponding revisions of grammatical errors within the text have been made in the revised manuscript (**lines 53-54, 100-104, 217-219, 219-220, 277-281**).

(2) Additionally, we corrected **Fig. 2b** and **Fig. 2d** (now **Supplementary Fig. 2d**). We also have removed the word “mere” (**Results, lines 108-112**).

Fig. 2b. Heatmap indicating the expression levels of gene signatures of previously reported Tpex and Tex cell states, and exhaustion signature in conventional CD8⁺ T subtypes.

Supplementary Fig. 2d. Clonal expansion of the clusters of conventional CD8⁺ T cells quantified by STARTRAC indices (PRJCA005422). Each dot represents a patient. Box, median \pm interquartile range. *P*-values: Wilcoxon Rank-Sum test. For **d**, a total of *n* = 14 HGSOc samples, biological replicates.

*“4. ****Presentation of expression data:**** the authors use a rainbow-style colormap to show expression values in Fig. 2a, 3a, 4]. There is however no standardization as to which color represents 0. I would instead encourage the use of a divergent color palette with a clear center, such as the coolwarm palette already used in Fig. 2b. The labels of the colorbar in Fig. 3a are not aligned properly.”*

Response and revision in the manuscript: We thank the reviewer for this helpful suggestion. In response, we have revised the figures.

Fig. 3a. UMAP plots depicting the clusters of CD8⁺ T cells, colored by cluster. Conventional and unconventional CD8⁺ T cells are highlighted with black dotted lines, respectively (left). Dot plot showing marker genes of conventional CD8⁺ T clusters (right). Dot size representing percent of expressing cells in each cluster and color represents z-score of normalized mean expression level of selected genes. For **a**, n = all 34 scRNA-seq cohort samples, biological replicates.

Fig. 4a. UMAP projection of 13 myeloid clusters colored by clusters (left). Dot plot showing expression patterns of selected genes across indicated clusters. Dot size represents percent of expressing cells in each cluster and color represents z-score of normalized mean expression level of selected genes (right). For **a**, n = all 34 scRNA-seq cohort samples, biological replicates.

Fig. 4j. Heatmap showing the RNA expression of transcription factors along the pseudotime trajectory. n = 17 scRNA-seq cohort solid site samples, biological replicates.

“5. **Fig. 2g-i:** it is not clear why clusters T03 and T07 were excluded here.”

Response and revision in the manuscript: We appreciate the reviewer’s insightful comment.

(1) In our work, we focused the pseudotime analysis on CD8⁺ T cell subsets with cytotoxic potential that are responsive to TCR-mediated stimulation in solid tumors, rather than interferon signaling. ISG⁺CD8⁺ TILs exhibit transcriptional inertness to TCR-based stimulation²⁵, and therefore we excluded this population from pseudotime inference in the main analysis. Additionally, subcluster T03 (T_{rm}) was excluded due to its predominant enrichment in ascites, making it less representative of the solid tumor microenvironment (**Results, lines 156-159**).

(2) In parallel, we also expanded the pseudotime analysis to include all conventional CD8⁺ T cell clusters (**Supporting Figure 8**). This comprehensive analysis also revealed two distinct differentiation trajectories. In contrast to the original model—where one trajectory ended in a precursor exhausted/memory state (T02, T04) and the other in terminal exhaustion (T06)—the full-model trajectories culminated in T06 (exhausted) and T07 (ISG⁺) cells, respectively. This reflects the potential different differentiation pathway of CD8⁺ T cells between interferon and TCR-mediated stimulation, in line with previous studies^{25,26}.

Supporting Fig 8. The trajectory plot of all conventional CD8⁺ T cell clusters of tumor sites along pseudotime inferred by Monocle2. Pseudotime of six conventional CD8⁺ T clusters. Each point corresponds to a single cell. Clusters information are shown. The pie charts showing the percentage of clusters in each state.

“6. **Fig. 4d:** Once again, the choice of only showing clusters M05, M06, and M09 is not clearly explained in the text. I would also appreciate it if the same analysis was performed on all macrophage clusters (or at least the TAMs) and put in the supplement (same for the previous point).”

Response and revision in the manuscript: We thank the reviewer for this valuable comment. In Fig. 4d, we focused on clusters M05, M06, and M09 for the pseudotime analysis because our primary aim was to investigate the tumor-associated macrophages

in the solid TME. Among these, M06 and M09 were the two dominant populations specifically enriched in solid tumor sites, while M05 represented monocyte-like precursors that likely give rise to these subsets. As suggested, we have performed additional pseudotime analyses incorporating all monocyte-macrophage clusters. These results showed a similar bifurcating pattern, with two distinct differentiation trajectories culminating in *SELENOP*⁺ and *SPP1*⁺ macrophage-enriched endpoints, supporting our original findings (**Supporting Figure 9**) (**Results, lines 295-300**).

Supporting Fig 9. The trajectory plot of all monocyte macrophage clusters of tumor sites along pseudotime inferred by Monocle2. Pseudotime of five monocyte macrophage clusters. Each point corresponds to a single cell. Clusters information are shown. The pie charts showing the percentage of clusters in each state.

“7. **Fig. 4e:** would benefit from adding plot titles directly in the figure, showing which enrichment analysis belongs to which state.”

Response and revision in the manuscript: We appreciate the reviewer’s suggestion and have revised **Fig. 4e** accordingly to improve its readability and interpretability.

Fig. 4e. Pathway enrichment analysis of the differential genes of cells in different state. n = 17 scRNA-seq cohort solid site samples.

“8. **Fig. 6c:** there should be a distinction between 0 and negative importances.”

Response and revision in the manuscript: We thank the reviewer for pointing this out and have revised **Fig. 6c** accordingly to enhance the interpretability of the results.

Fig. 6c. Heatmap showing the importances among all cell types signature scores in spots of spatial transcriptomics data. Data were summarized from $n = 24$ spatial RNA cohort samples, biological replicates.

“9. ****Supplementary Fig. 6:**** in every other $R_{o/e}$ plot, the numbers are shown, but not in this one.”

Response and revision in the manuscript: We thank the reviewer for noticing this inconsistency. We have now revised **Supplementary Fig. 6c.**

Supplementary Fig. 6c. Tissue preference of each macrophage clusters in adnexal sites and peritoneal foci in cohort (syn52458609), estimated by the $R_{o/e}$ analysis ($P = 2.2 \times 10^{-16}$). P -values are calculated by chi-squared test.

“0. ****Code Availability:**** the authors claim that the code is available (Line 1254), however there is no link to a GitHub repository (or equivalent).”

Response and revision in the manuscript: We thank the reviewer for pointing this out. We have now uploaded the code to a publicly accessible GitHub repository. The link to the repository has been added to the **Code Availability** section of the manuscript (**lines 1110-1112**).

REVIEWER 6:

“Liu et al. characterized the spatial heterogeneity of HGSOc across different progression stages and tissue sites using scRNA -seq and spatial transcriptomics. Their study identifies the co-occurrence and spatial colocalization of SELENOP⁺ macrophages and precursor exhausted CD8⁺ T (Tpex) cells, revealing that SELENOP⁺ macrophages promote CD8⁺ T cell activation in early tumors. They further showed that as the tumor progresses and metastasizes, SELENOP⁺ macrophages are reprogramed into SPP1⁺ macrophages by hypoxia-induced malignant epithelial cells through VEGFA-EPHB2 signaling. Notably, anti-VEGFA therapy restores SELENOP⁺ cells in ovarian cancer cell lines and mouse models. While the study provides insights into the role of macrophage subsets in HGSOc progression, it would benefit from a more rigorous analysis. The key findings are not effectively emphasized, some results are overly descriptive, and the mechanistic validation requires stronger supporting evidence.”

General response: We thank the reviewer for the constructive and encouraging comments that highlight the significance of our study. In response to the concerns raised, we have carefully revised the manuscript to incorporate a more rigorous analysis, better highlight the key findings, present the results in a more objective and concise manner, and provide additional supporting evidence to strengthen the mechanistic validation. Please find our detailed point-by-point responses below.

“Here are my major concerns:

1. The authors highlight the inclusion of early-stage patients as a strength, but similar comparisons have been made in previous studies (such as PMID35675036, 38278958). While one is cited in the discussion, these works should also be discussed. Additionally, the authors should compare their findings with these studies to clarify how their results align or differ from existing literature.”

Response and revision in the manuscript: We thank the reviewer for the valuable comment. We have carefully reviewed the two referenced studies^{27,28}. Although our findings, like those of the aforementioned studies, identified alterations in specific immune cell subsets associated with ovarian cancer progression, our work goes a step further by simultaneously examining both tumor and immune cell compartments within the ovarian cancer microenvironment. Importantly, we revealed a novel cross-talk among macrophages, T cells, and tumor cells associated with the progression of HGSOc. Such mechanistic insights have not been systematically addressed in previous studies (**Discussion, lines 583-589**).

“2. The authors repeatedly use the term “significant” without indicating statistical tests, such as in the R_{0/e} analysis (Fig. 1f, Extended Fig. 2i, j). They should either remove or revise their claims or perform appropriate statistical tests. Specifically, as the R_{0/e} analysis is throughout the study and underpins several conclusions, a chi-square test should be

applied to assess tissue preference. Additionally, the equation used for $R_{o/e}$ analysis should be explicitly provided.”

Response and revision in the manuscript: We thank the reviewer for this insightful comment. In response, we have performed chi-square tests to support our $R_{o/e}$ analyses for assess tissue preference. The statistical methods and corresponding P values have been added to the relevant figures’ legends in the revised manuscript (**Fig. 1e, Fig. 3b, Supplementary Fig. 2c,j, Supplementary Fig. 6a,c,d, Supplementary Fig. 15a,d,j,k and Supplementary Fig. 8a,c**). In addition, the equation used for the $R_{o/e}$ analysis is explicitly provided in the revised manuscript in the **Methods** section (**lines 737-744**). We also removed the term "significant" without indicating statistical tests throughout the revised manuscript.

“3. Although Tregs are not the focus of this study, their absence is notable. The authors should address their relevance in HGSOC progression. Additionally, since T cells and NK cells were initially grouped together, separating them into distinct clusters would allow for more precise characterization.”

Response and revision in the manuscript: We thank the reviewer for this valuable suggestion.

(1) We extended our analysis to include $CD4^+$ T cell heterogeneity, with a specific focus on regulatory T cells (Tregs). We identified a $FOXP3^+$ $CD4^+$ Treg subpopulation that is enriched in solid tumor regions and shows a progressive increase from early to late stages (**Supplementary Fig. 2g-j**) (**Results, lines 180-189**).

(2) We separated T cells and NK cells into distinct clusters in our analysis to enable more precise characterization (**Fig. 1c-f**) (**Supplementary Note 1**).

Supplementary Fig. 2g-j. h, g, UMAP plots depicting the clusters of all CD4⁺T cells, colored by cluster. **h**, Dot plot showing marker genes of CD4⁺ T clusters. **i**, Dot plot showing expression patterns of functional signatures across indicated clusters. **j**, Tissue preference of each CD4⁺T clusters, estimated by $R_{o/e}$ ($P = 2.2 \times 10^{-16}$). P -values are calculated by chi-squared test. For **g, h, i**, $n = 34$ scRNA-seq cohort samples, biological replicates. For **j** (left), $n = 25$ scRNA-seq cohort samples except for Nor.Ovr and PLF.UF, biological replicates. For **j** (right), total $n = 17$ scRNA-seq cohort solid site samples, biological replicates.

Fig. 1c-f. c, Uniform Manifold Approximation and Projection (UMAP) plots of the clustering of main cell types from all samples of scRNA-seq cohort. Cell lineages are highlighted with black dotted lines. Individual cells (dots) are colored by clusters. NK, natural killer; CAF, cancer-associated fibroblasts; SMC, smooth muscle cells. **d**, Track plot indicating selected marker genes in each cell type. **e**, Tissue preference of each major cluster across solid and fluid sites (left), EAT, LAT and Met groups (right), estimated by $R_{o/e}$ ($P = 2.2 \times 10^{-16}$). P -values are calculated by chi-squared test. **f**, Cell proportions of major cell types in different groups, colored by corresponding cell type colors in **c**. For **c, d, f**, $n =$ all 34 scRNA-seq cohort samples, biological replicates. For **e**, $n = 25$ scRNA-seq cohort samples except for Nor.Ovr and PLF.UF, biological replicates.

“4. Quantifying the spatial distribution of macrophage and T_{pex} is valuable. However, the authors should include zoom-in images to visually support the computationally inferred spatial relationships between SELENOP⁺ or SPP1⁺ macrophage and T_{pex}.”

Response and revision in the manuscript: We thank the reviewer for this helpful suggestion. In response, we made the corresponding improvements. The updated images are presented in **Fig. 3h** of the revised manuscript.

Fig. 3h. Representative spatial co-localizations of *SELENOP*⁺ macrophage with T04_CD8T-GZMH in tumor spots of slide from adnexal sites (left) and zoom-in image (right) in our spatial transcriptomics cohort.

*“5. The *SELENOP*⁺ macrophages appear abundant in scRNA-seq and Visium data but are scarce in IF staining (Fig. 3j and 4b). The authors should provide an explanation for this discrepancy, such as differences in detection sensitivity, technical limitations, or biological factors affecting protein versus transcript expression.”*

Response and revision in the manuscript: We thank the reviewers for this crucial comment. The discrepancy in *SELENOP*⁺ macrophage abundance was primarily attributed to the different underlying principles of the detection methods. RNA expression was quantified using methods that yield continuous variables, whereas protein detection by immunofluorescence is generally interpreted in an all-or-none manner at the single-cell level. This methodological difference likely accounts for the apparent discrepancy in data representation²⁹⁻³¹.

*“6. In the BRCA analysis, the BRCAmut group includes two EAT samples, comprising half of the total samples, while the wt group consists only of late-stage samples. This raises the concern that the observed high proportions of T_{pex}, T_{ex}, and *SELENOP*⁺ macrophages in BRCAmut may be driven by differences in sample composition rather than the BRCA mutation itself. The authors should clarify how they account for these confounding factors.”*

Response and revision in the manuscript: We thank the reviewer for this insightful comment. We have removed the two EAT samples from the BRCA mut group to eliminate potential bias related to sample composition. In addition, we performed whole-exome

sequencing (WES) on two samples (OCL3 and OCL6) for which *BRCA* mutation status had not been previously determined (**Supplementary Methods 7**); both were inferred to be *BRCA* wild-type and subsequently incorporated into the BRCAw group (**Supplementary Table 9**), thereby expanding and balancing the cohort. Importantly, even after these adjustments, the enrichment of T_{pex}, T_{ex}, and *SELENOP*⁺ macrophages in BRCAmut group remained evident (**Supplementary Fig. 16a,c**). We also observed consistently elevated cytotoxicity scores in CD8⁺CXCL13⁺ T_{ex} (T06) and CD8⁺GZMH⁺ T_{pex} (T04) cells, as well as higher proinflammatory scores in *SELENOP*⁺ macrophages (M06) in the BRCAmut group compared to BRCAw (**Supplementary Fig. 16b,d**). We believe these revisions directly address the reviewer's concern and further reinforce the robustness of our conclusions (**Results, lines 510-522** and **Supplementary Note 6**).

Supplementary Fig. 16. **a**, Tissue preference of conventional CD8⁺ T subpopulation between BRCAmut and BRCAw groups, estimated by $R_{o/e}$ ($P = 9.725e-10$). P -values are calculated by chi-squared test. **b**, Violin plots showing tumor specific and cytotoxic signature scores of T04_CD8T-GZMH (left) and T06_CD8T-CXCL13 (right) between BRCAmut and BRCAw groups. P values calculated by two-sided Wilcoxon tests. BRCAmut, *BRCA* mutant group. BRCAw, *BRCA* wild-type group; **c**, Tissue preference of macrophage subpopulation between BRCAmut and BRCAw groups, estimated by $R_{o/e}$ ($P = 0.00000971$). P -values are calculated by chi-squared test. **d**, Violin plots showing proinflammatory (left) and antigen processing and presentation (right) signature scores of *SELENOP*⁺ macrophages between BRCAmut and BRCAw groups. P values calculated by two-sided Wilcoxon tests. For **a-d**, data were summarized from all $n = 6$ samples. BRCAw: $n = 4$, BRCAmut: $n = 2$, biological replicates. Box of violin plot represents median \pm interquartile range, and the whiskers

extend up to the minimum and maximum values.

“7. Given the significant growth differences between PDOs and the OVACR3 tumor cell line, both models were subjected to 48-hour normal and hypoxic cultures. Have alternative hypoxic culture durations been explored? Additionally, the authors should specify the cell numbers of PDOs and OVACR3 used to prepare the conditioned medium.”

Response and revision in the manuscript: We appreciate the reviewer’s thoughtful question.

(1) We assessed VEGFA expression in different ovarian cancer cell lines exposed to hypoxia for 24 h, 48 h, and 72 h, and in patient-derived organoids (PDOs) for 72 h and 96 h. VEGFA expression in OVCAR3 cells and PDOs showed the most pronounced upregulation under hypoxic conditions at 48 and 96 h, respectively, compared to the normoxic controls (**Fig. 6e, j**). In addition, we also assessed VEGFA expression in CAOV3, COV362 and ID8 exposed to hypoxia for different durations, with VEGFA expression most significantly upregulated in hypoxic CAOV3 at 48 h, and in hypoxic COV362 and ID8 at 72 h, compared to normoxia (**Fig. 6e**) (**Results, lines 427-431, 436-438**).

(2) We observed that SELENOP expression in the THP-1 cells were markedly reduced following co-culture with hypoxia-conditioned media (CM) from OVCAR3 (48 h), compared to normoxic controls. This effect could be rescued by VEGFA neutralization (**Fig. 6f**). CM collected from the 96-h hypoxia-treated PDOs also significantly suppressed SELENOP expression in MDMs (**Fig. 6k**). Parallel experiments using hypoxia-conditioned media from CAOV3, COV362 and ID8 yielded consistent results (**Fig. 6f**). Also, we observed a decrease trend in PDOs volume under hypoxic conditions (**Fig. 6l**) (**Results, lines 431-434, line 439-440**).

(4) We also specified the cell numbers of OVACR3 (5×10^5) and PDOs (~ 200 clones) used to prepare the CM (**Methods, lines 996, lines 1023-1025**).

Fig. 6e,f, j-l. Western blot images and quantifications of VEGFA in cell lines under normoxic or hypoxic conditions for 24 h, 48 h, and 72 h (**e**). SELENOP in THP-1 or BMDMs after cultured with conditioned medium (CM) from cell lines under normoxic or hypoxic conditions for 48 h, with or without VEGFA antibody (VEGFA Ab) (**f**). Western blot images and quantifications of VEGFA in PDOs under normoxic or hypoxic conditions for 72 h and 96 h (**j**). SELENOP in MDMs after cultured with CM from PDOs under normoxic or hypoxic conditions (96 h) (**k**). MDMs, human monocyte-derived macrophages. **l.** Representative images of PDOs under normoxic (upper) or hypoxic conditions (lower). Data represent the mean \pm SD. For **e, j, k**, two-sided unpaired Student's t test, $n = 3$. For **f**, one-way ANOVA with Bonferroni post hoc test, $n = 3$.

“8. The authors show that SELENOP⁺ macrophages activate CD8⁺ T cells via SeP, but validation is limited. While exogenous SeP increased GZMB and PRF1 expression and co-culture assays assessed cytotoxicity, these alone are insufficient. Blocking SeP secretion in macrophages would strengthen mechanistic validation.”

Response and revision in the manuscript: We thank the reviewer for this insightful suggestion. In response, we generated the *Selenop* knockdown BMDMs by lentivirus transfection, with reduced SELENOP expression (**Supplementary Fig. 5a**) (**Methods, lines 940-952**). Macrophage-derived CM and CD8⁺ T cell coculture system were established as illustrated in **Fig 3k** (**Methods, lines 954-958**). CM from *Selenop*-knockdown BMDMs (KD1 and KD2) induced a weaker cytotoxic response effect, with reduced both GZMB and PRF1 expression, as well as the cytotoxic activity of CD8⁺ T cells, which could be reversed by supplementation of rmSELENOP (**Supplementary Fig. 4h-j**). Meanwhile, we also established *Selenop*-overexpressing macrophages (BMDMs-OE-Selenop and THP-1-OE-SELENOP) (**Supplementary Fig. 5a,b**) (**Methods, lines 940-952**). The CM from these cells enhanced the cytotoxic activity of CD8⁺ T cells (**Fig. 3l** and **Supplementary Fig. 4f,g,k-m**). In fact, we also performed the antigen presentation assay, with OVA-loaded BMDMs cocultured directly with OT-1 T cells (**Supplementary Fig. 4n**) (**Methods, lines 960-965**). The results were also consistent

with those observed in the macrophage-conditioned medium and CD8⁺ T cell co-culture system. Taken together, these results support the ability of *SELENOP*⁺ macrophages to activate CD8⁺T cells via *SELENOP* (**Supplementary Fig. 4o-t**) (**Results, lines 247-267**).

Supplementary Fig. 5a,b. **a**, Quantification of *SELENOP* by Western blot in BMDMs after infected with a lentivirus encoding *Selenop* (**left**), or after *Selenop* knockdown by sh*Selenop-1* (KD1) or sh*Selenop-2* (KD2) (**right**). **b**, Quantification of *SELENOP* by Western blot in THP-1 after infected with a lentivirus encoding *SELENOP*. Data represent the mean \pm SD. Two-sided unpaired Student's t test for two groups, one-way ANOVA with Bonferroni post hoc test for multiple groups, $n = 3$, biological replicates.

Fig. 3k,l. **k**, CD8⁺ T cell cytotoxicity when cocultured with macrophage-derived conditioned medium (CM). **l**, The total apoptosis rate of ID8-OVA cells induced by corresponding CD8⁺ T cells. Data represent the mean \pm SD, two-sided unpaired Student's t test, $n = 3$, biological replicates.

Supplementary Fig. 4f-t. The proportions of CD3⁺CD8⁺GZMB⁺, CD3⁺CD8⁺PRF1⁺ T cells of OT-1 T cells or human CD8⁺ T cells co-cultured with CM from BMDMs-puroNC and BMDMs-OE-Selenop (**f, g**); BMDMs-BSDNC, BMDMs-KD1-Selenop, BMDMs-KD2-Selenop, BMDMs-KD1-Selenop and BMDMs-KD2-Selenop supplemented with rmSELENOP (**h, i**); THP-1-puroNC and THP-1-OE-SELENOP (**k, l**), respectively, and the total apoptosis rate of ID8-OVA cells (**j**) or OVCAR3 (**m**) induced by corresponding CD8⁺ T cells. **n**, Antigen presentation assay. The proportions of CD3⁺CD8⁺GZMB⁺, CD3⁺CD8⁺PRF1⁺ T cells of OT-1 T cells cocultured directly with BMDMs-puroNC and BMDMs-OE-Selenop (**o, p**); BMDMs-BSDNC, BMDMs-KD1-Selenop, BMDMs-KD2-Selenop, BMDMs-KD1-Selenop and BMDMs-KD2-Selenop supplemented with rmSELENOP (**r, s**), respectively, and the total apoptosis rate of ID8-OVA cells (**q, t**) induced by corresponding CD8⁺ T cells. BMDMs, bone marrow derived macrophages; BMDMs-puroNC, BMDMs transfected with negative control lentivirus; BMDMs-OE-Selenop, BMDMs overexpressing *Selenop* after lentiviral transfection. BMDMs-BSDNC, BMDMs

transfected with negative control shRNA; BMDMs-KD1-Selenop, BMDMs with *Selenop* knockdown after transfected with sh*Selenop*-1. BMDMs-KD2-Selenop, BMDMs with *Selenop* knockdown after transfected with sh*Selenop*-2. THP-1-puroNC, THP-1 transfected with control negative lentivirus; THP-1-OE-SELENOP, THP-1 overexpressing *SELENOP* after lentiviral transfection. For **b-d**, **f-n**, **p-u**, data represent the mean \pm SD. For **f-n**, **p-u**, One-way ANOVA with Bonferroni post hoc test for multiple groups, two-sided unpaired Student's t test for two groups, n = 3, biological replicates.

“9. The bioluminescence images show only one representative mouse per group. Can the authors confirm whether the same mouse was imaged on days 7, 14, and 42? To enhance data reliability, it is recommended that images of at least three mice per group be included.”

Response and revision in the manuscript: We agree and thank the reviewer for the comment. We have included the same three representative mice images per group on days 7, 14, and 49 in the revised figures (Fig. 7a-c). This modification improves the reliability and robustness of the data (Results, lines 445-448).

Fig. 7a-c. **a**, Schematic diagram of the protocol for *in vivo* experiments. C57BL/6 mice inoculated with ID8-luciferase cells are treated with 2.5 mg/kg IgG (control mouse; n = 9) or 2.5 mg/kg anti-VEGFA antibody (anti-VEGFA treatment mouse; n = 9). Representative images (**b**) and quantifications (**c**) of C57BL/6 mice detected at 7d (left), 14d (middle) and 49d (right) by bioluminescence imaging after inoculation with ID8-luciferase cells. 49d after inoculation with ID8-luciferase cells, mice in both groups were euthanized (n = 9 mice per group). For **c**, data represent the mean \pm SD. P-values: two-sided unpaired Student's t test.

“Minor points:

1. The authors should provide the full name of ISG and PARPi and others if they are first mentioned.”

Response and revision in the manuscript: We thank the reviewer for pointing this out. We have now provided the full names for all abbreviations (e.g., ISG and PARPi) at their first appearance in the manuscript (**Introduction, lines 50-51**) (**Supplementary Note 2**).

“2. The authors should indicate the cell fate on top of the corresponding barplot in Fig4e for better readability.”

Response and revision in the manuscript: We thank the reviewer for pointing this out. We have added the corresponding cell fate annotations above each bar in **Fig. 4e** to improve readability.

Fig. 4e. Pathway enrichment analysis of the differential genes of cells in different state. n = 17 scRNA-seq cohort solid site samples, biological replicates.

“3. In the first paragraph of the first results section, line 12, there is an extra “and” at the end of the sentence.”

Response and revision in the manuscript: We thank the reviewer for the careful reading and have removed the redundant “and” in the first paragraph of the results section (**lines 106**).

REFERENCES OF RESPONSE

1. Domcke, S., Sinha, R., Levine, D.A., Sander, C. & Schultz, N. Evaluating cell lines as tumour models by comparison of genomic profiles. *Nature communications* **4**, 2126 (2013).
2. Launonen, I.M., *et al.* Chemotherapy induces myeloid-driven spatially confined T cell exhaustion in ovarian cancer. *Cancer cell* **42**, 2045-2063.e2010 (2024).

3. Zhang, L., *et al.* Lineage tracking reveals dynamic relationships of T cells in colorectal cancer. *Nature* **564**, 268-272 (2018).
4. Cheng, S., *et al.* A pan-cancer single-cell transcriptional atlas of tumor infiltrating myeloid cells. *Cell* **184**, 792-809.e723 (2021).
5. Burk, R.F. & Hill, K.E. Selenoprotein P-expression, functions, and roles in mammals. *Biochimica et biophysica acta* **1790**, 1441-1447 (2009).
6. Mizuno, A., *et al.* An efficient selenium transport pathway of selenoprotein P utilizing a high-affinity ApoER2 receptor variant and being independent of selenocysteine lyase. *The Journal of biological chemistry* **299**, 105009 (2023).
7. Burk, R.F. & Hill, K.E. Regulation of Selenium Metabolism and Transport. *Annual review of nutrition* **35**, 109-134 (2015).
8. Mu, C., *et al.* Spatial Transcriptome and Single Nucleus Transcriptome Sequencing Reveals Tetrahydroxy Stilbene Glucoside Promotes Ovarian Organoids Development Through the Vegfa-Ephb2 Pair. *Advanced science (Weinheim, Baden-Wurttemberg, Germany)* **12**, e2410098 (2025).
9. Berek, J.S., Kehoe, S.T., Kumar, L. & Friedlander, M. Cancer of the ovary, fallopian tube, and peritoneum. *International journal of gynaecology and obstetrics: the official organ of the International Federation of Gynaecology and Obstetrics* **143 Suppl 2**, 59-78 (2018).
10. Gao, Q., *et al.* Heterotypic CAF-tumor spheroids promote early peritoneal metastasis of ovarian cancer. *The Journal of experimental medicine* **216**, 688-703 (2019).
11. Hagiwara, A., *et al.* Milky spots as the implantation site for malignant cells in peritoneal dissemination in mice. *Cancer research* **53**, 687-692 (1993).
12. McPherson, A., *et al.* Divergent modes of clonal spread and intraperitoneal mixing in high-grade serous ovarian cancer. *Nature genetics* **48**, 758-767 (2016).
13. Tan, D.S., Agarwal, R. & Kaye, S.B. Mechanisms of transcoelomic metastasis in ovarian cancer. *The Lancet. Oncology* **7**, 925-934 (2006).
14. Ford, C.E., Werner, B., Hacker, N.F. & Warton, K. The untapped potential of ascites in ovarian cancer research and treatment. *British journal of cancer* **123**, 9-16 (2020).
15. Bellinger, F.P., *et al.* Changes in selenoprotein P in substantia nigra and putamen in Parkinson's disease. *Journal of Parkinson's disease* **2**, 115-126 (2012).
16. Parant, F., *et al.* Selenium Discrepancies in Fetal Bovine Serum: Impact on Cellular Selenoprotein Expression. *International journal of molecular sciences* **25**(2024).
17. Ma, S., *et al.* YTHDF2 orchestrates tumor-associated macrophage reprogramming and controls antitumor immunity through CD8(+) T cells. *Nature immunology* **24**, 255-266 (2023).

18. Jeong, J.M., *et al.* CX3CR1+ macrophages interact with HSCs to promote HCC through CD8+ T-cell suppression. *Hepatology (Baltimore, Md.)* **82**, 655-668 (2025).
19. Wu, N., *et al.* MerTK(+) macrophages promote melanoma progression and immunotherapy resistance through AhR-ALKAL1 activation. *Science advances* **10**, eado8366 (2024).
20. Lin, S.C., *et al.* Periostin promotes ovarian cancer metastasis by enhancing M2 macrophages and cancer-associated fibroblasts via integrin-mediated NF- κ B and TGF- β 2 signaling. *Journal of biomedical science* **29**, 109 (2022).
21. Wang, C., *et al.* The heterogeneous immune landscape between lung adenocarcinoma and squamous carcinoma revealed by single-cell RNA sequencing. *Signal transduction and targeted therapy* **7**, 289 (2022).
22. Cui, X., Liu, S., Song, H., Xu, J. & Sun, Y. Single-cell and spatial transcriptomic analyses revealing tumor microenvironment remodeling after neoadjuvant chemoimmunotherapy in non-small cell lung cancer. *Molecular cancer* **24**, 111 (2025).
23. Solinas, G., *et al.* Tumor-conditioned macrophages secrete migration-stimulating factor: a new marker for M2-polarization, influencing tumor cell motility. *Journal of immunology (Baltimore, Md. : 1950)* **185**, 642-652 (2010).
24. Bosschaerts, T., *et al.* Alternatively activated myeloid cells limit pathogenicity associated with African trypanosomiasis through the IL-10 inducible gene selenoprotein P. *Journal of immunology (Baltimore, Md. : 1950)* **180**, 6168-6175 (2008).
25. Corvino, D., *et al.* Type I Interferon Drives a Cellular State Inert to TCR-Stimulation and Could Impede Effective T-Cell Differentiation in Cancer. *European journal of immunology* **55**, e202451371 (2025).
26. Zheng, L., *et al.* Pan-cancer single-cell landscape of tumor-infiltrating T cells. *Science (New York, N.Y.)* **374**, abe6474 (2021).
27. Xu, J., *et al.* Single-Cell RNA Sequencing Reveals the Tissue Architecture in Human High-Grade Serous Ovarian Cancer. *Clinical cancer research : an official journal of the American Association for Cancer Research* **28**, 3590-3602 (2022).
28. Chai, C., *et al.* Single-cell transcriptome analysis of epithelial, immune, and stromal signatures and interactions in human ovarian cancer. *Communications biology* **7**, 131 (2024).
29. Ma, W., Hu, Z.B. & Drexler, H.G. Sensitivity of different methods for the detection of myeloperoxidase in leukemia cells. *Leukemia* **8**, 336-342 (1994).
30. Hopert, A., Uphoff, C.C., Wirth, M., Hauser, H. & Drexler, H.G. Specificity and sensitivity of polymerase chain reaction (PCR) in comparison with other methods for the detection of mycoplasma contamination in cell lines. *Journal of immunological methods* **164**, 91100 (1993).

31. Kokosková, B., Mráz, I. & Hyblová, J. Comparison of specificity and sensitivity of immunochemical and molecular techniques for reliable detection of *Erwinia amylovora*. *Folia microbiologica* **52**, 175-182 (2007).